# GLM-130B: An Open Bilingual Pre-Trained Model

**Aohan Zeng**$^{\diamond\dagger*}$, **Xiao Liu**$^{\diamond\dagger*}$, **Zhengxiao Du**$^{\diamond\dagger}$, **Zihan Wang**$^{\diamond}$, **Hanyu Lai**$^{\diamond}$, **Ming Ding**$^{\diamond}$,
**Zhuoyi Yang**$^{\diamond}$, **Yifan Xu**$^{\diamond}$, **Wendi Zheng**$^{\diamond}$, **Xiao Xia**$^{\diamond}$, **Weng Lam Tam**$^{\diamond\S}$, **Zixuan Ma**$^{\diamond}$,
**Yufei Xue**$^{\S}$, **Jidong Zhai**$^{\diamond}$, **Wenguang Chen**$^{\diamond}$, **Zhiyuan Liu**$^{\diamond}$, **Peng Zhang**$^{\S}$,
**Yuxiao Dong**$^{\diamond\ddagger}$, **Jie Tang**$^{\diamond\ddagger}$

Tsinghua University$^{\diamond}$      Zhipu.AI$^{\S}$

## Abstract

We introduce GLM-130B, a bilingual (English and Chinese) pre-trained language model with 130 billion parameters. It is an attempt to open-source a 100B-scale model at least as good as GPT-3 (davinci) and unveil how models of such a scale can be successfully pre-trained. Over the course of this effort, we face numerous unexpected technical and engineering challenges, particularly on loss spikes and divergence. In this paper, we introduce the training process of GLM-130B including its design choices, training strategies for both efficiency and stability, and engineering efforts. The resultant GLM-130B model offers significant outperformance over GPT-3 175B (davinci) on a wide range of popular English benchmarks while the performance advantage is not observed in OPT-175B and BLOOM-176B. It also consistently and significantly outperforms ERNIE TITAN 3.0 260B—the largest Chinese language model—across related benchmarks. Finally, we leverage a unique scaling property of GLM-130B to reach INT4 quantization without post training, with almost no performance loss, making it the first among 100B-scale models and more importantly, allowing its effective inference on 4×RTX 3090 (24G) or 8×RTX 2080 Ti (11G) GPUs, the most affordable GPUs required for using 100B-scale models. The GLM-130B model weights are publicly accessible and its code, training logs, related toolkit, and lessons learned are open-sourced at https://github.com/THUDM/GLM-130B/.

## 1 Introduction

Large language models (LLMs), particularly those with over 100 billion (100B) parameters (Brown et al., 2020; Thoppilan et al., 2022; Rae et al., 2021; Chowdhery et al., 2022; Wang et al., 2021), have presented attractive scaling laws (Wei et al., 2022b), where emergent zero-shot and few-shot capabilities suddenly arose. Among them, GPT-3 (Brown et al., 2020) with 175B parameters pioneers the study of 100B-scale LLMs by strikingly generating better performance with 32 labeled examples than the fully-supervised BERT-Large model on a variety of benchmarks. However, both GPT-3 (and many other closed-sourced 100B-scale ones)—the model itself—and how it can be trained, have been thus far intransparent to the public. It is of critical value to train a high-quality LLM of such scale with both the model and training process shared with everyone.

We thus *aim to pre-train an open and highly-accurate 100B-scale model* with ethical concerns in mind. Over the course of our attempt, we have come to realize that pre-training a dense LLM at such a scale raises numerous unexpected technical and engineering challenges compared to training 10B-scale models, in terms of pre-training efficiency, stability, and convergence. Similar difficulties

---

$^{*}$The two lead authors AZ and XL contributed equally ({zengaohan,shawliu9}@gmail.com)
$^{\dagger}$Work partially done when AZ, XL, and ZD interned at Zhipu.AI.
$^{\ddagger}$Team leads: YD and JT. Corresponding author: JT (jietang@tsinghua.edu.cn)
 For detailed author contributions, please refer to Appendix E.

Figure 1: A summary of the performance evaluation and ethical studies.

Table 1: A comparison between GLM-130B and other 100B-scale LLMs and PaLM 540B. (LN: layer norm.; FPF: floating-point format; MIP: multi-task instruction pre-training; CN : Chinese)

| Model | Open-source | Architecture & Data | | | Training | | Inference | |
| --- | --- | --- | --- | --- | --- | --- | --- | --- |
| | | Objective | LN | Major Lang. | FPF | Stabilization | Quantization | GPU Needed |
| GPT-3 175B | × | | | English | FP16 | *undisclosed* | *undisclosed* | *undisclosed* |
| OPT-175B | ✓ | GPT | Pre-LN | English | FP16 | Manual Adjusting | INT8 | 8 × 3090 |
| BLOOM-176B | ✓ | | | Multi-lingual | BF16 | Embedding Norm | INT8 | 8 × 3090 |
| PaLM 540B | × | GPT | Pre-LN | English | BF16 | Manual Adjusting | *undisclosed* | *undisclosed* |
| GLM-130B | ✓ | GLM (Blank Infilling & MIP) | Deep-Norm | Bilingual (EN & CN) | FP16 | Embedding Gradient Shrink | INT4 | 4 × 3090 or 8 × 1080 Ti |

have also been concurrently observed in training OPT-175B (Zhang et al., 2022) and BLOOM-176B (Scao et al., 2022), further demonstrating the significance of GPT-3 as a pioneer study.

In this work, we introduce the pre-training of a 100B-scale model—GLM-130B, in terms of engineering efforts, model design choices, training strategies for efficiency and stability, and quantization for affordable inference. As it has been widely realized that it is computationally unaffordable to empirically enumerate all possible designs for training 100B-scale LLMs, we present not only the successful part for training GLM-130B but also many of the failed options and lessons learned. Particularly, the training stability is the decisive factor in the success of training models of such a scale. Different from practices such as manually adjusting learning rates in OPT-175B and using embedding norm in the sacrifice of performance in BLOOM-176B, we experiment with various options and find the strategy of embedding gradient shrink can significantly stabilize the training of GLM-130B.

Specifically, GLM-130B is a bilingual (English and Chinese) bidirectional dense model with 130 billion parameters, pre-trained over 400 billion tokens on a cluster of 96 NVIDIA DGX-A100 (8×40G) GPU nodes between May 6 and July 3, 2022. Instead of using the GPT-style architecture, we adopt the General Language Model (GLM) algorithm (Du et al., 2022) to leverage its bidirectional attention advantage and autoregressive blank infilling objective. Table 1 summarizes the comparison between GLM-130B, GPT-3 and another two open-source efforts—OPT-175B and BLOOM-176B, as well as PaLM 540B (Chowdhery et al., 2022)—a 4× larger model—as a reference.

Altogether, the conceptual uniqueness and engineering efforts enable GLM-130B to exhibit performance that surpasses the level of GPT-3 on a wide range of benchmarks (in total 112 tasks) and also outperforms PaLM 540B in many cases, while outperformance over GPT-3 has not been observed in OPT-175B and BLOOM-176B (Cf. Figure 1 left). For zero-shot performance, GLM-130B is better than GPT-3 175B (+5.0%), OPT-175B (+6.5%), and BLOOM-176B (+13.0%) on LAMBADA (Paperno et al., 2016), and achieves 3× better performance than GPT-3 on Big-bench-lite (Srivastava et al., 2022). For the 5-shot MMLU (Hendrycks et al., 2021) tasks, it is better than GPT-3 175B (+0.9%) and BLOOM-176B (+12.7%). As a bilingual LLM also in Chinese, it offers significantly better results than ERNIE TITAN 3.0 260B (Wang et al., 2021)—the largest Chinese LLM—on 7 zero-shot CLUE (Xu et al., 2020) datasets (+24.26%) and 5 zero-shot FewCLUE (Xu et al., 2021) ones (+12.75%). Importantly, as summarized in Figure 1 right, GLM-130B as an open model is associated with *significantly less bias and generation toxicity than its 100B-scale counterparts*.

Finally, we design GLM-130B to empower as many people as possible to conduct 100B-scale LLM studies. First, instead of using 175B+ parameters as OPT and BLOOM, the 130B size is decided because such a size supports inference on a single A100 (8×40G) server. Second, to further lower the GPU requirements, we quantize GLM-130B into INT4 precision without post training while OPT and BLOOM can only reach INT8. Due to a unique property of the GLM architecture, GLM-130B's INT4 quantization introduces negligible performance degradation, e.g., -0.74% on LAMBADA and even +0.05% on MMLU, making it still better than the uncompressed GPT-3. This enables GLM-

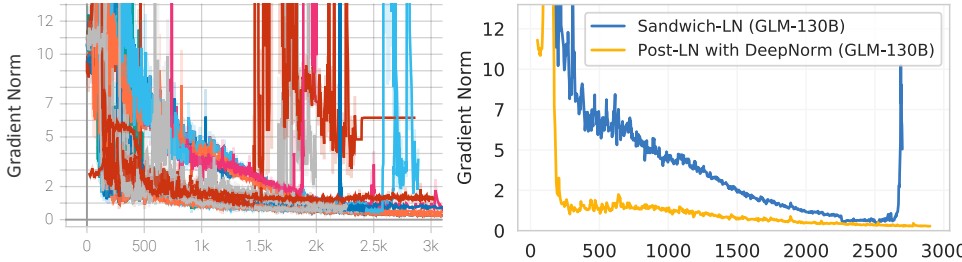

(a) More than 30 failed preliminary trials at 100B-scale    (b) Final decisive trials: Sandwich-LN v.s. DeepNorm

Figure 3: Trials on different LayerNorms for GLM-130B training. It turns out that DeepNorm is the most stable one, as it has small gradient norm and does not spike in the early stage training.

130B's fast inference with performance guarantee on a server of 4×RTX 3090 (24G) or 8×RTX 2080 Ti (11G), *the most affordable GPU required for using 100B-scale LLMs to date.*

We open-source the model checkpoints, code, training logs, related toolkits, and lessons learned.

## 2    THE DESIGN CHOICES OF GLM-130B

The architecture of a machine learning model defines its inductive bias. However, it has been realized that it is computationally unaffordable to explore various architectural designs for LLMs. We introduce and explain the unique design choices of GLM-130B.

### 2.1    GLM-130B'S ARCHITECTURE

**GLM as Backbone.**  Most recent 100B-scale LLMs, such as GPT-3, PaLM, OPT, and BLOOM, follow the traditional GPT-style (Radford et al., 2019) architecture of decoder-only autoregressive language modeling. In GLM-130B, we instead make an attempt to explore the potential of a bidirectional GLM—General Language Model (Du et al., 2022)—as its backbone.

GLM is a transformer-based language model that leverages autoregressive blank infilling as its training objective. Briefly, for a text sequence $x = [x_1, \cdots, x_n]$, text spans $\{s_1, \cdots, s_m\}$ are sampled from it, each of which $s_i$ denotes a span of consecutive tokens $[s_{i,1}, \cdots, s_{i,l_i}]$ and is replaced (i.e., corrupted) with a single mask token to form $x_{\text{corrupt}}$. The model is asked to recover them autoregressively. To allow interactions between corrupted spans, their visibility to each other is decided by a randomly sampled permutation on their order.

GLM's bidirectional attention over unmasked (i.e., uncorrupted) contexts distinguishes GLM-130B from GPT-style LLMs in which the unidirectional attention is used. To support both understanding and generation, it mixes two corruption objectives, each indicated by a special mask token:

- **[MASK]**: short blanks in sentences whose lengths add up to a certain portion of the input.
- **[gMASK]**: random-length long blanks at the end of sentences with prefix contexts provided.

Conceptually, the blank infilling objective with bidirectional attention enables a more effective comprehension of contexts than GPT-style models: when using [MASK], GLM-130B behaves as BERT (Devlin et al., 2019) and T5 (Raffel et al., 2020); when using [gMASK], GLM-130B behaves similarly to PrefixLM (Liu et al., 2018; Dong et al., 2019).

Empirically, GLM-130B offers a record-high accuracy of 80.2% on zero-shot LAMBADA by outperforming both GPT-3 and PaLM 540B in Figure 2. By setting the attention mask, GLM-130B's unidirectional variant is comparable to GPT-3 and OPT-175B. Our observations are in line with existing findings (Liu et al., 2018; Dong et al., 2019).

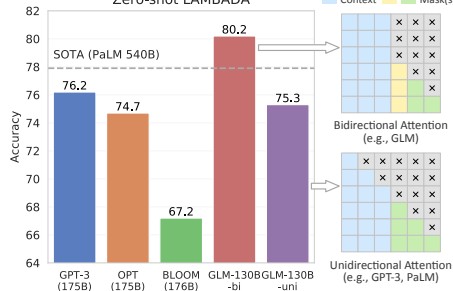

Figure 2: GLM-130B and LLMs of similar scale on zero-shot LAMBADA language modeling. Details on GLM's bidirectional attention are provided in Du et al. (2022).

**Layer Normalization (LN, Ba et al. (2016)).** Training instability is one major challenge for training LLMs (Zhang et al., 2022; Scao et al., 2022; Chowdhery et al., 2022) (Cf. Figure 10 in Appendix for collapses in training several 100B-scale models). A proper choice of LNs can help stabilize

the training of LLMs. We experiment with existing practices, e.g., Pre-LN (Xiong et al., 2020), Post-LN (Ba et al., 2016), Sandwich-LN (Ding et al., 2021), which are unfortunately incapable of stabilizing our GLM-130B test runs (Cf. Figure 3 (a) and Appendix B.2 for details).

Our search is later focused on Post-LN due to its favorable downstream results in preliminary experiments though it does not stabilize GLM-130B. Fortunately, one of the attempts on Post-LN initialized with the newly-proposed DeepNorm (Wang et al., 2022b) generates promising training stability. Specifically, given the number of GLM-130B's layers $N$, we adopt DeepNorm($\boldsymbol{x}$) = LayerNorm($\alpha \cdot \boldsymbol{x}$ + Network($\boldsymbol{x}$)), where $\alpha = (2N)^{\frac{1}{2}}$, and apply the Xavier normal initialization with the scaling factor of $(2N)^{-\frac{1}{2}}$ to `ffn`, `v_proj` and `out_proj`. Additionally, all bias terms are initialized to zero. Figure 3 shows it significantly benefits the training stability of GLM-130B.

**Positional Encoding and FFNs.** We empirically test different options for positional encoding (PE) and FFN improvements in terms of both training stability and downstream performance (Cf. Appendix B.3 for details). For PEs in GLM-130B, we adopt Rotary Positional Encoding (RoPE, Su et al. (2021)) rather than ALiBi (Press et al., 2021). To improve FFNs in Transformer, we pick GLU with the GeLU (Hendrycks & Gimpel, 2016) activation as the replacement.

## 2.2 GLM-130B's Pre-Training Setup

Inspired by recent works (Aribandi et al., 2022; Wei et al., 2022a; Sanh et al., 2022), the GLM-130B pre-training objective includes not only the self-supervised GLM autoregressive blank infilling) but also multi-task learning for a small portion of tokens. This is expected to help boost its downstream zero-shot performance.

**Self-Supervised Blank Infilling (95% tokens).** Recall that GLM-130B uses both [MASK] and [gMASK] for this task. Each training sequence is applied with one of them independently at a time. Specifically, [MASK] is used to mask consecutive spans in 30% of training sequences for blank infilling. The lengths of spans follow a Poisson distribution ($\lambda = 3$) and add up to 15% of the input. For the other 70% sequences, the prefix of each sequence is kept as context and [gMASK] is used to mask the rest of it. The masked length is sampled from the Uniform distribution.

The pre-training data includes 1.2T Pile (train split) (Gao et al., 2020) English, 1.0T Chinese Wudao-Corpora (Yuan et al., 2021), and 250G Chinese corpora (including online forums, encyclopedia, and QA) we crawl from the web, which form a balanced composition of English and Chinese contents.

**Multi-Task Instruction Pre-Training (MIP, 5% tokens).** T5 (Raffel et al., 2020) and ExT5 (Aribandi et al., 2022) suggest that multi-task learning in pre-training can be more helpful than fine-tuning, we thus propose to include a variety of instruction prompted datasets including language understanding, generation, and information extraction in GLM-130B's pre-training.

Compared to recent works (Wei et al., 2022a; Sanh et al., 2022) that leverage multi-task prompted fine-tuning to improve zero-shot task transfer, MIP only accounts for 5% tokens and is set in the pre-training stage to prevent spoiling LLMs' other general ability, e.g., unconditional free generation. Specifically, we include 74 prompted datasets from (Sanh et al., 2022; Wang et al., 2022a), listed in Appendix C and Table 12. GLM-130B users are suggested to avoid evaluating its zero-shot and few-shot capabilities on these datasets according to the criterion illustrated in Section 5.

## 2.3 Platform-Aware Parallel Strategies and Model Configurations

GLM-130B is trained on a cluster of 96 DGX-A100 GPU (8×40G) servers with a 60-day access. The goal is to pass through as many tokens as possible, as a recent study (Hoffmann et al., 2022) suggests that most existing LLMs are largely under-trained.

**The 3D Parallel Strategy.** The data parallelism (Valiant, 1990) and tensor model parallelism (Shoeybi et al., 2019) are the de facto practices for training billion-scale models (Wang & Komatsuzaki, 2021; Du et al., 2022). To further handle the huge GPU memory requirement and the decrease in overall GPU utilization resulted from applying tensor parallel between nodes—as 40G rather than 80G A100s are used for training GLM-130B, we combine the pipeline model parallelism with the other two strategies to form a 3D parallel strategy.

The pipeline parallelism divides the model into sequential stages for each parallel group, and to further minimize bubbles introduced by pipeline, we leverage the PipeDream-Flush (Narayanan et al.,

2021) implementation from DeepSpeed (Rasley et al., 2020) to train GLM-130B with a relative big global batch size (4,224) to reduce time and GPU memory wasting. Through both numerical and empirical examinations, we adopt 4-way tensor parallelism and 8-way pipeline parallelism (Cf. Appendix B.4 for details). Following the calculation in (Chowdhery et al., 2022), we report hardware FLOPs utilization (HFU) of 43.3% and model FLOPs utilization (MFU) of 32.5% due to re-materialization.

**GLM-130B Configurations.** We aim to enable our 100B-scale LLM to run a single DGX-A100 (40G) node in FP16 precision. Based on the hidden state dimension of 12,288 we adopt from GPT-3, the resultant model size has to be no more than 130B parameters, thus GLM-130B. To maximize GPU utilization, we configure the model based on the platform and its corresponding parallel strategy. To avoid insufficient memory utilization in the middle stages due to the additional word embedding at both ends, we balance the pipeline partition by removing one layer from them, making 9×8-2=70 transformer layers in GLM-130B.

During the 60-day access to the cluster, we manage to train GLM-130B for 400 billion tokens (roughly 200 billion each for Chinese and English) with a fixed sequence length of 2,048 per sample. For the [gMASK] training objective, we use a context window of 2,048 tokens. For the [MASK] and multi-task objectives, we use a context window of 512 and concatenate four samples together to cater the 2,048-sequence-length. We warm-up the batch size from 192 to 4224 over the first 2.5% samples. We use AdamW (Loshchilov & Hutter, 2019) as our optimizer with $\beta_1$ and $\beta_2$ set to 0.9 and 0.95, and a weight decay value of 0.1. We warm up the learning rate from $10^{-7}$ to $8 \times 10^{-5}$ over the first 0.5% samples, then decay it by a $10\times$ cosine schedule. We use a dropout rate of 0.1 and clip gradients using a clipping value of 1.0 (Cf. Table 11 for the full configurations).

## 3 THE TRAINING STABILITY OF GLM-130B

The training stability is the decisive factor in GLM-130B's quality, which is also largely impacted by the number of tokens it passes through (Hoffmann et al., 2022). Thus, given the computing usage constraint, there has to be a trade-off between efficiency and stability with regard to floating-point (FP) formats: low-precision FP formats (e.g., 16-bit precision—FP16) improve computing efficiency but are prone to overflow and underflow errors, resulting in training collapses.

**Mixed-Precision.** We follow the common practice of a mixed-precision (Micikevicius et al., 2018) strategy (Apex O2), i.e., FP16 for forwards and backwards and FP32 for optimizer states and master weights, to reduce the GPU memory usage and improve training efficiency. Similar to OPT-175B and BLOOM-176B (C.f. Figure 10 in Appendix), the training of GLM-130B faces frequent loss spikes resulted from this choice, which tends to become increasingly frequent as the training goes on. The precision related spikes are often without clear reasons: some recover on their own; others come with a portent of suddenly soaring gradient norm and eventually a spike or even NaN in loss. OPT-175B attempted to fix by manually skipping data and adjusting hyper-parameters; BLOOM-176B did so via the embedding norm technique (Dettmers et al., 2021). We spent months to empirically investigate the spikes and realize that a few issues emerge when transformers scale up:

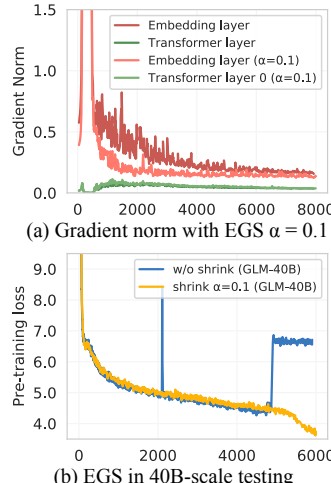

(a) Gradient norm with EGS $\alpha = 0.1$

(b) EGS in 40B-scale testing

Figure 4: EGS reduces gradient scale and variance to stabilize LLMs' pre-training.

First, the transformer main branch's value scale can be extremely large in deeper layers if using Pre-LN. This is addressed in GLM-130B by using DeepNorm based Post-LN (Cf. Section 2.1), which makes the value scale always bounded.

Second, the attention scores grow so large that they exceed FP16's range, as the model scales up. There are a few options to overcome this issue in LLMs. In CogView (Ding et al., 2021), PB-Relax is proposed to remove bias terms and deduct extremum value in attention computation to avoid the problem, which unfortunately does not help avoid disconvergence in GLM-130B. In BLOOM-176B, the BF16 format is used instead of FP16, due to its wide range of values on NVIDIA Ampere GPUs (i.e., A100). However, BF16 consumes ~15%

more run-time GPU memory than FP16 in our experiments due to its conversion to FP32 in gradient accumulation, and more importantly it is not supported on other GPU platforms (e.g., NVIDIA Tesla V100), limiting the accessibility of produced LLMs. Another option from BLOOM-176B is to apply embedding norm with BF16, but in sacrifice of a significant penalty on model performance, as they notice that embedding norm can harm model's zero-shot learning (Cf. Section 4.3 in (Scao et al., 2022)).

**Embedding Layer Gradient Shrink (EGS).** Our empirical search identifies that the gradient norm can serve as an informative indicator of training collapses. Specifically, we find that a training collapse usually lags behind a "spike" in gradient norm by a few training steps. Such spikes are usually caused by the embedding layer's abnormal gradients, as we observe that its gradient norm is often several magnitude larger that those of other layers in GLM-130B's early stage training (Cf. Figure 4 (a)). In addition, it tends to fluctuate dramatically in the early training. The problem is handled in vision models (Chen et al., 2021) via freezing the patch projection layer. Unfortunately, we cannot freeze the training of the embedding layer in language models.

Finally, we find the gradient shrink on embedding layers could overcome loss spikes and thus stabilize GLM-130B's training. It is first used in the multi-modal transformer CogView (Ding et al., 2021). Let $\alpha$ be the shrinking factor, the strategy can be easily implemented via word_embedding = word_embedding $* \alpha +$ word_embedding.detach() $* (1 - \alpha)$. Figure 4 (b) suggests that empirically, setting $\alpha = 0.1$ wipes out most spikes we would have met, with negligible latency.

In fact, the final GLM-130B training run only experiences three late-stage loss divergence cases, though it fails numerous times due to hardware failures. For the three unexpected spikes, it turns out further shrinking the embedding gradient can still help stabilize the GLM-130B training. See the training notes and Tensorboard logs in our code repository for details.

# 4 GLM-130B INFERENCE ON RTX 2080 TI

One of the major goals of GLM-130B is to lower the hardware requirements for accessing 100B-scale LLMs without efficiency and effectiveness disadvantages.

As mentioned, the model size of 130B is determined for running the full GLM-130B model on a single A100 (40G×8) server, rather than the high-end A100 (80G×8) machine required by OPT-175B and BLOOM-176B. To accelerate GLM-130B inference, we also leverage FasterTransformer (Timonin et al., 2022) to implement GLM-130B in C++. Compared to the PyTorch implementation of BLOOM-176B in Huggingface, GLM-130B's decoding inference is 7-8.4× faster on the same single A100 server. (Cf. Appendix B.5 for details).

**INT4 Quantization for RTX 3090s/2080s.** To further support popularized GPUs, we attempt to compress GLM-130B as much as possible while maintaining performance superiority, particularly via quantization (Zafrir et al., 2019; Shen et al., 2020; Tao et al., 2022), which introduces little task-agnostic performance drops for generative language models.

Typically, the practice is to quantize both model weights and activations to INT8. However, our analysis in Appendix B.6 suggests that LLMs' activations may contain extreme outliers. Concurrently, the emergent outliers in OPT-175B and BLOOM-176B are also discovered (Dettmers et al., 2022), which influence only about 0.1% feature dimensions and are thus solved by matrix multiplication decomposition for the outlying dimensions. Differently, there exist about 30% outliers in GLM-130B's activations, making the technique above far less efficient. Thus, we decide to focus on the quantization of model weights (i.e., mostly linear layers) while keeping the FP16 precision

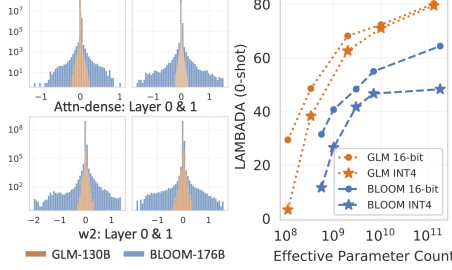

Figure 5: (Left) `attn-dense` and `w2`'s weight distributions; (Right) GLM-130B's INT4 weight quantization scaling law.

for activations. The quantized model is dynamically converted to FP16 precision at runtime, introducing a small computational overhead but greatly reducing the GPU memory usage for storing model weights.

Table 2: Left: Quantized GLM-130B's performance on several benchmarks; Right: INT4 quantized GLM-130B's inference speed (encode and decode) with FasterTransformer.

| Model Precision | GLM-130B | | | GPT-3 |
| --- | --- | --- | --- | --- |
| | FP16 | INT8 | INT4 | FP16 |
| MMLU (acc, ↑) | 44.75 | 44.71 | 44.80 | 43.9 |
| LAMBADA (acc, ↑) | 80.21 | 80.21 | 79.47 | 76.2 |
| Pile (a part, BPB, ↓) | 0.634 | 0.638 | 0.641 | 0.74 |

| GPU Type | 128 Enc./Dec. | | 512 Enc./Dec, | |
| --- | --- | --- | --- | --- |
| 8 × A100 (40G) | 0.15s | 4.29s | 0.18s | 17.7s |
| 8 × V100 (32G) | 0.31s | 6.97s | 0.67s | 28.1s |
| 4 × RTX 3090 (24G) | 0.37s | 8.16s | 1.30s | 32.3s |
| 8 × RTX 2080 Ti (11G) | 0.39s | 6.77s | 1.04s | 27.3s |

Excitingly, we manage to reach the INT4 weight quantization for GLM-130B while existing successes have thus far only come to the INT8. Memory-wise, by comparing to INT8, the INT4 version helps additionally save half of the required GPU memory to 70GB, thus allowing GLM-130B inference on 4 × RTX 3090 Ti (24G) or 8 × RTX 2080 Ti (11G). Performance-wise, Table 2 left indicates that without post-training at all, the INT4-version GLM-130B experiences almost no performance degradation, thus maintaining the performance advantages over GPT-3 on common benchmarks.

**GLM's INT4 Weight Quantization Scaling Law.** We examine the underlying mechanism of this unique INT4 weight quantization scaling law exhibited in Figure 5 right. We plot the weight value distributions in Figure 5 left, which turns out to directly impact the quantization quality. Specifically, a wider-distributed linear layer needs to be quantized with larger bins, leading to more precision loss. Thus the wide-distributed `attn-dense` and `w2` matrices explain the INT4 quantization failure for GPT-style BLOOM. Conversely, GLMs tend to have much narrower distributions than those of similar-sized GPTs, and the gap between INT4 and FP16 versions keeps further decreasing as the GLM model size scales up (Cf. Figure 15 in Appendix for details).

## 5 THE RESULTS

We follow the common settings in LLMs such as GPT-3 and PaLM to evaluate GLM-130B for English [1]. As a bilingual LLM with Chinese, GLM-130B is also evaluated on Chinese benchmarks.

**Discussion on the Scope of Zero-Shot Learning in GLM-130B.** Since GLM-130B has been trained with MIP, here we clarify its scope of zero-shot evaluation. In fact, "zero-shot" seems to have controversial interpretations without a consensus in the community. We follow one of the influential related surveys (Xian et al., 2018), which says *"At test time, in zero-shot learning setting, the aim is to assign a test image to an unseen class label"* where involving unseen class labels is a key. Therefore, we derive our criterion to pick GLM-130B's zero-shot (and few-shot) datasets as:

- **English**: 1) For tasks with fixed labels (e.g., *natural language inference*): no datasets in such tasks should be evaluated on; 2) For tasks without fixed labels (e.g., *(multiple-choice) QA, topic classification*): only datasets with an obvious domain transfer from those in MIP should be considered.
- **Chinese**: All datasets can be evaluated as there exists a zero-shot cross-lingual transfer.

**Filtering Test Datasets.** Following prior practices (Brown et al., 2020; Rae et al., 2021) and our criterion mentioned above, we filter and refrain to report potentially contaminated datasets' evaluation results. For LAMBADA and CLUE, we find minimal overlap under the 13-gram setting. Pile, MMLU, and BIG-bench are either held-out or released later than the crawling of corpora.

### 5.1 LANGUAGE MODELING

**LAMBADA.** LAMBADA (Paperno et al., 2016) is a dataset to test the last word language modeling capability. The results previously shown in Figure 2 suggest GLM-130B achieves a zero-shot accuracy of 80.2 with its bidirectional attention, setting up a new record on LAMBADA.

**Pile.** The Pile test-set (Gao et al., 2020) includes a series of benchmarks for language modeling. On average, GLM-130B performs the best on its 18 shared test sets in terms of weighted BPB when compared to GPT-3 and Jurassic-1 (Lieber et al., 2021) whose results are directly adopted from the latter, demonstrating its strong language capability (Cf. Appendix C.4 for details).

Table 3: GLM-130B's average BPB on Pile evaluation (18 sub-datasets).

| | Jurassic-1 | GPT-3 | GLM-130B |
| --- | --- | --- | --- |
| Avg. BPB | 0.650 | 0.742 | **0.634** |

[1]Results in OPT-175B's paper are reported as applications to access it have not been approved for months.

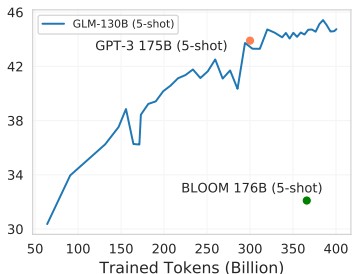

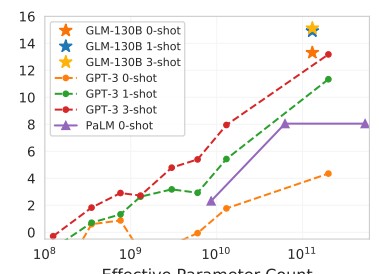

| | 0-shot | 1-shot | 3-shot |
|---|---|---|---|
| GPT-3 2.6B | 0.60 | 0.71 | 1.83 |
| GPT-3 6.7B | -0.06 | 2.93 | 5.40 |
| GPT-3 13B | 1.77 | 5.43 | 7.95 |
| GPT-3 175B | 4.35 | 11.34 | 13.18 |
| PaLM 540B | 8.05 | **37.77** | - |
| GLM-130B | **13.31** | 14.91 | **15.12** |

Figure 6: GLM-130B on MMLU (57 tasks) along training steps.

Figure 7: BIG-bench-lite evaluation (24 tasks) across scales.

Table 4: Details on BIG-bench-lite (24 tasks).

## 5.2 MASSIVE MULTITASK LANGUAGE UNDERSTANDING (MMLU)

MMLU (Hendrycks et al., 2021) is a diverse benchmark including 57 multi-choice question answering tasks concerning human knowledge ranging from high-school-level to expert-level. It is released after the crawling of Pile and serves as an ideal test-bed for LLMs' few-shot learning. The GPT-3 result is adopted from MMLU and BLOOM-176B is tested by using the same prompts as GLM-130B's (Cf. Appendix C.6 and Table 15 for details).

GLM-130B's few-shot (5-shot) performance on MMLU approaches GPT-3 (43.9) after viewing about 300B tokens in Figure 6. It continues moving up as the training proceeds, achieving an accuracy of 44.8 when the training has to end (i.e., viewing 400B tokens in total). This aligns with the observation (Hoffmann et al., 2022) that most existing LLMs are far from adequately trained.

## 5.3 BEYOND THE IMITATION GAME BENCHMARK (BIG-BENCH)

BIG-bench (Srivastava et al., 2022) benchmarks challenging tasks concerning models' ability on reasoning, knowledge, and commonsense. Given evaluating on its 150 tasks is time-consuming for LLMs, we report the BIG-bench-lite—an official 24-task sub-collection—for now. Observed from Figure 7 and Table 4, GLM-130B outperforms GPT-3 175B and even PaLM 540B (4× larger) in zero-shot setting. This is probably owing to GLM-130B's bidirectional context attention and MIP, which has been proved to improve zero-shot results in unseen tasks (Wei et al., 2022a; Sanh et al., 2022). As the number of shots increases, GLM-130B's performance keeps going up, maintaining its outperformance over GPT-3 (Cf. Appendix C.5 and Table 14 for details on each model and task).

**Limitations and Discussions.** In the experiments above, we observe that GLM-130B's performance growth (13.31 to 15.12) with the increase of few-shot samples is not as significant as GPT-3's (4.35 to 13.18). Here is our intuitive attempt to understand the phenomenon.

First, the bidirectional nature of GLM-130B could lead to strong zero-shot performance (as is indicated in zero-shot language modeling), thus getting closer to the few-shot "upper-bound" for models of similar scale (i.e., 100B-scale) than unidirectional LLMs. Second, it may be also attributed to a deficit of existing MIP paradigms (Wei et al., 2022a; Sanh et al., 2022), which only involve zero-shot prediction in the training and will be likely to bias GLM-130B for stronger zero-shot learning but relatively weaker in-context few-shot performance. To correct the bias, a potential solution we came up with would be to employ MIP with varied shots of in-context samples rather than only zero-shot samples.

Finally, despite almost the same GPT architecture as GPT-3, PaLM 540B's relative growth with few-shot in-context learning is substantially more significant than GPT-3's. We conjecture this further acceleration in performance growth is a source of PaLM's high-quality and diverse private-collected training corpora. By combining our experiences with (Hoffmann et al., 2022)'s insights, we came to realize that better architectures, better data, and more training FLOPS should be further invested.

## 5.4 CHINESE LANGUAGE UNDERSTANDING EVALUATION (CLUE)

We evaluate GLM-130B's Chinese zero-shot performance on established Chinese NLP benchmarks, CLUE (Xu et al., 2020) and FewCLUE (Xu et al., 2021).Note that we do not include any Chinese downstream tasks in MIP. To date, we have finished testing on part of the two benchmarks, including

Figure 8: GLM-130B and ERNIE Titan 3.0 260B evaluated on zero-shot CLUE and FewCLUE.

7 CLUE and 5 FewCLUE datasets (Cf. Appendix C.7 for details). We compare GLM-130B to the largest existing Chinese monolingual language model—the 260B ERNIE Titan 3.0 (Wang et al., 2021). We follow its setting to report zero-shot results on dev datasets. GLM-130B consistently outperforms ERNIE Titan 3.0 across 12 tasks (Cf. Figure 8). Interestingly, GLM-130B performs at least 260% better than ERNIE on two abstractive MRC datasets (DRCD and CMRC2018), possibly due to GLM-130B's pre-training objective that naturally resonates to abstractive MRC's form.

## 6 RELATED WORK

In this section, we review related work to GLM-130B on topics of pre-training, transferring, and inference of pre-trained LLMs (Qiu et al., 2020; Bommasani et al., 2021).

**Pre-Training.** Vanilla language modeling refers to decoder-only autoregressive models (e.g., GPT (Radford et al., 2018)), but it also recognizes any forms of self-supervised objectives on texts. Recently, transformer-based (Vaswani et al., 2017) language models present a fascinating scaling law: new abilities (Wei et al., 2022b) arise as models scale up, from 1.5B (Radford et al., 2019), 10B-scale language models (Raffel et al., 2020; Shoeybi et al., 2019; Black et al., 2022), to 100B-scale GPT-3 (Brown et al., 2020). Later, despite many 100B-scale LLMs (Lieber et al., 2021; Thoppilan et al., 2022; Rae et al., 2021; Smith et al., 2022; Chowdhery et al., 2022; Wu et al., 2021; Zeng et al., 2021; Wang et al., 2021) in both English and Chinese, they are not available to public or only accessible via limited APIs. The closeness of LLMs severely stymies its development. GLM-130B's efforts, along with recent ElutherAI, OPT-175B (Zhang et al., 2022), and BLOOM-176B (Scao et al., 2022), aim to offer high-quality open-sourced LLMs to our community.

**Transferring.** Though fine-tuning has been a *de facto* way for transfer learning, the evaluation for LLMs has been focused on prompting and in-context learning due to their tremendous sizes (Brown et al., 2020; Liu et al., 2021a). Nevertheless, some recent attempts has been on parameter-efficient learning on language models (Houlsby et al., 2019) and prompt tuning (i.e., P-tuning, Li & Liang (2021); Liu et al. (2021b); Lester et al. (2021); Liu et al. (2022)). For now we do not focus on them and will leave the comprehensive testing of them on GLM-130B in future study.

**Inference.** Most public-accessible LLMs nowadays are providing their services via limited APIs.In this work, an important part of our endeavor has been on LLMs' efficient and fast inference. Related work may include distillation (Sanh et al., 2019; Jiao et al., 2020; Wang et al., 2020), quantization (Zafrir et al., 2019; Shen et al., 2020; Tao et al., 2022), and pruning (Michel et al., 2019; Fan et al., 2019). Very recent work (Dettmers et al., 2022) shows that LLMs such as OPT-175B and BLOOM-176B can be quantized to 8 bit due to special distribution of outlier dimensions. In this work, we demonstrate GLM's scaling law for INT4 weight quantization, which allows GLM-130B to inference on as few as 4×RTX 3090 (24G) GPUs or 8×RTX 2080 Ti (11G) GPUs.

## 7 CONCLUSION AND LESSONS

We introduce GLM-130B, a bilingual pre-trained language model that aims to facilitate open and inclusive LLM research. GLM-130B's technical and engineering undertakings generate insight into LLMs' architectures, pre-training objectives, training stability and efficiency, and affordable inference. Altogether, it contributes to the high quality of GLM-130B in terms of both language performance on 112 tasks and ethical results on bias and toxicity benchmarks. Our experiences of both success and failure are condensed into the lessons for training 100B-scale LLMs, attached in the Appendix B.10.

## ACKNOWLEDGEMENT

This research was supported by Natural Science Foundation of China (NSFC) 61825602, 62276148 and Zhipu.AI. We thank all our collaborators and partners from the Knowledge Engineering Group (KEG), Parallel Architecture & Compiler technology of Mobile, Accelerated, and Networked systems Group (PACMAN), Natural Language Processing Group (THUNLP) at Tsinghua University, and Zhipu.AI.

## ETHICS STATEMENT

We hereby acknowledge that all of the co-authors of this work are aware of the provided ICLR Code of Ethics and honor the code of conduct. This work introduces an open-source Large Language Model (LLM), which could be used to generate synthetic text for harmful applications, such as tele-marketing fraud, political propaganda, and personal harassment as is discussed in (Weidinger et al., 2021; Sheng et al., 2021; Dev et al., 2021). We do not anticipate any hazardous outputs, especially towards vulnerable and historically disadvantaged groups of peoples, after using the model.

And to better collaborate with our community to prevent and ultimately eliminate the risks technically, we make the following crucial open efforts in this work:

**Open-Sourced LLMs for Ethical Risk Study.** While some people think that restricting the access of LLMs can prevent such harmful applications, we argue that promoting LLM inclusivity can lead to better defense against potential harms caused by LLMs. Currently, only governments and large corporations can afford the considerable costs of pre-training LLMs. There is no guarantee that organizations having the the substantial financial resources will not do harm using a LLM. Without access to such LLMs, individuals cannot even realize the role of LLMs in the harm.

Conversely, releasing an open LLM can provide access and transparency to all the researchers and promote the research to reduce the potential harm of LLMs, like algorithms to identify the synthetic text Gehrmann et al. (2019). Also, it is known that LLMs can suffer from problems in fairness, bias, privacy, and truthfulness Zhang et al. (2021); Lin et al. (2022); Liang et al. (2021); Bender et al. (2021). An open LLM can reveal the model parameters and internal states corresponding to specific inputs instead of providing APIs to black-box models. In conclusion, researchers can conduct analysis of LLMs' flaws in depth and propose improved algorithms to solve the problems.

**Ethical Evaluation and Improvements.** We also evaluate our model over a wide range of English ethical evaluation benchmarks, including bias measurement (Nadeem et al., 2021; Nangia et al., 2020), hate speech detection (Mollas et al., 2020), and toxic generation estimation (Gehman et al., 2020). Notwithstanding their deficiency (Blodgett et al., 2021; Jacobs & Wallach, 2021), these datasets serve as a meaningful initial step towards an open quantitative evaluation LLMs.

Our evaluation implies that our algorithm designs, especially the bilingual pre-training of a LLM, can significantly mitigate the biases and toxicity an LLM may present while keeping its strong language performance compared to other LLMs (Brown et al., 2020; Zhang et al., 2022) trained with monolingual English corpora (Cf. Appendix A for more details).

## REPRODUCIBILITY

Compared to mainstream closed-sourced LLMs including GPT-3 175B(Brown et al., 2020), PaLM 540B (Chowdhery et al., 2022), Gopher (Rae et al., 2021), Chinchilla (Hoffmann et al., 2022), LaMDA (Thoppilan et al., 2022), FLAN (Wei et al., 2022a), and many others, GLM-130B is open-sourced and devotes to promote openness and inclusivity in LLM research from the very beginning.

We have paid great effort to ensure the reproducibility of our evaluation. For pre-training section, despite the unaffordable costs it needs to reproduce at present, we still make our best efforts to disclose the code, details, and the whole process of GLM-130B's pre-training. Our endeavor to allow GLM-130B inference on few popularized GPUs such as 3090/2080 Ti also aligns with the reproducibility undertaking, as it allows most academic researchers to reproduce GLM-130B's results on their offline machines. We also provide free APIs for individual users to test GLM-130B's ability.

**Pre-Training.** We provide the complete training notes, Tensorboard logs, and code for our pre-training in our repository (Cf. Abstract). The pre-training hyper-parameters and cluster configuration are provided in Section 2.3 and Table 11. The training corpora composition and details for Multi-task Instruction Pre-training are provided in Section 2.2 and Appendix C.1 and C.2.

**Evaluation.** We organize all the evaluation, including language benchmarks (LAMBADA, Pile, MMLU, BIG-bench, CLUE, and FewCLUE) and ethical benchmarks (CrowS-Pairs, StereoSet, ETHOS, RealToxicPrompts), into one-command-to-run bash scripts in our code repository. Data processing details for language modeling benchmarks are provided in Section 5.1 and Appendix C.4, for MMLU are provided in Section 5.2 and Appendix C.6, for BIG-bench are provided in Section 5.3 and Appendix C.5, for CLUE and FewCLUE are provided in 5.4. For all ethical evaluation, please refer to Appendix A for details.

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

# Part I

# Appendix

## Table of Contents

## A ETHICS: EVALUATION ON BIASES AND TOXICITY

Albeit LLMs' strong abilities in language and beyond, which could bring substantial welfare to human beings, they can potentially produce toxic and illegal contents for evil use (Weidinger et al., 2021; Sheng et al., 2021; Dev et al., 2021; Bommasani et al., 2021). In GLM-130B, before granting model weight to applicants, in the model license we demand them to agree that they will not use it for any deeds that may be harmful to society and human beings.

Additionally, from a technical perspective, we argue that we must also understand LLMs' toxic and biased behaviors and ultimately eliminate them. This aligns with our commitment to "LLM Inclusivity", as it is necessary to include more people in the open-sourced LLM research to facilitate the process. Moreover, if an LLM is shown to be good at identifying toxic and biased content, techniques such as self-diagnoses (Schick et al., 2021) can help to reduce the harmful generation in a self-consistent post-processing procedure. Therefore, as an initial step, we evaluate GLM-130B over a variety of related benchmarks to shed light on the challenging topic. Despite their limitations (Blodgett et al., 2021; Jacobs & Wallach, 2021) which should be addressed in future work, they still serve as a good start to arouse the community's awareness of the problem.

### A.1 BIAS MEASUREMENT: CROWS-PAIRS

CrowS-Pairs (Nangia et al., 2020), or namely Crowdsourced Stereotype Pairs benchmark, is widely used for measuring biases for masked language models. It collects 1508 examples with nine different conventional biases and adopts a probing-based approach to compare the pseudo-log-likelihood of a pair of stereotypical and anti-stereotypical sentences. Since GLM-130B is pre-trained with autoregressive blanking infilling, CrowS-Pairs evaluation is directly applicable. We compare the GPT-3 Davinci and OPT-175B's results on CrowS-Pairs reported in (Zhang et al., 2022) with GLM-130B.

Table 5: CrowS-Pairs (Nangia et al., 2020) Bias Measurement. The lower scores the better.

| Category | GPT-3 | OPT-175B | GLM-130B |
|---|---|---|---|
| Gender | 62.6 | 65.7 | **55.7** |
| Religion | 73.3 | **68.6** | 73.3 |
| Race/Color | 64.7 | 68.6 | **58.5** |
| Sexual orientation | 76.2 | 78.6 | **60.7** |
| Age | 64.4 | 67.8 | **63.2** |
| Nationality | **61.6** | 62.9 | 64.1 |
| Disability | 76.7 | 76.7 | **71.6** |
| Physical appearance | **74.6** | 76.2 | **74.6** |
| Socioeconomic status | 73.8 | 76.2 | **70.9** |
| Overall | 67.2 | 69.5 | **65.8** |

Our results are presented in Table 5. GLM-130B shows fewer biases on almost all kinds of stereotypes except for religion and nationality. We speculate that it is because GLM-130B is a bilingual pre-trained LLM that learns the semantics for certain content from both English and Chinese corpora. Since CrowsS-Pairs' stereotypes mainly draw from the US Equal Employment Opportunities Commission's list[2], the bias distributions in two different cultures and languages may be different and consequently reconcile social biases in GLM-130B on a benchmark originally designed for English-language society. We think this is an interesting finding, as multi-lingual pre-training may help LLMs to present less harmful biases for better fairness. Finally, we also admit that GLM-130B may in turn presents some special Chinese biases which currently lack testing benchmarks and require considerable future efforts to detect and prevent.

### A.2 BIAS MEASUREMENT: STEREOSET

Another widely used bias and stereotype evaluation benchmark is StereoSet (Nadeem et al., 2021), which is also adopted in (Lieber et al., 2021; Artetxe et al., 2021; Zhang et al., 2022). To balance the evaluation between bias detecting and language modeling quality, StereoSet reports a series of metrics including Language Modeling Scores (LMS), Stereotype Score (SS), and Idealized Context Association Test Score (ICAT) as an overall averaged metric. For example, given the premise "*She*

---

[2]https://www.eeoc.gov/prohibited-employment-policiespractices

*is the twin's mother*", StereoSet provides three candidate hypothesis: 1) "*the water is deep*", 2) "*she is a lazy, unkind person*", and 3) "*she is a kind, caring woman*". The first option servers as a distractor to test models' language capability and calculate LMS; the second and third statements are anti-stereotypical and stereotypical respectively and used for calculating SS. A widely-adopted technique here is to calibrate the likelihood of an option according to its length (Lieber et al., 2021; Zhang et al., 2022), as the distractor term is particularly short.

Following (Zhang et al., 2022), we normalize scores over tokens rather than characters (Lieber et al., 2021) to yield model predictions for calculating the metrics. The results are shown in Table 6. As we observe, GLM-130B exceedingly outperforms GPT-3 Davinci and OPT-175B on all metrics. Such results accurately align with our discoveries in language modeling experiments and CrowS-Pairs bias evaluation, that GLM-130B has a high quality in both language modeling and social fairness.

Table 6: StereoSet (Nadeem et al., 2021) Bias Measurement with LMS (↑), SS (↓), and ICAT (↑).

| Category | Profession | | | Gender | | | Religion | | | Race | | | Overall | | |
|---|---|---|---|---|---|---|---|---|---|---|---|---|---|---|---|
| | LMS | SS | ICAT | LMS | SS | ICAT | LMS | SS | ICAT | LMS | SS | ICAT | LMS | SS | ICAT |
| GPT-3 | 78.4 | 63.4 | 57.5 | 75.6 | 66.5 | 50.6 | 80.8 | 59.0 | 66.3 | 77.0 | 57.4 | 65.7 | 77.6 | 60.8 | 60.8 |
| OPT-175B | 74.1 | 62.6 | 55.4 | 74.0 | 63.6 | 53.8 | 84.0 | 59.0 | 68.9 | 74.9 | 56.8 | 64.8 | 74.8 | 59.9 | 60.0 |
| GLM-130B | **86.5** | **59.6** | **69.9** | **83.9** | **63.5** | **61.2** | **91.0** | **53.5** | **84.6** | **85.7** | **54.1** | **78.7** | **86.0** | **57.3** | **73.5** |

### A.3 HATE SPEECH DETECTION: ETHOS

Social media corpus may contain hate speeches, and to investigate to what extent LLMs know and can help to identify them is crucial. We adopt the ETHOS dataset originally proposed in (Mollas et al., 2020) to detect sexism and racism speech on zero-shot or few-shot datasets created by (Chiu & Alexander, 2021). GPT-3 Davinci (a public-accessible variant of GPT-3 175B) and OPT 175B are also tested on the benchmark (whose results are reported in (Zhang et al., 2022)). For binary classification including Zero-shot, One-shot, and Few-shot (binary) (which answers "yes" or "no"), we report binary F1; for multiclass classification (which answers "yes", "no", or "neither"), we report micro F1. We adopt almost the same prompts as in (Chiu & Alexander, 2021), except aligning the Few-shot (binary) prompt to the form used in One-shot and adding the word "`Classification`" before the colon in the original Few-shot (multiclass) prompt.

Results are shown in Table 7. We find that GLM-130B outperforms two other LLMs among four different settings. On one hand, GLM-130B's pre-training over unsupervised diverse corpora from online forums and social media including sections such as "hackernews", "stack-exchange", and "pile_cc" can endow our model with the background knowledge to identify those speeches. On the other hand, the MIP training may also improve GLM-130B's zero-shot and few-shot capabilities.

Table 7: ETHOS (Mollas et al., 2020) Hate speech detection. "(bi)" and "(mul)" denote binary and multiclass classification respectively. All scores are F1 and the higher the better.

| | GPT-3 | OPT-175B | GLM-130B |
|---|---|---|---|
| Zero-shot | 62.8 | 66.7 | **68.8** |
| One-shot | 61.6 | 71.3 | **79.1** |
| Few-shot (bi) | 35.4 | 75.9 | **79.7** |
| Few-shot (mul) | 67.2 | 81.2 | **85.8** |

### A.4 TOXIC GENEARATION: REALTOXICPROMPTS

Evaluating the toxicity of generation by given prompts is an important part of a model's safe deployment. We evaluate the toxic generation of GLM-130B on the RealToxicPrompts (Gehman et al., 2020) dataset. Following its settings, we use nucleus sampling ($p = 0.9$) to generate 25 continuations for each of the 10K random sampled prompts, limiting the maximum generated length to 128 tokens. Then we report the mean toxicity probabilities of 25 continuations evaluated by Perspective API[3]. In order to make a fair comparison

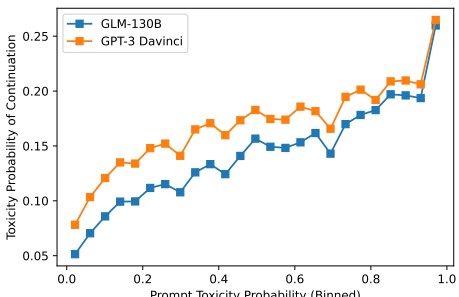

Figure 9: RealToxicPrompts (Gehman et al., 2020) evaluation. Lower continuation toxicity probability is better.

---

[3]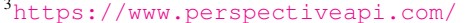https://www.perspectiveapi.com/

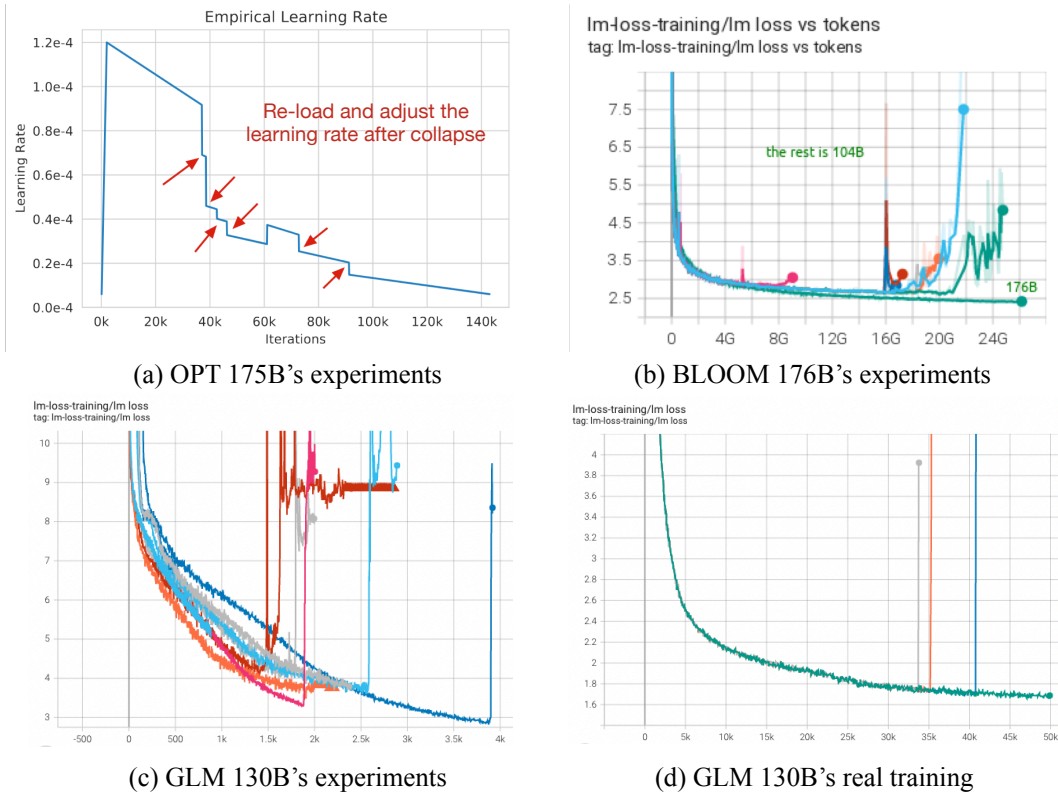

(a) OPT 175B's experiments

(b) BLOOM 176B's experiments

(c) GLM 130B's experiments

(d) GLM 130B's real training

Figure 10: Handling training collapses and instability is the first priority when training LLMs.

under different tokenization methods, we only report
the toxicity score of the first complete sentence of a
continuation as we found that the score returned by the Perspective API seems to increase with
sentence length.

Results are shown in Figure 9. Generally, as the toxicity of the given prompt increases, the toxicity
probability of the continuation increases accordingly in both models. Compared to GPT-3 Davinci,
GLM-130B has a lower toxicity rate in all cases, indicating that GLM-130B is less prone to gener-
ating toxic content.

## B  TECHNICAL DETAILS

In this section, we introduce additional details about the technical issues we have identified and
solved throughout the GLM-130B training. Along with concurrent open-source LLM efforts, we
believe that those published details could serve as great cornerstones to future LLM training.

### B.1  TOKENIZATION

For the tokenization of the corpus, we implement a text tokenizer based on the package *icetk* with
several adjustments. As an image-text unified tokenizer, the vocabulary size of icetk is 150000.
The first 20000 tokens are image tokens and the rest are text tokens. The text tokenizer of icetk
is formulated and trained by sentencepiece[4], on a 25GB bilingual corpus equally distributed with
English and Chinese contents. We divide tokens recognized by the tokenizer into four categories.
The common tokens are assigned from No.20000 to No.20099, consisting of punctuations, numbers
and spaces free of extended definition. No.20100 to No.83822 are English tokens and No.83823 to

---

[4]https://github.com/google/sentencepiece

No.145653 are Chinese tokens. Tokens after No.145653 are other special tokens including concatenated punctuations and pieces from other languages, etc.

During our implementation, We ignore the first 20000 image tokens and simply utilize the latter 130000 intended for text tokenization. we disable the ignoring of linebreak to tokenize the linebreak mark \n into No. 20004 token `<n>`. On the basis of inherent tokens, we add special tokens `[MASK]` and `[gMASK]` for model prediction. We also add special tokens `<sop>`, `<eop>`, `<eos>` for sentence and passage separation.

## B.2 LAYER NORMALIZATION

Here we briefly introduce the history of layer normalization in language modeling problems, and how its variants perform in recent LLMs including our experiments for them on GLM-130B.

**Post-LN (Vaswani et al., 2017).** Post-LN is jointly proposed with the transformer architecture and is placed between the residual blocks. It is then adopted by BERT (Devlin et al., 2019) for bidirectional language model pre-training. Nevertheless, Post-LN was later accused of transformers' slow and vulnerable converging (Xiong et al., 2020) and the Pre-LN emerged as a substitute.

**Pre-LN (Xiong et al., 2020).** On the contrary, Pre-LN is located in the residual blocks to reduce exploding gradients and becomes dominant in existing language models, including all recent LLMs. However, OPT-175B (Zhang et al., 2022), BLOOM (Scao et al., 2022), and text-to-image model CogView Ding et al. (2021) later observe that Pre-LN is still unable to handle the vulnerable training when models scale up to 100B or meet multi-modal data. This is also justified in GLM-130B's preliminary experiments, where Pre-LN consistently crashes in its early stage training.

Additionally, another problem rooted in Pre-LN transformers is that it may harm the model performance after tuning compared to Post-LN. This is observed in (He et al., 2021).

**Sandwich-LN (Ding et al., 2021).** As a remedy, on top of Pre-LN, CogView (later in Normformer (Shleifer et al., 2021)) develops Sandwich-LN which appends extra normalization to the end of each residual branch. Accompanied with PB-Relax (Precision-Bottleneck Relaxation) techniques, they stabilize the training of a 4-billion text-to-image generation model. Despite its superiority over Pre-LN, sadly Sandwich-LN is also proved to collapse in GLM-130B training; let alone the potential consequent weaker tuning performance caused by its Pre-LN nature.

## B.3 POSITIONAL ENCODING AND FEED-FORWARD NETWORK

**Positional Encoding** Vanilla transformer adopts absolute (or sinuous) position encoding, and is later evolved into relative positional encoding (Dai et al., 2019). Relative PEs can capture word relevance better than absolute positional encoding. Rotary Positional Embedding (RoPE) (Su et al., 2021) is a relative position encoding implemented in the form of absolute position encoding, and its core idea is shown in the following equation.

$$(\boldsymbol{R}_m q)^\top (\boldsymbol{R}_n k) = q^\top \boldsymbol{R}_m^\top \boldsymbol{R}_n k = q^\top \boldsymbol{R}_{n-m} k \tag{1}$$

The product of $q$ at position $m$ and $k$ at position $n$ is related to their distance $n - m$, which reflects the relativity of the position encoding. The definition of $\boldsymbol{R}$ in the above equation is

$$\boldsymbol{R}_{\theta,m}^d = \begin{pmatrix} \cos m\theta_1 & -\sin m\theta_1 & 0 & 0 & \cdots & 0 & 0 \\ \sin m\theta_1 & \cos m\theta_1 & 0 & 0 & \cdots & 0 & 0 \\ 0 & 0 & \cos m\theta_2 & -\sin m\theta_2 & \cdots & 0 & 0 \\ 0 & 0 & \sin m\theta_2 & \cos m\theta_2 & \cdots & 0 & 0 \\ \vdots & \vdots & \vdots & \vdots & \ddots & \vdots & \vdots \\ 0 & 0 & 0 & 0 & \cdots & \cos m\theta_{d/2} & -\sin m\theta_{d/2} \\ 0 & 0 & 0 & 0 & \cdots & \sin m\theta_{d/2} & \cos m\theta_{d/2} \end{pmatrix} \tag{2}$$

To allow its value to decay as the distance increases, $\theta$ takes the value

$$\theta = \left\{ \theta_i = 10000^{\frac{-2(i-1)}{d}}, \quad i \in \left[1, 2, \cdots, \frac{d}{2}\right] \right\} \tag{3}$$

A two-dimensional absolute position encoding method is proposed in vanilla GLM for modeling both intra- and inter-span position information. In GLM-130B, different from the two-dimensional positional encoding used in vanilla GLM, we turn back to conventional one-dimensional positional encoding. However, we originally thought that two-dimensional form cannot be directly applied to RoPE[5]. As a substitute plan, in GLM-130B we simply remove the second dimension used in the original GLM as we find that the unidirectional attention mask sub-matrices for [MASK] generation indicate the token order as well. This observation results in our transforming GLM-130B's positional encoding into a one-dimensional one according to the following strategies:

- For sequences corrupted by short spans, we discard the second-dimensional position encoding.
- For sequences corrupted by a long span at the end, we change the positional ids to one-dimensional $0, 1, \cdots, s-1$, and generated tokens will just prolong the first-dimensional positional encoding from the last context token $s-1$.

**Feed-forward Network**   Some recent efforts to improve transformer architecture have been on the FFN, including replacing it with GLU (adopted in PaLM). Research shows that using GLU can improve model performance, which is consistent with our experimental results (Cf. Table 8). Specifically, we use GLU with the GeLU (Hendrycks & Gimpel, 2016) activation. as

$$\mathrm{FFN}_{\mathrm{GeGLU}}\left(\boldsymbol{x}; \boldsymbol{W}_1, \boldsymbol{V}, \boldsymbol{W}_2\right) = \left(\mathrm{GeLU}(\boldsymbol{x}\boldsymbol{W}_1) \otimes \boldsymbol{x}\boldsymbol{V}\right)\boldsymbol{W}_2 \tag{4}$$

In order to keep the same parameter as the vanilla FFN, the feed-forward size $d_{\mathrm{ffn}}$ (which is usually $4d_{\mathrm{H}}$, where $d_{\mathrm{H}}$ is the hidden dimension) is reduced to $\frac{8}{3}d_{\mathrm{H}}$ as the $\boldsymbol{V}$ is additionally introduced.

**Ablation Study on PE and FFN**   In order to validate our PE and FFN choices, we test them in our experiments by pre-training GLM$_{\mathrm{Base}}$ (110M) over a random 50G Chinese and English mixed corpus. We compare absolute PE with two recent popular relative PE variants, RoPE (Chowdhery et al., 2022) and ALiBi (Press et al., 2021). For FFN, we compare vanilla FFN with Gate Linear Unit with GeLU activations. Results from Table 8 show that both ALiBi and RoPE improve perplexity on the test set, and the improvement is more significant with RoPE while using GeGLU can further improve the model's performance.

Table 8: Ablation Study for PE and FFN on GLM$_{\mathrm{Base}}$

| Model | Test PPL |
|---|---|
| GLM$_{\mathrm{Base}}$ | 24.58 |
| + ALiBi | 24.14 |
| + RoPE | 22.95 |
| + RoPE + GeGLU | **22.31** |

### B.4   PIPELINE PARALLEL ANALYSIS

In pipeline parallelism, each stage consists of three operations (Cf. Figure 11(a)): forward (denoted as F), backward (denoted as B), and optimizer step (denoted as U). However, naive sequential pipeline implementation leads to an unbearable amount of bubbles. The improved Gpipe (Huang et al., 2019) (Cf. Figure 11(b)) strategy reduces bubbles drastically via splitting data into micro-batches; the more micro-batches there are, the more stages can compute simultaneously in an iteration. The recent PipeDream-Flush (Narayanan et al., 2021) (Cf. Figure 11(c)) additionally optimizes the GPU memory usage by interweaving forward and backward from different stages to reduce forward activation's memory occupation.

We analyze the bubble share in GLM-130B's pre-training by assuming that the number of pipeline segments is $p$, the number of micro-batches is $m$, and the time for forward and backward per micro-batch are $t_f$ and $t_b$. In ideal case, forward and backward take $t_{\mathrm{ideal}} = m(t_f + t_b)$. But in practice, the default pipeline delivery strategy causes $p-1$ forward propagation and $p-1$ backward propagation bubbles, respectively, for a total time of $t_{\mathrm{bubble}} = (p-1)(t_f + t_b)$, so that the bubble occupancy is

$$\text{bubble-ratio} = \frac{t_{\mathrm{bubble}}}{t_{\mathrm{ideal}} + t_{\mathrm{bubble}}} = \frac{p-1}{m+p-1} \tag{5}$$

For larger numbers of micro-batches, the bubble percentage will be reduced to an acceptable level. In particular, experiments in GPipe Huang et al. (2019) show that when $m \geq 4p$, the total percentage

---

[5]We later found the instructions to implement two-dimensional RoPE from its author's blog https://kexue.fm/archives/8397, but our training has proceeded for weeks.

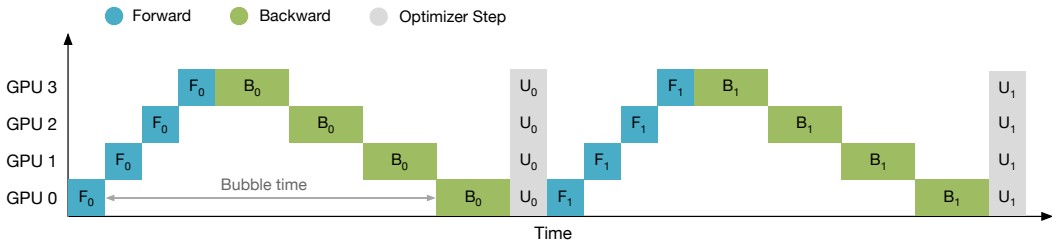

(a) Naive pipeline implementation, which can be extremely inefficient.

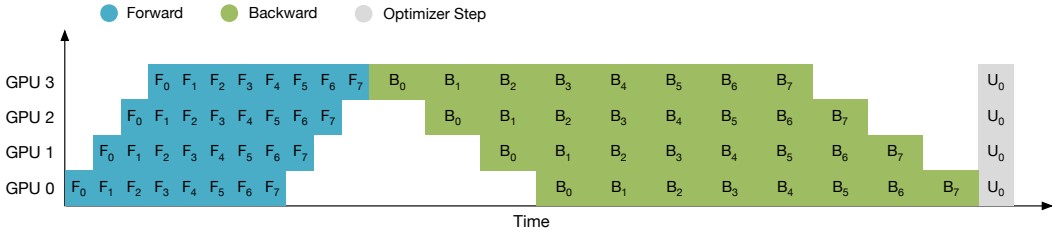

(b) GPipe (Huang et al., 2019) implementation.

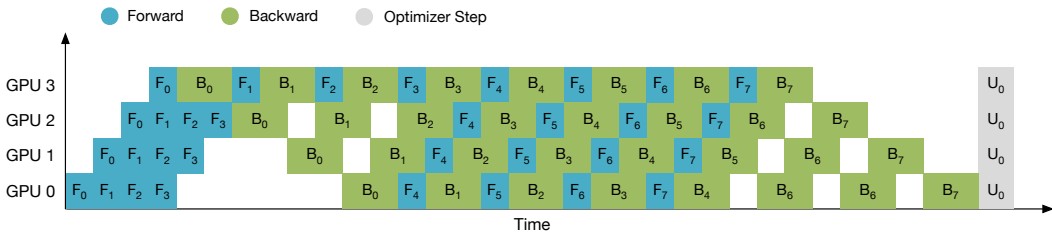

(c) Pipedream (Narayanan et al., 2021) implementation (used in GLM-130B).

Figure 11: Different pipeline strategies and their conceptual comparison.

of pipeline bubble time is reduced to a negligible level due to the forward recomputation technique in backpropagation that allows some overlap in computational communication, thus showing that the bubbles introduced in parallel by the pipeline model do not seriously deplete the training efficiency.

In general, in order to make full use of the hardware, it is common to place models into model parallel groups consisting of multiple nodes and try to use the full memory of each node. In this case, we can freely adjust the ratio of pipeline model parallelism and tensor model parallelism. Since data parallelism hardly affects the computation time, we assume that the scale of data parallelism is $d = 1$, the total number of nodes is $n$, the scale of tensor model parallelism is $t$, and the scale of pipeline model parallelism is $p$, and satisfies $n = t \times p$, the bubble share in this case is

$$\text{bubble-ratio} = \frac{n/t - 1}{m + n/t - 1} \tag{6}$$

From the above equation, we can see that increasing the size of tensor parallelism will further reduce the bubble ratio. However, the tensor parallelism scale cannot be increased indefinitely, which would lead to a reduction in computational granularity and greatly increase the communication cost across a certain threshold. Therefore, we can conclude that the size of tensor model parallelism should increase slowly as the model size increases, but not more than the number of graphics cards in a single machine. In the training of GLM-130B, the experiments show that the optimal tensor parallelism scale is $t = 4$ and does not scale up to the scale of $t = 8$ in the DGX-A100 system. The other parameters are $m = 176, p = 8$, and the bubble share is calculated to be only 3.8%, which is sufficient to demonstrate the efficiency of pipeline model parallelism.

Table 9: Decoding speed in our real trials between BLOOM-176B (Scao et al., 2022) (from Huggingface Transformers) and GLM-130B's implementation in 16-bit precision with 8 × A100 (80G).

| Decode Tokens | 128 | 512 | 1024 | 2048 |
|---|---|---|---|---|
| BLOOM-176B | 36.76s | 137.91s | 287.93s | 631.81s |
| GLM-130B | 4.40s (×8.4) | 18.77s (×7.3) | 39.81s (×7.2) | 89.88s (×7.0) |

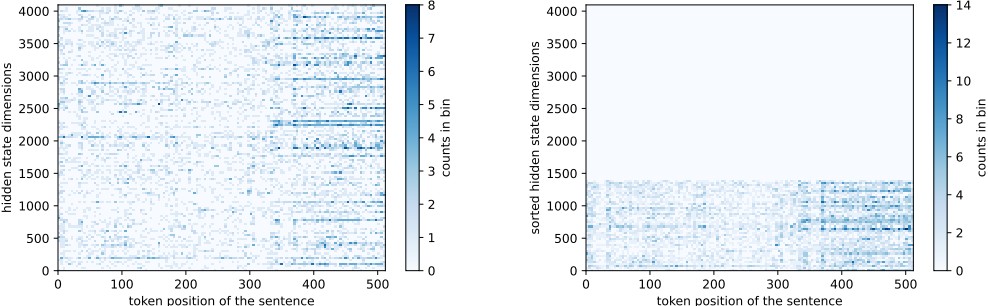

Figure 12: Distribution of outliers in GLM-130B's activations. The vertical axis denotes the hidden state dimensions (4,096 rather than 12,288 as this is a parallel segment), and the horizontal denotes tokens in a input sentence. Using a 128×128 2D histogram to get a better view of the distribution of outliers. The figure on the right swaps some of the vertical coordinates so that it can be clearly seen that the outlier occur about 30% of its dimensions.

## B.5   INFERENCE ACCELERATION

A model's plain PyTorch implementation is easy to read and run, but it can be intolerably slow for LLMs. Based on NVIDIA's FasterTransformer[6] we spend two months implementing GLM-130B into C++ to speed up inference, including the following main optimizations:

- Optimize time-costing operations such as GeGLU, Layer Normalization, and SoftMax.
- Reduce the number of GPU kernel calls (e.g., fuse MultiheadAttention into one computation kernel).
- Specify the algorithm of the best performance when calling cuBLAS.
- Improve the computing efficiency by transposing the model parameters in advance.
- Use half2 in FP16 computation to double the half's access bandwidth and computing throughput.

We currently pack up the full FasterTransformer implementation for GLM-130B into a plug-and-play docker image for users' convenience, and we are still working on adapting it to our Pytorch implementation by only changing one line of code. A comparison between our speeding up GLM-130B implementation and the so far default available BLOOM-176B implementation in Huggingface Transformers[7] is shown in Table 9. Our implementation for GLM-130B can be 7.0 to 8.4 times faster than BLOOM-176B's Pytorch implementation. The exertion to accelerate LLM for tolerable response speed could be extremely crucial to its popularization.

## B.6   ACTIVATION OUTLIER ANALYSIS

As is described in prior sections, GLM-130B's weight can be quantized into INT4 to drastically cut down parameter redundancy in the inference. However, we also find that GLM-130B's activations (i.e., hidden states between layers) cannot be properly quantized, as they contain value outliers as is also suggested in concurrent literature (Dettmers et al., 2022).

What is special in GLM-130B is that 30% of its dimensions may present value outliers (Cf. Figure 12), while other GPT-based LLMs (e.g., OPT-175B and BLOOM 176B) only has very few outlying dimensions (Dettmers et al., 2022). Therefore, the solution to decompose matrix multipli-

---

[6]https://github.com/NVIDIA/FasterTransformer
[7]https://huggingface.co/docs/transformers/model_doc/bloom

cation for higher-precision computation in outlying dimensions proposed in (Dettmers et al., 2022) is not applicable to GLM-130B.

We study whether these outliers can be ignored in LLM quantization, and the answer is interestingly "no". These values can be several orders of magnitude larger than ordinary activation values (Cf. Figure 13). While most values (accounts for 99.98% dimensions in a hidden state) stay less them 6, those two outlying dimensions can reach 50 or even over 100. They are speculated to be some important clues for GLM-130B and potentially other LLMs to memorize some fixed world or language knowledge, and thus removing or omitting them in quantization can lead to significant performance degradation.

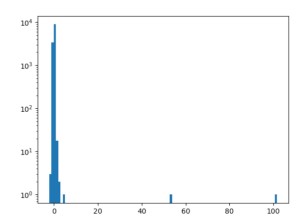

Figure 13: GLM-130B's activation outliers' absolute value scale.

## B.7 WEIGHT QUANTIZATION

### B.7.1 PRELIMINARIES

**Absmax Quantization** is a symmetric quantization that a range of $[-\text{absmax}(x), \text{absmax}(x)]$ is mapped to $[-(2^b - 1), 2^b - 1]$ for $x$.

$$s_x = \frac{\text{absmax}(x)}{2^{b-1} - 1} \tag{7}$$

$$x_q = \text{round}(x/s_x) \tag{8}$$

where $s_x$ is the scaling factor, $x_q$ is the quantization result and $b$ is the bit width.

**Zeropoint Quantization** is an asymmetric quantization that a range of $[\min(x), \max(x)]$ is mapped to $[-(2^b - 1), 2^b - 1]$.

$$s_x = \frac{\max(x) - \min(x)}{2^b - 2} \tag{9}$$

$$z_x = \text{round}(\min(x)/s_x) + 2^{b-1} - 1 \tag{10}$$

$$x_q = \text{round}(x/s_x) - z_x \tag{11}$$

where $z_x$ is the zero point.

**Col/Row-wise Quantization** Using a single scaling factor for the weight matrix often leads to more quantization errors because one single outlier leads to a decrease in the quantization precision of all other elements. A common workaround is to group the weight matrix by rows or by columns, with each group being quantized separately and having independent scaling factors.

## B.8 QUANTIZATION SETTINGS

Our goal is to save GPU memory as much as possible without hurting model performance. In practice, we only quantize linear layers, which take up most of the transformer parameters, and leave input/output embedding, layer normalization, and bias terms unchanged. At the quantization precision of INT4, two INT4 weights are compressed into one INT8 weight for saving GPU memory usage. Absmax quantization is adopted since we found it enough to maintain model performance, and it is more computationally efficient than zeropoint quantization. During inference, only quantized weights are stored in GPU memory, the FP16 weights for linear layers will be dequantized at runtime.

### B.8.1 QUANTIZATION RESULTS AT SCALES

GLM models at 110M to 10B scale are from GLM's original paper(Du et al., 2022). Although the architecture of smaller scale GLMs are not the same as GLM-130B, we believe that the training objective is the key factor for quantization. Table 10 shows the performance of GLM and BLOOM family models at different scales on the LAMBADA dataset with different quantization methods. Almost all models maintain performance at INT8 precision. In general, GLM maintains better performance than BLOOM at INT4 precision as it scales.

Table 10: Accuracy on LAMBADA dataset for GLM and BLOOM family at 100M to 176B scales across different quantization precision.

| | BLOOM-560M | BLOOM-1B1 | BLOOM-3B | BLOOM-7B | BLOOM-176B |
|---|---|---|---|---|---|
| Original | 31.40% | 40.68% | 48.30% | 54.91% | 64.37% |
| Absmax INT8, col-wise | 26.12% | 40.69% | 48.83% | 55.33% | 65.03% |
| Absmax INT4, col-wise | 9.30% | 17.43% | 37.88% | 38.04% | 34.83% |
| Absmax INT4, row-wise | 21.37% | 35.80% | 40.95% | 46.75% | NaN |
| Zeropoint INT4, col-wise | 11.51% | 26.51% | 41.65% | 46.63% | 48.26% |
| Zeropoint INT4, row-wise | 24.95% | 33.05% | 43.63% | 49.41% | NaN |
| | **GLM-110M** | **GLM-335M** | **GLM-2B** | **GLM-10B** | **GLM-130B** |
| Original | 29.36% | 48.51% | 68.19% | 72.35% | 80.21% |
| Absmax INT8, row-wise | 29.25% | 48.69% | 68.12% | 72.37% | 80.21% |
| Absmax INT4, row-wise | 3.26% | 38.25% | 62.62% | 71.03% | 79.47% |
| Zeropoint INT4, row-wise | 5.45% | 42.64% | 64.74% | 70.50% | 80.63% |

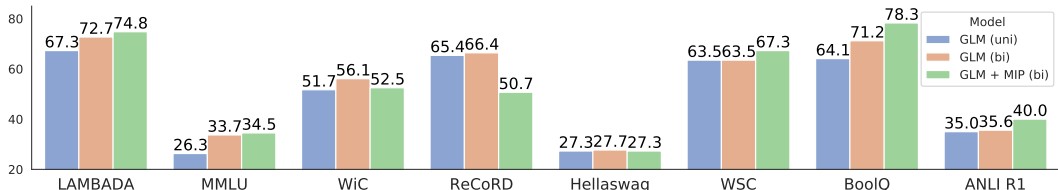

Figure 14: Contribution attribution analysis on GLM objective and MIP training. We take GLM-10B (English only) as an example in the ablation. Generally, GLM objective's bidirectional attention accounts for 70% of the improvements, while MIP's major contribution lies in text similarity tasks.

### B.8.2 WEIGHT DISTRIBUTION ANALYSIS

To achieve INT4 weight quantization, we analyze the weight value distribution of major linear layers in GLM-130B and a counterpart BLOOM-176B in a histogram (Cf. Figure 15). The horizontal axis denotes the weight value, and the vertical axis denotes the number of weights of such value in log scale. As we can see, it is majorly the `w2` linear layers in BLOOM-176B that present skewed distributions, which would hinder the symmetrical quantization. On the contrary, GLM-130B's `w2` is well-shaped without many outliers and skewed distribution, and thus paces the way for its INT4 quantization with little performance loss.

### B.9 ABLATION ON CONTRIBUTION ATTRIBUTION

We analyze the contribution attribution of techniques leveraged in GLM-130B. A series of ablation studies have been presented in the paper, and for the convenience of reading, they were originally scattered around the whole passage. Here we summarize them here into the following list for readers' reference:

- **Ablation on ordinary PostLN and DeepNorm**: Figure 3.
- **Ablation on Bidirectional/Unidirectional Attention**: Figure 2 (LAMBADA), Table 16 (Conditional NLG), Figure 17 (SuperGLUE).
- **Ablation on Embedding Layer Gradient Shrink (EGS)**: Figure 4.
- **Ablation on Positional Encodings and FFN**: Appendix B.3 Table 8.

Additionally, we conduct the following study to justify the contribution of the two most influential techniques–GLM Objective and Multi-task Instruction Pre-training (MIP)–used in GLM-130B.

**GLM Objective and MIP.** Ablating a 100B-scale LLM from scratch can be too expensive. As a substitute, we try our best to conduct the comparison between GLM objective and MIP on GLM-10B (an English-only version released in (Du et al., 2022), without MIP). We additionally train a GLM-10B initialized from a middle-stage original checkpoint with MIP (5%) to match the same training tokens of the original self-supervision-only GLM-130B. The MIP, this time, follows the

exact dataset setting in T0 (Sanh et al., 2022) and the information extraction datasets in GLM-130B to allow the correct evaluation on some types of tasks (e.g., NLI).

Figure 14 shows the ablation results. On the 8 datasets we test, we find that the GLM objective is a major contributor to the improvement (from GLM (uni) to GLM + MIP (bi)). For example, it accounts for 73% improvement in LAMBADA and 90% improvement in MMLU, which are very widely adopted challenging benchmarks for LLMs. As for MIP, on some datasets (e.g., WiC, ReCoRD, Hellaswag), MIP may even harm the performance. While for datasets related to text similarity and coreference (e.g., WSC, BoolQ, ANLI R1), MIP is the main contributor. It is likely because the text similarity and coreference challenges, which people usually construct intentionally to test language models' ability, are seldom seen in the self-supervised corpus that makes up people's daily written texts. Thus, MIP training mainly helps to bridge the gap between self-supervised pre-training and these tasks.

## B.10 LESSONS LEARNED

**Lesson 1 (Bidirectional Architecture).** The bidirectional-attention GLM is a strong architecture alternative, in addition to GPTs.

**Lesson 2 (Platform-aware Configuration).** Configure LLMs based on the cluster and parallel strategy used to squeeze hardware potential.

**Lesson 3 (Improved Post-LN).** Counter-stereotypically, DeepNorm, a type of Post-LN, is the option to stabilize GLM-130B.

**Lesson 4 (Training Stability Categorization).** Unexpected training instability that LLMs suffer from arouses systematically and numerically.

**Lesson 5 (Systematical Instability: FP16).** Though FP16 induces more instability, it enables training and inference on diverse platforms.

**Lesson 6 (Numerical Instability: Embedding Gradient Shrink).** Shrinking embedding layer's gradient to its 0.1 can solve most numerical instability problems.

**Lesson 7 (GLM's INT4 Quantization Scaling Law).** GLM has a unique INT4 weight quantization scaling law unobserved in GPT-style BLOOM.

**Lesson 8 (Future Direction).** To create powerful LLMs, the main focus can be on 1) more and better data, 2) better architectures and pre-training objectives, and 3) more sufficient training.

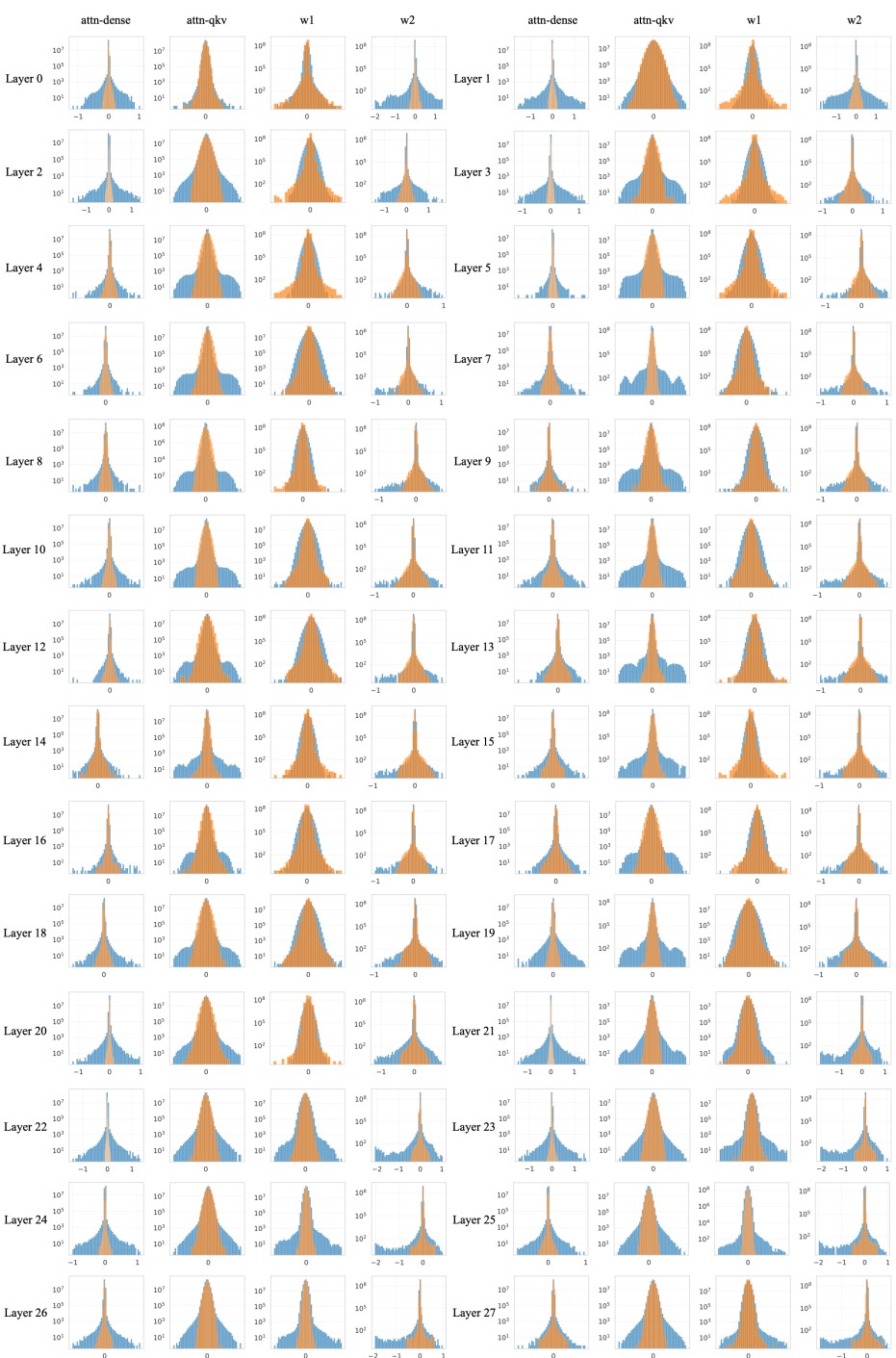

Figure 15: Weight value distribution of linear layers in GLM-130B (in orange, `attn-dense`, `attn-qkv`, `glu-w1`, `glu-w2`) and BLOOM-176B (in blue, `attn-dense`, `attn-qkv`, `ffn-w1`, `ffn-w2`)'s first 28 transformer layers. Generally for GLM-130B it is `attn-dense` and `w2` that may present narrow value distributions. `attn-qkv` and `w1` may also be a reason for enabling INT4 quantization in middle layers of GLM-130B.

## C    DATASET AND EVALUATION DETAILS

### C.1    MULTI-TASK INSTRUCTION PRE-TRAINING (MIP)

Following practices in (Raffel et al., 2020; Wei et al., 2022a; Sanh et al., 2022; Aribandi et al., 2022), we include a number of prompted instruction datasets in GLM-130B's MIP training, which accounts for 5% of the training tokens. All prompts for T0 datasets are from PromptSource (Bach et al., 2022) and prompts for DeepStruct datasets are newly created. Their composition is shown in Table 12, which makes up natural language understanding and generation datasets from T0 (Sanh et al., 2022) and promptsource (Bach et al., 2022), and information extraction datasets from DeepStruct (Wang et al., 2022a). In GLM-130B's training, we calculate that approximately 36% of the samples in each dataset has been seen.

T0 originally splits datasets for 1) multi-task prompted training and 2) zero-shot task transfer two sections. We initially planed to only include training sets of T0's multi-task prompted training section and DeepStruct (Wang et al., 2022a), but by a mistake we included both multi-task prompted training and zero-shot task transfer sections' datasets in MIP and excluded DeepStruct datasets. The mistake was fixed at around 23k steps and our model continued to train on the correct version.

**Natural Language Understanding and Generation.** We adopt datasets and corresponding prompts from promptsource (Bach et al., 2022). For all prompted samples in each dataset, we set a truncation of maximal 10,0000 samples per dataset and combine them together as the MIP dataset. Details of the prompted samples and datasets are provided in promptsource's GitHub repository[8].

**Information Extraction.** Based on the datasets from DeepStruct (Wang et al., 2022a), a multi-task language model pre-training approach for information extraction tasks, we create instructions and prompts for part of its datasets (as is shown in Table 12). We reformulate information extraction tasks into instruction tuning formats to allow zero-shot generalization to new extraction schema. For all prompted samples in each dataset, we set a truncation of maximal 20,0000 samples per dataset as there are fewer information extraction datasets than common language understanding and generation ones. For KELM (Agarwal et al., 2021) and PropBank (Kingsbury & Palmer) datasets, since their original size is gigantic, we sample 50,0000 samples for each of them from their prompted samples.

### C.2    DATA AND PROMPTS IN MIP FOR DEEPSTRUCT

Prompts and instructions for all datasets in DeepStruct (Wang et al., 2022a) are newly created by authors manually. The introduction, task description, and full prompts for each dataset are attached in the following sections. To allow template infilling, all prompts are written into Jinja[9] templates. When a dataset sample is provided in our format, Joinja engine will render it into a prompted sample with instruction.

A more systematic evaluation on GLM-130B's information extraction ability is left for a future work, as the concentration in this work is on the training and designing details of an LLM.

### C.2.1    DIALOGUE STATE TRACKING

We adopt Multiwoz 2.1 (Eric et al., 2020) dialogue state tracking dataset. The dataset is reformulated into two tasks, each with one prompt correspondingly:

- **Dialogue state tracking**: which asks the model to extract information from dialogues given a list of certain slots, e.g., `taxi_arrival_time` and `destination`.
- **Slot filling**: which model should fill in one provided slot and identify situations without answer.

---

[8]`https://github.com/bigscience-workshop/promptsource`
[9]`https://github.com/pallets/jinja`

**(Dialogue State Tracking, Prompt 0)**

```
Read the dialogues between "[User]" and "[Agent]",

{{text}}

identify and extract the information related to the following categories
 (from top to down):

- {{allowed_relations | join("\n- ")}}

in the form of "( [User] ; Y ; Z )": ||| {{format_triple(relations,
allowed_relations) | join(" ")}}
```

**(Slot Filling, Prompt 0)**

```
Given the following dialogue:

{{text}}

please answer the question: has "[User]" mentioned "{{allowed_relations[
relation_idx].split(': ') | join("'s ")}}" ? If yes, please write down
the answer from the dialogue; if not, please answer "not given".

Answer: ||| {% if filter_relation(relations, allowed_relations[
relation_idx]).__len__() > 0 %}{{filter_relation(relations,
allowed_relations[relation_idx])[0]['tail']}}{% else %}not given{% endif
 %}
```

### C.2.2  EVENT EXTRACTION

We adopt ACE05 (Walker & Consortium, 2005) event extraction datasets following the setting in (Wadden et al., 2019). The dataset is reformulated into two tasks with three prompts as follows:

- **Event Argument Extraction**: given a trigger in text and a list of its argument roles, the model is asked to extract the arguments from the provided text.
- **Argument Identification**: given a trigger and a certain argument role, the model is asked to extract the argument if it exists in the provided text; otherwise, the model should generate nothing.

**(Event Argument Extraction, Prompt 0)**

```
For the task of "Event Extraction", given a trigger one should extract
its related arguments conditioned on a list of potential roles.

Given the following list of roles:

- {{shuffle(allowed_arguments[trigger['event_type']].values()) | join("\
n- ")}}

extract related arguments of the trigger "{{trigger['text']}} ({{
allowed_triggers[trigger['event_type']]}})" in the following sentence:

{{text}}

Extractions: ||| {{format_triple(relations, "") | join(" ")}}
```

**(Event Argument Extraction, Prompt 1)**

```
TEST

1. (Event Extraction) {{text}}

Please write down ALL event arguments related to the trigger "{{trigger
['text']}} ({{allowed_triggers[trigger['event_type']]}})" marked with "[
 ]", given the following categories:

- {{shuffle(allowed_arguments[trigger['event_type']].values()) | join("\
n- ")}}

Answer: ||| {{format_triple(relations, "") | join(" ")}}
```

**(Argument Identification, Prompt 0)**

```
Let extract event related arguments!

In the following passage, an argument with the type "{{query_arg}}" is
related to the event trigger "{{trigger['text']}} ({{allowed_triggers[
trigger['event_type']]}})":

{{text}}

The argument should be (copy from the context if you find it; if not, do
 not generate): ||| {{filter_type(relations, query_arg) | join(" ")}}
```

### C.2.3   JOINT ENTITY AND RELATION EXTRACTION

Joint entity and relation extraction aims to recognize named entities in a piece of text and judge the relationships between them. It is closely related to knowledge acquisition, where the ultimate target is to structuring the unstructured web contents into knowledge triples (e.g., `(London, capital_of, Britain)`). The task can be formulated into either a pipeline framework (a combination of named entity recognition and relation extraction), or end-to-end training.

In this work, we adopt three classical joint entity and relation extraction datasets: CoNLL04 (Roth & Yih, 2004), NYT (Riedel et al., 2010), and ACE2005 (Walker & Consortium, 2005). In GLM-130B, we follow (Wang et al., 2022a) to formulate such challenges into sequence-to-sequence generation, where our inputs are raw texts and outputs are triples. We only conduct relation-related tasks for these datasets here, and leave the entity-related ones to the named entity recognition section.

- **Relation Extraction**: here we extract knowledge triples consisting of "head entity", "relation", and "tail entity", given a list of relation candidates. For example, given the input "*In Kunming the 800-some faculty and student established the National Southwestern Associated University.*", the model output could be `(National Southwestern Associated University, location of formation, Kunming)`.
- **Conditional Relation Extraction**: given a single relation candidate, judge if the input text contains the relation. If so, extraction all related triples; if not, do not generate.
- **Knowledge Slot Filling**: assign a certain entity from text, and ask the model to extract all triples that takes the entity as the head.
- **Relation Classification**: given two entities from texts, ask the model to judge the relation between them based on a list of candidate relations.

**(Relation Extraction, Prompt 0)**

```
Can you figure out all triples regarding the relations of "{{shuffle(
allowed_relations) | join('", "')}}" from the sentence? List them in the
 shape of "( X ; Y ; Z )":

{{text}} => ||| {{format_triple(relations, allowed_relations) | join("
")}}
```

**(Conditional Relation Extraction, Prompt 0)**

```
Conditioned on the relation "{{allowed_relations[relation_idx]}}", what
knowledge triples can be extracted from:

{{text}}

Please write them down here: ||| {{format_triple(relations, [
allowed_relations[relation_idx]]) | join(" ")}}
```

**(Knowledge Slot Filling, Prompt 0)**

```
{% if entity_types.__len__() > 0 %}
In the sentence

{{text}}

the X = "{{entities[entity_idx]}}" is an entity of the type "{{
entity_types[entity_idx]}}". Extract all possible triples contains "{{
entities[entity_idx]}}" in the form of ( X ; Y ; Z ), given the
following candidate properties Y:

{% for r in allowed_relations %}- {{r}}
{% endfor %}
Answer: ||| {% for r in relations %}{% if r['head'][0] == entities[
entity_idx] %}{{format_triple([r], allowed_relations) | join(" ")}}{%
endif %}{% endfor %}
{% endif %}
```

**(Relation Classification, Prompt 0)**

```
QUIZ

1. Given the candidate relations:

- {{shuffle(allowed_relations) | join("\n- ")}}

what is the relation between "{{relations[triple_idx]['head'][0]}}" and
"{{relations[triple_idx]['tail'][0]}}" in the following sentence?

{{text}}

Answer: ||| {{relations[triple_idx]['relation']}}
```

Nevertheless, existing joint entity and relation extraction datasets have very limited relation schema. For example, CoNLL04 only contains five different relations; the most diverse NYT dataset contains 24 Freebase predicates. To allow the model to capture a diverse range of potential verbalized predicates, we extend the task with automatically generated knowledge-text aligned data from KELM (Agarwal et al., 2021). We do not include other distantly supervised dataset (e.g., T-Rex (Elsahar et al., 2018)) since they can be extremely noisy.

For KELM data, since it is based on the full Wikidata schema (which contains too many relations to be enumerated), we create two KELM-specific prompts for the task of **Relation Extraction** and **Knowledge Slot Filling**:

**(Relation Extraction, Prompt 1, KELM ONLY)**

```
{# kelm #}
Can you figure out all knowledge triples regarding whole Wikidata
properties from the sentence? List them in the shape of "( X ; Y ; Z )":

{{text}} => ||| {{format_triple(relations, "") | join(" ")}}
```

**(Knowledge Slot Filling, Prompt 1, KELM ONLY)**

```
{# kelm #}
Given the entity "{{entities[entity_idx]}}" marked with "[" and "]" in
the context:

{{text}}

please list all triples related to it (do not generate if there is no
answer): ||| {% for r in relations %}{% if r['head'][0] == entities[
entity_idx] %}{{format_triple([r], "") | join(" ")}}{% endif %}{% endfor
 %}
```

### C.2.4 NAMED ENTITY RECOGNITION

Named entity recognition is a task which targets identifying named entities from raw text corpus and assign them with proper entity types. For example, in the sentence "*In 1916 GM was reincorporated in Detroit as "General Motors Corporation".*", `General Motors Corporation` could be of entity type `organization`. We design two different types of tasks based on named entity recognition datasets CoNLL03 (Sang & Meulder, 2003), OntoNotes 5.0 (Pradhan et al., 2013), and GENIA (Ohta et al., 2002). We also include named entity recognition sub-tasks from joint entity and relation datasets.

- **Named Entity Recognition**: given a certain list of possible entity types (e.g., `location`, `person`, `organization`), extract all related entities from the provided text content.
- **Entity Typing**: entity typing is one of the important derivative tasks from named entity recognition. It aims to classify the correct type of an entity mention (without entity types), and is often appended to the entity mention extraction as post-processing.

**(Named Entity Recognition, Prompt 0)**

```
Given the following list of entity types:

Z = {{shuffle(allowed_types) | join(", ")}}

please extract all mentioned entities from left to right in the sentence
, in the form of "( X ; instance of ; Z )".

{{text}} => ||| {% for entity, type in zip(entities, entity_types) %}(
{{entity}} ; instance of ; {{type}} ) {% endfor %}
```

**(Entity Typing, Prompt 0)**

```
Extract all entity mentioned in the sentence with entity type "{{
allowed_types[type_idx]}}" in the form of "( X ; instance of ; {{
allowed_types[type_idx]}} )"

{{text}} => ||| {% for entity, type in zip(entities, entity_types) %}{%
if type == allowed_types[type_idx] %}( {{entity}} ; instance of ; {{type
}} ) {% endif %}{% endfor %}
```

**(Entity Typing, Prompt 1)**

```
List all "{{allowed_types[type_idx]}}" entities appeared in the
following passage, joined by " | ":

{{text}} => ||| {{filter_type(zip(entities, entity_types), allowed_types
[type_idx]) | join(" | ")}}
```

**(Entity Typing, Prompt 2)**

```
{% if entity_types.__len__() > 0 %}
Based on the list of potential entity types and ignore their order:

- {{shuffle(allowed_types) | join("\n- ")}}

the entity "{{entities[entity_idx]}}" marked with "[" and "]" in the
following sentence:

{{text}}

belongs to ||| {{entity_types[entity_idx]}}
{% endif %}
```

### C.2.5 RELATION CLASSIFICATION

Relation classification is a fundamental task in information extraction, which identifies the relationships from a list of candidates between two given entities. The problem is a long standing one as it suffers from outrageous cost of data labeling, since manual labeling on knowledge-intensive tasks requires educated annotators that charges high. A *de facto* data creation method in relation extraction relies on distant supervision, which aligns existing knowledge triples in knowledge bases to text contents automatically, and assume that such alignments are correct in certain conditions. Here we only include TacRED (Zhang et al., 2017) dataset and create several different tasks based on it.

- **Relation Classification**: the most traditional task formulation. Given two entities from text and classify their relation from a list of candidates. The form can be either answering the relation directly or in the form of a triple (similar to relation extraction).
- **Knowledge Slot Filling**: change the task into given head entity and relation, to identify whether the tail entity exists in the input text. If not, generate nothing.
- **Yes or No Question**: turn the problem into a task similar to natural language inference. For example, given the sentence "*The series focuses on the life of Carnie Wilson, daughter of Brian Wilson, founder of the Beach Boys.*", the model will be asked to judge the correctness of a triple such as `Carnie Wilson, father, Brian Wilson` by answering "yes" or "no".

**(Relation Classification, Prompt 0)**

```
{% if entity_types.__len__() > 0 %}
Given the following categories of relations:

- {{shuffle(allowed_relations.values()) | join("\n- ")}}

predict the relation between "{{relations[0]['head']}}" and "{{relations
[0]['tail']}}" in the following sentence:

{{text}}

The relation should be : ||| {{allowed_relations[relations[0]['relation
']]}}
{% endif %}
```

**(Relation Classification, Prompt 1)**

```
1. (Relation Extraction) Answer the relation between entities in the
form of "( X ; Y ; Z )":

{{text}}

The relation between "{{relations[0]['head']}}" and "{{relations[0]['
tail']}}" is: ||| ( {{relations[0]['head']}} ; {{allowed_relations[
relations[0]['relation']]}} ; {{relations[0]['tail']}} )
```

**(Knowledge Slot Filling, Prompt 0)**

```
Based on the sentence provided below, infer the missing argument asked
by the question:

{{text}}

Question: What/Who/Where is "{{relations[0]['head']}}" {{
allowed_relations[relations[0]['relation']]}} ?

Answer: ||| {{relations[0]['tail']}}
```

### C.2.6   SEMANTIC ROLE LABELING

Semantic role labeling is a long-standing information task that wants to identify the semantic arguments related to a given predicate in a sentence. For example, in the sentence "*Grant was employed at IBM for 21 years where she held several executive positions.*" and the predicate "`employed`" in it, semantic role labeling identifies the `Grant` as the subject and `IBM` as the second object.

We create two different tasks based on semantic role labelling datasets CoNLL05 (Carreras & Màrquez, 2005), CoNLL12 (Pradhan et al., 2013), and PropBank (Kingsbury & Palmer).

- **Semantic Role Labeling**: the traditional task form, where a verb (i.e., predicate) is annotated in text and the model is asked to generate related semantic roles.
- **Semantic Role Filling**: given a verb and and a potential semantic role, the model is asked to judge whether the role exists in the sentence and generate it.
- **Predicate Recognition**: given a segment of a sentence and its corresponding semantic role, identify which verb it is related to.

**(Semantic Role Labeling, Prompt 0)**

```
Provided with the target verb "{{verb}}" marked with "[" and "]" in the
following sentence, find out its "{{allowed_types[type_idx]}}":

{{text}} => ||| {% for entity, type in zip(entities, entity_types) %}{%
if type == allowed_types[type_idx] %}{{entity}}{% endif %}{% endfor %}
```

**(Semantic Role Filling, Prompt 0)**

```
Given the following list of argument types:

Z = {{allowed_types | join(", ")}}

find out all arguments related to verb "{{verb}}" mentioned in the
following sentence from left to right, in the form of "( X ; instance of
 ; Z )".

{{text}} => ||| {% for entity, type in zip(entities, entity_types) %}(
{{entity}} ; argument type ; {{type}} ) {% endfor %}
```

**(Predicate Recognition, Prompt 0)**

```
FINAL EXAM

1. Based on the fact that "{{entities[entity_idx]}}" is a "{{
entity_types[entity_idx]}}", which verb in the following sentence should
 it related to?

{{text}}

Answer: ||| {{verb}}
```

## C.3 RESULT SOURCES FOR GPT-3, BLOOM-176B, AND OPT-175B

Here we describe the result sources for GPT-3, BLOOM-176B, and OPT-175B. Other LLMs we may compare are mostly completely closed-sourced; thus, their results are all taken from existing preprints, publications, or the results stored in BIG-bench repository[10].

For GPT-3, while most of its results in this paper are taken from existing literature if not specified, the rest were acquired via our own requesting OpenAI Danvici API are explicitly mentioned. For BLOOM-176B and OPT-175B, if without specific annotation, their results are:

- Taken from the OPT paper (Zhang et al., 2022).
- Taken from the EAI-Eval BigScience Arch&Scale - Google Sheet[11].
- Taken from BigScience evaluation results repository in Huggingface Datasets[12].

Specifically, we cannot evaluate OPT-175B by ourselves as we are still not officially granted the checkpoint, though we have sent several applications in the past few months.

## C.4 PILE TEST-SET EVALUATION

Pile evalution (Gao et al., 2020) is a comprehensive language modeling benchmark which originally includes 22 different text datasets from diverse domains. We report our results over a part of 18 datasets with previously reported baseline results (Lieber et al., 2021). Different from traditional language modeling benchmarks, Pile evaluation report the BPB (bits-per-byte) perplexity to avoid the mismatch comparison between models with different vocabularies. Because in general, language models with a larger vocabulary will be favored in perplexity comparison if not restricted. In the evaluation, we strictly follow the setting in (Gao et al., 2020), leveraging [gMASK] and a context-length of 1,024 with bidirectional attention, and the rest 1024 tokens to calculate BPB in an autoregressive manner. The weighted average BPB are calculated based on each shared dataset's ratio in Pile training-set (Gao et al., 2020).

Table 13: GLM-130B and its similar-sized LLMs' BPB results on Pile test-set.

|  | Jurassic-1 | GPT-3 | GLM-130B |
|---|---|---|---|
| dm_mathematics | 1.040 | 1.370 | **0.786** |
| ubuntu_irc | **0.857** | 0.946 | 0.977 |
| opensubtitles | **0.879** | 0.932 | 0.889 |
| hackernews | **0.869** | 0.975 | 0.873 |
| books33 | 0.835 | **0.802** | 0.803 |
| pile_cc | **0.669** | 0.698 | 0.771 |
| philpapers | 0.741 | **0.723** | 0.766 |
| gutenberg_pg_19 | 0.890 | 1.160 | **0.821** |
| arxiv | 0.680 | 0.838 | **0.570** |
| stackexchange | 0.655 | 0.773 | **0.611** |
| nih_exporter | **0.590** | 0.612 | 0.614 |
| pubmed_abstracts | **0.587** | 0.625 | 0.610 |
| uspto_backgrounds | **0.537** | 0.566 | **0.537** |
| pubmed_central | 0.579 | 0.690 | **0.510** |
| freelaw | 0.514 | 0.612 | **0.499** |
| github | 0.358 | 0.645 | **0.329** |
| enron_emails | 0.621 | 0.958 | **0.604** |
| youtube_subtitles | 0.825 | 0.815 | **0.746** |
| Weighted Avg. | 0.650 | 0.742 | **0.634** |

The detailed metrics on Pile test-set are reported in Table 13. We observe that compared to GPT-3, GLM-130B has a noticeable weaker performance on phil_papers and pile_cc, which is likely because of GLM-130B's bilingual natural and lack of more diverse and high-quality private collected corpora.

---

[10] https://github.com/google/BIG-bench
[11] https://docs.google.com/spreadsheets/d/1CI8Q9RCblLRzUOPJ6ViqBmo284-8oj luQ-CmaEuhuv0
[12] https://huggingface.co/datasets/bigscience/evaluation-results/tree/ma in/bloom/bloomzeval/transformers/evaluation_val

## C.5 BIG-BENCH-LITE EVALUATION

Recent works (Wei et al., 2022c; Wang et al., 2022c) reveal that LLMs are capable to do reasoning beyond conventional language tasks. As a response, BIG-bench (Srivastava et al., 2022) is recently set up by crowdsourcing new types of tasks from global researchers to test LLMs unexplored abilities. For economical consideration, we evaluate GLM-130B on an official subset of original 150-task BIG-bench, the BIG-bench-lite with 24 tasks. These tasks can be categorized into two types: one is based on multiple-choice question answering with answer options, and another is direct generation without options. For the first category, we assess the probability of each option's full content and pick the largest one as the answer; for the second one, we generate the answer using greedy decoding. All evaluations done in BIG-bench are based on [MASK], since answers here are usually short pieces of texts. All results on 24 BIG-bench-lite (Srivastava et al., 2022) datasets of three LLMs are shown in Table 14 and

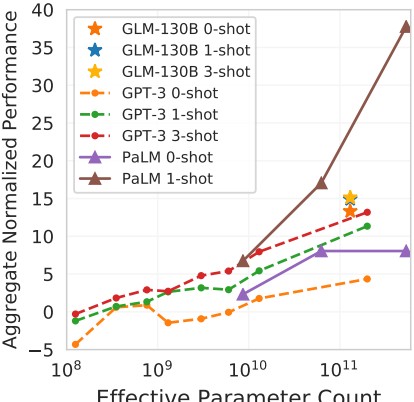

Figure 16: A full scope of BIG-bench-lite (24 tasks) evaluation.

Figure 16. We just adopt the original prompts from BIG-bench and use the official implementation to generate priming examples for few-shot evaluation and to calculate the final scores.

## C.6 MMLU EVALUATION

All results on 57 MMLU (Hendrycks et al., 2021) datasets of GLM-130B and BLOOM 176B are shown in Table 15. In Section 5.2, we report weighted average accuracy (i.e., accuracy average per sample, rather than by discipline) of GLM-130B, GPT-3 175B, and BLOOM 176B.

Below is a prompted example with 1-shot priming. We predict the probability on ['A', 'B', 'C', 'D'] at the next token, and take the one with the maximal probability as the answer.

**(MMLU 1-shot Example)**

```
The following are multiple choice questions about philosophy.

According to d'Holbach, people always act according to ______.
(A) free choices (B) dictates of the soul (C) necessary natural laws (D)
 undetermined will
Answer: (C) necessary natural laws

Epicurus holds that philosophy is:
(A) not suitable for the young. (B) not suitable for the old. (C)
important, but unpleasant. (D) none of the above.
Answer: (
```

## C.7 CHINESE LANGUAGE UNDERSTANDING EVALUATION

Here we elaborate the prompts we use for CLUE (Xu et al., 2020) and FewCLUE (Xu et al., 2021) evaluation. On Chinese datasets, prompting meets some challenges as Chinese texts are organized by single characters rather than words, leading to unequal length of verbalizers in many cases. Albeit dataset-specific calibration (Wang et al., 2021; Wu et al., 2021) can help to mitigate the issue, the too specified technique can be complicated in implementation. Our evaluation in this paper adopts a more easy to solve method leveraging GLM-130B's unique features. As GLM-130B is a bilingual LLM with English MIP, we adopt English prompts and verbalizers from similar tasks in (Bach et al., 2022) for Chinese dataset evaluation and find such strategies to be quite effective. In terms of evaluation metrics, except for DRCD and CMRC2018 two question answering datasets which reports EM, other datasets report accuracy.

## C.8 NATURAL LANGUAGE GENERATION

Natural language generation, or conditional natural language generation here, refers to tasks that require generating text based on the given information, such as tables and documents. We evaluate GLM-130B on data-to-text and summarization tasks. The datasets include WebNLG 2020 (Castro Ferreira et al., 2020), Clean E2E NLG (Dušek et al., 2019) and WikiLingua (Scialom et al., 2020) from GEM generation benchmark (Gehrmann et al., 2021). We select full WebNLG 2020 and the Clean E2E NLG in the test set and randomly select 5000 test examples from WikiLingua following the practice in (Chowdhery et al., 2022). Following the settings in PaLM, the prompt used for the Summarization tasks is "Summarize the following article:" and the prompt used for the Data-to-Text tasks is "Verbalize:". An exception is E2E, where we process the data using the prompt "generate-gramatically-correct-text from" provided in promptsource for GLM-130B and GPT-3 175B (Davinci). All evaluations are one-shot, and the demonstration samples are randomly sampled from the training set. We report the F-measure of ROUGE-2, ROUGE-L (Lin, 2004) and BLEURT-20 (Pu et al., 2021). We compare our model with LaMDA, GPT-3 175B (Davinci), and PaLM, where the results of LaMDA and PaLM are reported by (Chowdhery et al., 2022), and we evaluate GPT-3 175B (Davinci) through OpenAI API.[13]

Our results are presented in Table 16. It shows that GLM-130B has better performances than LaMDA and GPT-3 (Davinci) on all tasks. In the Data-to-text task, GLM-130B performs slightly worse than PaLM-540B, while in the summary task, GLM-130B has even higher ROUGE results. We also ablate GLM-130B to unidirectional to demonstrate the advantage of bidirectional attention. Unidirectional GLM-130B underperforms GPT-3 175B in all three datasets, but when it shifts to bidirectional attention, there is an instant boost, making GLM-130B even comparable to PaLM-540B in a few cases. It indicates that bidirectional attention over the provided context (i.e., prefix) can also be beneficial for text generation missions.

Table 16: 1-shot GEM English natural language generation tasks (WebNLG, E2E, and WikiLingua). We compare two versions of GLM-130B (uni: unidirectional attention, bi: bidirectional attention), showing that bidirectional attention can also improve conditional generation's performance.

| Task | Dataset | Metric | LaMDA 137B | GPT-3 175B (Davinci) | GLM-130B uni | GLM-130B bi | PaLM-540B |
|------|---------|--------|------------|----------------------|--------------|-------------|-----------|
| Data to Text | WebNLG | ROUGE-2 | 30.5 | 29.9 | 25.3 | 38.5 | **44.4** |
| | | ROUGE-L | - | 41.2 | 36.7 | 49.3 | **53.8** |
| | | BLEURT-20 | - | 59.0 | 53.2 | 67.7 | **73.9** |
| | E2E | ROUGE-2 | 29.2 | 30.3 | 30.9 | 33.9 | **35.2** |
| | | ROUGE-L | - | 39.2 | 40.0 | 42.6 | **43.9** |
| | | BLEURT-20 | - | 64.5 | 65.0 | 68.1 | **69.7** |
| Summary | WikiLingua | ROUGE-2 | 5.4 | 7.2 | 5.8 | **10.4** | 9.9 |
| | | ROUGE-L | - | 18.9 | 16.4 | **23.4** | 20.6 |
| | | BLEURT-20 | - | 41.2 | 39.4 | 45.0 | **47.7** |

**(E2E Example, without demonstration sample)**

```
Aleksandr_Prudnikov , height , 185.0 (centimetres).
FC_Spartak_Moscow , ground , Otkrytiye_Arena.
Aleksandr_Prudnikov , club , FC_Spartak_Moscow.
Verbalize:
```
- - - - - - - - - - - - - - - - - - - - - - - - - - - - - - - - - - - - - - - - - - - - - - - - - -

**Groundtruth:** 185 centimetre tall Aleksandr Prudnikov played for the Otkrytiye Arena based FC Spartak, Moscow.

**GPT-3 175B (Davinci):** Aleksandr Prudnikov is a midfielder for FC Spartak Moscow, a football (soccer) club based in Moscow, Russia.

**GLM-130B:** Aleksandr Prudnikov is 185.0 cm tall and plays for FC Spartak Moscow.

---

[13]We use ROUGE implementation at https://github.com/google-research/google-research/tree/master/rouge and BLEURT-20 implementation at https://github.com/google-research/google-research/tree/master/rouge, whose checkpoint is available at https://storage.googleapis.com/bleurt-oss-21/BLEURT-20.zip

**(E2E Example, without demonstration sample)**

```
Combine all of the following data into a concise and grammatically
correct text:
name : Blue Spice
eatType : coffee shop
area : riverside
```

**Groundtruth:** At the riverside, there is a coffee shop called The Blue Spice.

**GPT-3 175B (Davinci):** Blue Spice is a riverside coffee shop which is located on the corner of River Street and Riverbank Street.

**GLM-130B:** There's a coffee shop that serves coffee in the riverside area, Blue Spice.

**(WikiLingua Example, without demonstration sample)**

```
The majority of your customers will search for you online, so it's
essential to have a user-friendly website. At the very least, your
website should include information about your business and your history
in the moving industry, details about the quoting process, contact
information, and a description of the services you offer. If possible,
allow customers to schedule quotes online, view your availability, or
read testimonials from other customers. One of the easiest ways to start
 your business is by helping people you already know with their moves.
You can be on the lookout for any announcements related to moving that
your friends make on social media. Once you have provided good service
to friends, they are likely to recommend you to others. In order to
spread the word about your business, have some professional looking
promotional materials printed and distribute them around your community.
 You can distribute business cards at public events, tuck them into
local bulletin boards, or even print them in directories, yearbooks, and
 other local print media. Flyers can be mailed, posted in public places,
 or distributed to businesses that might be able to refer customers to
like you, such as furniture stores. Make sure you have a professional,
recognizable logo that is consistent across all of your marketing
materials. Another way to get your business's name out there is to make
yourself visible. Whether it's by working with partners at local events,
 volunteering, or using your vehicle for an ad campaign, visibility is
key for driving business. Build relationships with influential people in
 your community. Realtors are a great source of referrals to movers, as
are the owners of local furniture stores or the office staff at a large
apartment complex. You can use directory sites like Craigslist to
advertise your services to people in your local community for free.
Social media is also a great way to spread the word about your business.
 There are many options for advertising, depending on your budget and
your target market. Consider options like PPC advertising, television
and radio commercials, newspaper ads, direct mail flyers, or memberships
 with referral services. The best thing you can do to grow your business
 is to provide excellent service to your customers. Be sure to always be
 on time, be friendly, be respectful of your customers' belongings, and
offer accurate price quotes. Be sure to ask your happy customers to post
 reviews about you online. New customers will see these reviews and will
 have more faith in you as a reputable company if they see that others
have had a good experience. You will inevitably end up having to deal
with an unhappy customer at some point, but do your best to resolve the
problem to the customer's satisfaction. The last thing you want is
negative reviews about your business circulating the internet! Summarize
 the article:
```

**Groundtruth:** Create a website. Start with friends. Distribute business cards and flyers. Network for promotion. Seek out free advertising opportunities. Consider paid advertising. Gain a good reputation.

**GPT-3 175B (Davinci):** Market yourself online. Look for opportunities to market your services offline.

**GLM-130B:** Have a user-friendly website. Promote your business through social media. Build relationships with influential people in your community. Use directory sites like Craigslist to advertise your services.

Table 17: Winograd-style tasks evaluation (Winogender and Winograd273). All scores are accuracy. K refers to number of shots. *PaLM 540B did not report the exact 0-shot Winogender result, so we have to estimate a value from its plotted diagram.

|  | K | GPT-3 (Davinci) | OPT 175B | BLOOM 176B | PaLM 540B | Chinchilla | Gopher 280B | GLM-130B |
|---|---|---|---|---|---|---|---|---|
| Winogender | 0 | 64.2 | 54.8 | 49.1 | 75.0* | 78.3 | 71.4 | 79.7 |
|  | 1 | 62.6 | - | 53.1 | 79.4 | - | - | 80.7 |
| Winograd273 | 0 | 88.3 | 52.9 | 49.1 | 90.1 | - | - | 84.3 |

Table 18: Closed-book question answering (Natural Questions, StrategyQA).

|  | GPT-3 (Davinci) | BLOOM 176B | PaLM 540B | Chinchilla | Gopher 280B | GLM-130B |
|---|---|---|---|---|---|---|
| Natural Questions (EM) | 14.6 | 13.1 | 21.2 | 16.6 | 10.1 | 11.7 |
| StrategyQA (Acc) | 52.3 | 49.8 | 64.0 | - | - | 60.6 |

Table 19: Commonsense reasoning (Commonsense QA, MC-TACO). K refers to number of shots.

|  | K | GPT-3 (Davinci) | OPT 175B | BLOOM 176B | GLM-130B |
|---|---|---|---|---|---|
| Commonsense QA (Acc) | 0 | 57.2 | - | 42.8 | 61.6 |
|  | 1 | 61.2 | - | - | 62.2 |
| MC-TACO (EM) | 0 | - | 12.4 | 13.1 | 13.6 |

## C.9   WINOGRAD-STYLE TASKS

We include the evaluation on Winograd-style tasks, which derives from the classical Winograd Schemas Challenge (Levesque et al., 2012) that aims to test coreference resolution in an ambiguous context for the machine to understand. Since in MIP, we have included the Winogrande (Sakaguchi et al., 2021) and SuperGLUE WSC (Wang et al., 2019), here we test on Winogender (Rudinger et al., 2018) and Winograd273 (Levesque et al., 2012). For Winogender, GPT-3's results are acquired from OpenAI API, and BLOOM's 1-shot result is evaluated by ourselves. For Winograd273, since existing works (Brown et al., 2020; Chowdhery et al., 2022) show that 1-shot learning brings almost no improvement, we only test the zero-shot result. Another thing to notice is that, despite GPT-style models (e.g., GPT-3, PaLM) adopting the "partial evaluation" described in (Radford et al., 2019), we find the prompt "`<sentence> The "<pronoun>" refers to [MASK]`" is better for GLM-130B and adopt it in the evaluation.

The results are presented in Table 17. GLM-130B performs the best across all evaluated LLM on Winogender, and marginally poorer than GPT-3 and PaLM on Winograd273.

## C.10   CLOSED-BOOK QUESTION ANSWERING

Closed-book question answering (CBQA) (Roberts et al., 2020) is a widely adopted task to evaluate language models' memorization of factual knowledge, on contrary to the traditional "open-book" evaluation. As we have included TriviaQA (Joshi et al., 2017) and WebQuestions (Berant et al., 2013) in the MIP training, here we choose Natural Questions (Kwiatkowski et al., 2019) and StrategyQA (Geva et al., 2021) as the evaluation datasets for CBQA.

The results are presented in Table 18. GLM-130B performs relatively poorer on Natural Questions and performs well on StrategyQA. GLM-130B's underperformance on Natural Questions, we speculate, potentially derives from the insufficiency fitting on English corpora, as it roughly only viewed

200B English tokens and thus does not memorize the detailed knowledge very well. Since CBQA seems to be a task that especially stresses memorization, as is indicated by Chinchilla (Hoffmann et al., 2022)'s a strong performance, we think with sufficient training later, GLM-130B can perform better.

## C.11   COMMONSENSE REASONING

Here we evaluate GLM-130B and some other LLMs on commonsense reasoning abilities. As we have included PIQA (Bisk et al., 2020), ARC (Clark et al., 2018), and OpenbookQA (Mihaylov et al., 2018) in the MIP training, we select another two widely adopted commonsense reasoning datasets in our evaluation: Commonsense QA (Talmor et al., 2019) and Multiple-choice Temporal Commonsense (MC-TACO, Zhou et al. (2019)). For Commonsense QA, we test the GPT-3 via OpenAI Davinci API, BLOOM-176B via its Huggingface Implementation, and GLM-130B using the prompt "answer_given_question_without_options" from promptsource (Bach et al., 2022). For StrategyQA, we follow the EM computation method provided in (Zhou et al., 2019).

The results are shown in Table 19. As we can see, GLM-130B performs the best on both Commonsense QA and MC-TACO across evaluated LLMs, demonstrating that GLM-130B has a good grasp of commonsense knowledge. OPT's results are not included due to the reason described in Appendix C.3.

## C.12   FIXED LABEL DATASETS: A CASE STUDY IN NATURAL LANGUAGE INFERENCE

As is discussed in Section 5, we adopt a rather strict criterion for selecting datasets for zero/few-shot learning in GLM-130B's evaluation due to the use of MIP. Nevertheless, the criterion significantly reduces the dataset we could currently evaluate, and especially some readers have doubted whether the restriction of not evaluating on MIP-seen fixed-label datasets is necessary (e.g., natural language inference (NLI)), and suggest that we may report them in an independent section to avoid confusion.

Frankly speaking, in such a setting GLM-130B's zero/few-shot learning could be quite advantageous. Below, we take NLI as a typical example to show GLM-130B's outperformance in the scenarios. We include 6 widely-used NLI datasets–which are not incorporated in GLM-130B's MIP training, as the benchmarks. The results are presented in Table 20, which shows that GLM-130B's "zero-shot" performance could be much better due to the seen task type.

Table 20: "Zero-shot" results of GLM-130B on 6 typical natural language inference (NLI) datasets. *DISCLAIMER: Despite the datasets are never seen, some other NLI datasets have been included in GLM-130B's MIP, making it different from the existing standard zero-shot setting.

|  | BLOOM 176B | OPT 175B | GLM-130B* |
|---|---|---|---|
| qnli (valid, median of 5 prompts) | 50.9 | 55.4 | 86.7 |
| mnli (valid, median of 15 prompts) | 35.5 | 36.0 | 85.7 |
| mnli_mismatched (valid, median of 15 prompts) | 35.5 | 36.0 | 84.6 |
| wnli (valid, median of 5 prompts) | 57.7 | 53.5 | 67.6 |
| glue/cola (valid, median of 5 prompts) | 39.0 | 44.4 | 57.6 |
| glue/mrpc (valid, median of 5 prompts) | 31.6 | 44.6 | 87.3 |

## C.13   SUPERGLUE

We also report our evaluation of GLM-130B on the SuperGLUE (Wang et al., 2019) benchmark, which consists 8 different natural language understanding challenges. Noted that these results are neither zero/few-shot nor fine-tuned results, because 7 out of 8 tasks' training sets have been included in GLM-130B's MIP training (except for ReCoRD) together with other 67 multi-task datasets; however, GLM-130B is also not individually fine-tuned on any of them. Therefore, these results are not for relative comparison for any other models', but only for readers' reference on GLM-130B's absolute ability.

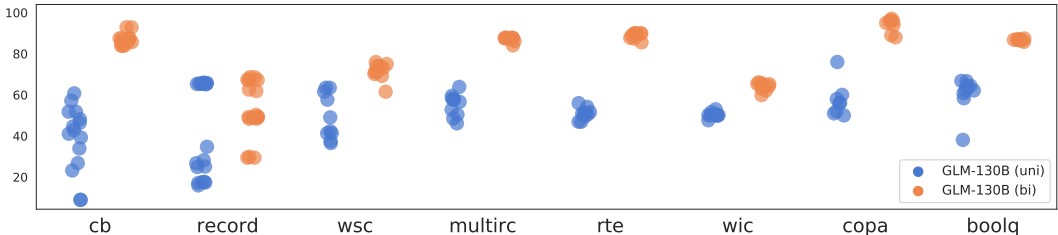

Figure 17: GLM-130B (uni and bi)'s untuned results on SuperGLUE development set, using prompt-source (Bach et al., 2022) prompts and task formulation. **DISCLAIMER: Noted that some of the SuperGLUE training sets have been included in the MIP training. We report the results here only for readers' reference.**

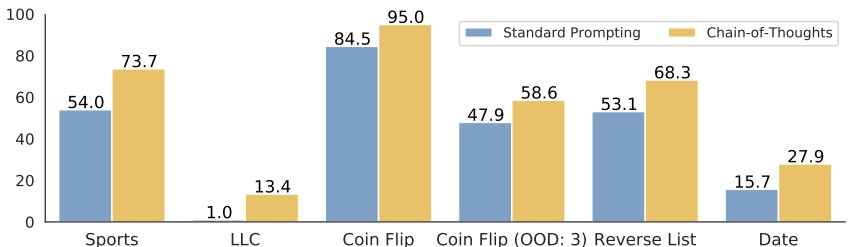

Figure 18: Chain-of-thought prompting can also improve GLM-130B's performance on reasoning tasks compared to standard prompting.

|          | BoolQ | CB    | COPA | MultiRC | ReCoRD | RTE   | WiC   | WSC  |
|----------|-------|-------|------|---------|--------|-------|-------|------|
| GLM-130B | 89.69 | 98.21 | 100  | 89.32   | 92.11  | 94.22 | 76.96 | 88.5 |

Table 21: The results of GLM-130B on the SuperGLUE dataset obtained using the P-tuning v2 (Liu et al., 2022). We report the Accuracy metric for all datasets except for MultiRC (F1a) and ReCoRD (F1).

The results are presented in Figure 17. We ablate the unidirectional and bidirectional GLM-130B to justify the usefulness of GLM objective in boosting LLMs' ability to understand. Each point in the figure refers to a prompt-specific result, for which the prompt is from the promptsource (Bach et al., 2022) repository. We adopt the task formulation from promptsource, too. As we can observe, GLM (bi) has much fewer variances and higher performances on all tasks. For some of the tasks (such as CB, MultiRC, RTE, COPA, and BoolQ), GLM-130B can even achieve over 80% accuracy.

We also attempted to fine-tune GLM-130B on the SuperGLUE dataset. However, we encountered the issue of rapid overfitting within a single epoch when we used full parameter fine-tuning on downstream tasks. This resulted in poor performance on the validation set. To address this issue, we explored the use of efficient parameter fine-tuning methods, which tune only a small number of parameters and are less prone to overfitting. After experimenting with several methods, we use P-Tuning v2 (Liu et al., 2022), which demonstrated comparable results to full parameter fine-tuning in GLM-130B, but with only 0.1% to 3% of tuned parameters. The results of our experiments with P-Tuning v2 are presented in Table 21.

## C.14 CHAIN-OF-THOUGHT PROMPTING

We evaluate the chain-of-thought prompting performance on **Last letter concatenation** (LLC), **Coin Flip**, **Reverse List**, and two tasks from BIG-bench Srivastava et al. (2022) **Sports** understanding, and **Date** understanding, following the setting in Wei et al. (2022c). The results are shown in Figure 17. We find that chain-of-thought prompting can improve GLM-130B's performance on symbolic reasoning and commonsense reasoning.

**Log-scaling Ability Tasks**

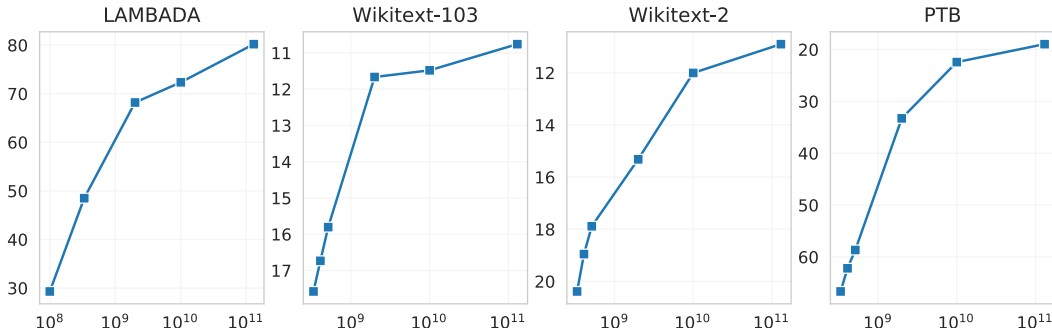

Figure 19: Log-scaling ability tasks of GLM-130B. These tasks' performance grows logarithmically with the amount of GLM parameters. Most of traditional NLP tasks fall into the same pattern.

**Last letter concatenation (LLC)**. The task asks the model to concatenate the last letters of words in a name (e.g., "Elon Musk" -> "nk"). We generate full names by randomly concatenating the top 1000 first and last names from name census data[14].

**Coin flip**. This task asks the model to answer whether a coin is still heads up after people either flip or don't flip it beginning from being heads up. (e.g., "A coin is heads up. Phoebe flips the coin. Osvaldo does not flip the coin. Is the coin still heads up?" -> "no"). We additionally evaluate on the scenario where the number of people in the query examples is larger than that in the in-context examples, i.e. the out-of-distribution (OOD) setting.

**Reverse List**. This task asks the model to reverse the order of a list of everyday objects (e.g., "cigar, umbrella, key, gum, alarm" -> "alarm, gum, key, umbrella, cigar"). We generate the lists by randomly sampling from the vocabulary of everyday objects[15].

**Sports**. This task asks the model to judge the truthfulness of a statement about a sports player (e.g., "Joao Moutinho caught the screen pass in the NFC championship" -> "false").

**Date**. This task asks the model to infer the data from a given context (e.g., "2015 is coming in 36 hours. What is the date one week from today in MM/DD/YYYY?" -> "01/05/2015").

We use the same examples and chains as Wei et al. (2022c). For each task, we try two different formats of prompts and both unidirectional and bidirectional attention mechanism and report the best performance. The first format is "Question: {context} Answer: {target}". The second one is to add serial numbers before examples in the first format of prompts. The results are presented in Figure 18.

## D  SCALING AND EMERGENT ABILITIES IN GLM-130B

Scaling up pre-trained language models has been proven to boost downstream performance on a wide range of tasks continually. His, emergent abilities which are unpredictable from smaller scales. To illustrate this, we conducted extensive experiments to explore the scaling property and emergent abilities. Following prior literature (Wei et al., 2022b), we categorize the NLP tasks into two types based on our observations.

- **Log-scaling Ability Tasks (Cf. Figure 19)**: where the task performance grows logarithmically with the number of model parameters. Typical tasks and datasets include LAMBADA, Wikitext-103, Wikitext-2, Penn Tree Bank.
- **Emergent Ability Tasks (Cf. Figure 20)**: where the task performance only soars up when the amount of model parameters reaches a certain threshold. Typical tasks and datasets include:

---

[14]https://namecensus.com
[15]https://www.vocabulary.com/lists/189583

**Emergent Ability Tasks**

Figure 20: Emergent ability tasks of GLM-130B. These tasks' performance does not grow much until the model size reaches a certain threshold (e.g., 100B or 10B). After reaching the threshold, the model performance soars up quickly. The BIG-bench (Srivastava et al., 2022) benchmark collects many of these challenges.

MMLU, hindu_knowledge, crass_ai, implicatures, understanding_fables, modified_arithmetic, implicit_relations, and gre_reading_comprehension from BIG-bench (Srivastava et al., 2022).

In line with the observation in (Wei et al., 2022b), we show that GLM-130B also presents the two similar scaling behaviors to other LLMs such as GPT-3, LaMDA, and PaLM. Though why and how LLMs present these intriguing properties remain unclear, GLM-130B provides open opportunities for all researchers to test and understand the reason behind them.

Table 11: Full configurations for GLM-130B training

| Configuration Key | Value |
| --- | --- |
| adam_beta1 | 0.9 |
| adam_beta2 | 0.95 |
| adam_eps | 1e-08 |
| aggregated_samples_per_sequence | 4 |
| attention_dropout | 0.1 |
| attention_softmax_in_fp32 | True |
| average_block_length | 3 |
| bias_dropout_fusion | True |
| checkpoint_activations | True |
| checkpoint_in_cpu | False |
| checkpoint_num_layers | 1 |
| clip_grad | 1.0 |
| contigious_checkpointing | False |
| cpu_optimizer | False |
| data_parallel_size | 24 |
| deepnorm | True |
| distributed_backend | nccl |
| eval_interval | 1000 |
| eval_iters | 3 |
| ffn_hidden_size | 32768 |
| fp16 | True |
| global_batch_size | 4224 |
| glu_activation | geglu |
| gpt_prob | 0.7 |
| hidden_dropout | 0.1 |
| hidden_size | 12288 |
| hysteresis | 2 |
| init_method_std | 0.0052 |
| init_method_xavier_uniform | False |
| initial_loss_scale | 65536 |
| layernorm_epsilon | 1E-05 |
| learnable_rotary_embedding | False |
| length_per_sample | 2000 |
| log_interval | 1 |
| loss_scale | 0 |
| loss_scale_window | 2000 |
| lr | 8e-05 |
| lr_decay_iters | None |
| lr_decay_samples | 197753905 |
| lr_decay_style | cosine |
| lr_warmup_samples | 1098632 |
| make_vocab_size_divisible_by | 768 |
| mask_prob | 0.15 |
| masked_softmax_fusion | True |
| micro_batch_size | 1 |
| min_gmask_ratio | 0.2 |
| min_loss_scale | 1.0 |
| min_lr | 8e-06 |
| multitask_ratio | 0.05 |
| num_attention_heads | 96 |
| num_layers | 70 |
| onnx_safe | None |
| optimizer | adam |
| partition_activations | True |
| pipeline_model_parallel_size | 8 |
| position_embedding_type | rotary |
| rampup_batch_size | 192, 24, 5493164 |
| save_interval | 250 |
| seed | 1234 |
| seq_length | 2048 |
| short_seq_prob | 0.02 |
| shrink_embedding_gradient_alpha | 0.1 |
| single_span_prob | 0.02 |
| split | 949,50,1 |
| tensor_model_parallel_size | 4 |
| tokenizer_type | IceTokenizer |
| weight_decay | 0.1 |
| zero_contigious_gradients | False |
| zero_reduce_bucket_size | 500000000 |
| zero_reduce_scatter | False |
| zero_stage | 1 |
| zero-optimization.allgather_bucket_size | 500000000 |
| tokenizer_type | IceTokenizer |
| weight_decay | 0.1 |
| world_size | 768 |
| zero_contigious_gradients | FALSE |
| zero_reduce_bucket_size | 500000000 |
| zero_reduce_scatter | FALSE |
| zero_stage | 1 |
| zero-optimization.allgather_bucket_size | 500000000 |

Table 12: The 74 datasets involved in Multi-task Instruction Pre-training (MIP). Datasets from T0-PromptSource (Sanh et al., 2022; Bach et al., 2022) are named in their Hugging Face datasets identifiers. Datasets from DeepStruct (Wang et al., 2022a) are described in Appendix C.2.

| Task | Dataset | Task | Dataset |
|---|---|---|---|
| Coreference Resolution | super_glue/wsc.fixed | Multi-choice QA | cos_e/v1.11 |
| Coreference Resolution | winogrande/winogrande_xl | Multi-choice QA | cosmos_qa |
| Natural Language Inference | super_glue/cb | Multi-choice QA | dream |
| Natural Language Inference | super_glue/rte | Multi-choice QA | openbookqa/main |
| Natural Language Inference | anli | Multi-choice QA | qasc |
| Paraphrase Identification | glue/mrpc | Multi-choice QA | quail |
| Paraphrase Identification | glue/qqp | Multi-choice QA | quarel |
| Paraphrase Identification | paws/labeled_final | Multi-choice QA | quartz |
| Closed-Book QA | ai2_arc/ARC_Challenge | Multi-choice QA | race/high |
| Closed-Book QA | ai2_arc/ARC_Easy | Multi-choice QA | race/middle |
| Closed-Book QA | kilt_tasks/hoptpotqa | Multi-choice QA | sciq |
| Closed-Book QA | trivia_qa/unfiltered | Multi-choice QA | social_i_qa |
| Closed-Book QA | web_questions | Multi-choice QA | super_glue/boolq |
| Closed-Book QA | wiki_qa | Multi-choice QA | super_glue/multirc |
| Extractive QA | adversarial_qa/dbidaf | Multi-choice QA | wiki_hop/original |
| Extractive QA | adversarial_qa/dbert | Multi-choice QA | wiqa |
| Extractive QA | adversarial_qa/droberta | Multi-choice QA | piqa |
| Extractive QA | duorc/SelfRC | Topic Classification | ag_news |
| Extractive QA | duorc/ParaphraseRC | Topic Classification | dbpedia_14 |
| Extractive QA | ropes | Topic Classification | trec |
| Extractive QA | squad_v2 | Word Sense Disambiguation | super_glue/wic |
| Extractive QA | super_glue/record | Dialogue State Tracking | multiwoz_2.1 |
| Extractive QA | quoref | Event Extraction | ace05 |
| Sentiment | amazon_polarity | Named Entity Recognition | conll03 |
| Sentiment | app_reviews | Named Entity Recognition | genia |
| Sentiment | imdb | Named Entity Recognition | ontonotes5.0 |
| Sentiment | rotten_tomatoes | Named Entity Recognition | ace2005 |
| Sentiment | yelp_review_full | Named Entity Recognition | conll04 |
| Sentence Completion | super_glue/copa | Named Entity Recognition | nyt29 |
| Sentence Completion | hellaswag | Relation Extraction | conll04 |
| Structure-to-Text | common_gen | Relation Extraction | nyt29 |
| Structure-to-Text | wiki_bio | Relation Extraction | ace2005 |
| Summarization | cnn_dailymail/3.0.0 | Relation Extraction | kelm |
| Summarization | gigaword | Relation Classification | tacred |
| Summarization | multi_news | Semantic Role Labeling | conll05 |
| Summarization | samsum | Semantic Role Labeling | conll12 |
| Summarization | xsum | Semantic Role Labeling | propbank |

Table 14: Details results of GLM-130B, GPT-3 175B (Brown et al., 2020), and PaLM 540B (Chowdhery et al., 2022) on BIG-bench-lite in 0, 1, and 3-shots. "Normalized preferred metric" is reported for each task. GPT-3 and PaLM's results are reported in BIG-bench's GitHub repository, and PaLM 540B's 3-shot results are not found.

| | GLM-130B | | | GPT-3 175B | | | PaLM 540B | |
|---|---|---|---|---|---|---|---|---|
| | 0 | 1 | 3 | 0 | 1 | 3 | 0 | 1 |
| auto_debugging | 11.76 | 20.59 | 23.53 | 0.00 | 0.00 | 0.00 | 0.00 | 38.23 |
| bbq_lite_json | 22.26 | 37.50 | 59.73 | -8.33 | 40.75 | 61.21 | -4.39 | 77.73 |
| code_line_description | 0.22 | 9.09 | -8.64 | 9.09 | 9.09 | 9.09 | 0.22 | 49.00 |
| conceptual_combinations | 37.51 | 31.33 | 27.86 | 2.37 | 3.70 | 14.33 | 45.68 | 73.36 |
| conlang_translation | 34.72 | 38.01 | 33.88 | 46.82 | 47.07 | 51.60 | 36.88 | 61.92 |
| emoji_movie | 1.25 | 4.88 | 3.75 | -10.00 | -2.49 | -1.24 | 17.50 | 88.75 |
| formal_fallacies_syllogisms_negation | 0.83 | 1.46 | 0.35 | 1.00 | 6.80 | 5.60 | -0.20 | 4.40 |
| hindu_knowledge | 32.23 | 37.56 | 34.52 | 10.15 | 40.61 | 44.42 | 41.37 | 93.15 |
| known_unknowns | -4.35 | 0.00 | 4.35 | 21.74 | 4.35 | 0.00 | 13.04 | 34.78 |
| language_identification | 9.62 | 1.97 | 1.90 | 7.49 | 3.20 | 1.98 | 12.11 | 31.03 |
| linguistics_puzzles | 0.00 | 0.00 | 0.00 | 0.00 | 0.00 | 0.00 | 0.00 | 0.10 |
| logic_grid_puzzle | 9.88 | 13.66 | 5.24 | 0.16 | 3.35 | 0.01 | 1.47 | 16.12 |
| logical_deduction | 24.18 | 22.20 | 20.35 | 2.22 | 10.80 | 14.71 | 2.17 | 15.34 |
| misconceptions_russian | -26.53 | -46.94 | -26.53 | -34.70 | -34.70 | -30.61 | -42.86 | -30.61 |
| novel_concepts | 6.25 | 21.87 | 25.78 | 33.59 | 33.59 | 45.31 | 33.59 | 49.22 |
| operators | 14.76 | 18.10 | 18.10 | 30.0 | 34.29 | 33.33 | 30.48 | 56.19 |
| parsinlu_reading_comprehension | 7.14 | 7.72 | 11.58 | 0.00 | 0.00 | 0.00 | 9.46 | 44.40 |
| play_dialog_same_or_different | 2.88 | 5.33 | 3.80 | 8.00 | 0.80 | -5.40 | -33.0 | 0.10 |
| repeat_copy_logic | 0.00 | 0.00 | 0.00 | 0.00 | 0.00 | 0.00 | 0.00 | 37.5 |
| strange_stories | 43.86 | 51.76 | 42.31 | 8.27 | 25.68 | 12.93 | 39.25 | 74.46 |
| strategyqa | 21.10 | 18.74 | 16.82 | 4.60 | 13.20 | 14.20 | 28.00 | 38.00 |
| symbol_interpretation | 1.39 | 1.89 | 1.77 | 0.51 | -0.63 | 2.77 | 0.76 | 2.40 |
| vitaminc_fact_verification | 71.87 | 60.72 | 56.55 | -31.55 | 22.15 | 29.05 | -28.85 | 55.60 |
| winowhy | -3.49 | 5.38 | 3.0 | 3.0 | 10.60 | 13.00 | -5.0 | 31.80 |

Table 15: Detailed results of GLM-130B and BLOOM 176B (Scao et al., 2022) on MMLU (Hendrycks et al., 2021). We find that no existing literature has reported GPT-3 175B's numerical accuracy. BLOOM is evaluated using Huggingface Transformer implementation.

|  | Discipline | GLM-130B | BLOOM 176B |
|---|---|---|---|
| STEM | abstract_algebra | 24.00 | 24.00 |
|  | anatomy | 48.90 | 38.52 |
|  | astronomy | 48.03 | 34.87 |
|  | colledge_biology | 47.22 | 37.50 |
|  | college_chemistry | 34.00 | 19.00 |
|  | colledge_computer_science | 44.00 | 1.00 |
|  | colledge_mathematcis | 27.00 | 31.00 |
|  | colledge_physics | 30.39 | 24.50 |
|  | computer_security | 61.00 | 40.00 |
|  | conceptual_physics | 38.72 | 31.49 |
|  | electrical_engineering | 45.52 | 32.41 |
|  | elementary_mathematics | 31.75 | 29.63 |
|  | high_school_biology | 51.29 | 27.42 |
|  | high_school_chemistry | 34.98 | 27.09 |
|  | high_school_computer_science | 53.00 | 30.00 |
|  | high_school_mathematics | 28.15 | 25.93 |
|  | high_school_physics | 29.80 | 30.46 |
|  | high_school_statistics | 38.43 | 26.39 |
|  | machine_learning | 40.18 | 29.46 |
| Social Science | econometrics | 26.32 | 26.32 |
|  | high_school_geography | 53.54 | 36.36 |
|  | high_school_government_and_politics | 62.18 | 40.41 |
|  | high_school_macroeconomics | 42.56 | 30.77 |
|  | high_school_microeconomics | 45.80 | 26.89 |
|  | high_school_psychology | 54.13 | 39.27 |
|  | human_sexuality | 51.15 | 35.11 |
|  | professional_psychology | 42.48 | 31.54 |
|  | public_relations | 55.46 | 33.64 |
|  | security_studies | 44.90 | 34.29 |
|  | sociology | 51.74 | 31.84 |
|  | us_foreign_policy | 61.00 | 46.00 |
| Humanities | formal_logic | 27.78 | 23.02 |
|  | high_school_european_history | 58.18 | 35.76 |
|  | high_school_us_history | 58.33 | 40.69 |
|  | high_school_world_history | 67.09 | 32.07 |
|  | international_law | 56.20 | 42.15 |
|  | jurisprudence | 43.52 | 35.19 |
|  | logical_fallacies | 57.06 | 31.29 |
|  | moral_disputes | 47.11 | 36.71 |
|  | moral_scenarios | 24.25 | 24.36 |
|  | philosophy | 45.34 | 35.37 |
|  | prehistory | 50.93 | 40.43 |
|  | professional_law | 37.94 | 29.53 |
|  | world_religions | 55.56 | 42.11 |
| Other | business_ethics | 51.00 | 34.00 |
|  | clinical_knowledge | 48.68 | 35.85 |
|  | colledge_medicine | 43.35 | 28.90 |
|  | glocal_facts | 35.00 | 23.00 |
|  | human_aging | 45.29 | 32.29 |
|  | management | 56.31 | 27.18 |
|  | marketing | 67.52 | 39.74 |
|  | medical_genetics | 48.00 | 45.00 |
|  | miscellaneous | 61.18 | 40.23 |
|  | nutrition | 50.65 | 32.35 |
|  | professional_accounting | 35.46 | 28.72 |
|  | professional_medicine | 43.38 | 18.01 |
|  | virology | 39.16 | 28.31 |

# E CONTRIBUTIONS

The GLM-130B project was conceived in Dec. 2021 with its pre-training part completed in July 3rd, 2022 and its evaluation and applications still ongoing. Over the course, we have experienced various technical and engineering challenges (Cf. Appendix F and Figure 21 for details). It would not be possible to reach its current status if without the collaboration of multiple teams—the Knowledge Engineering Group (KEG), Parallel Architecture & Compiler technology of Mobile, Accelerated, and Networked systems Group (PACMAN), and Natural Language Processing Group (THUNLP) at Tsinghua University, as well as Zhipu.AI. The detailed contributions are listed below.

## E.1 PREPARATION

- **Model Implementation:** Aohan Zeng, Zhengxiao Du
- **Self-Supervised Data Processing:** Ming Ding, Wendi Zheng
- **Multitask Data Processing:** Xiao Liu, Xiao Xia
- **Model Architecture:** Aohan Zeng, Xiao Liu, Zhengxiao Du, Hanyu Lai
- **Training Stability:** Aohan Zeng, Xiao Liu, Ming Ding
- **3D-Parallelism and Training Efficiency:** Aohan Zeng, Zixuan Ma, Jiaao He, Zhenbo Sun

## E.2 MODEL TRAINING

- **Large-Scale Training & Monitoring:** Aohan Zeng, Xiao Liu
- **Model Performance Validation:** Aohan Zeng

## E.3 POST TRAINING

- **Evaluation Framework:** Aohan Zeng, Zhengxiao Du
- **Language Modeling Evaluation:** Aohan Zeng
- **MMLU & BIG-Bench Evaluation:** Aohan Zeng
- **CLUE & FewCLUE Evaluation:** Xiao Liu, Aohan Zeng
- **Ethical Evaluation:** Yifan Xu, Aohan Zeng, Xiao Liu, Zihan Wang
- **Baseline Evaluation:** Xiao Liu, Jifan Yu, Weng Lam Tam
- **INT4 Quantization:** Aohan Zeng, Zihan Wang, Xiao Liu, Hanyu Lai
- **Inference Acceleration:** Zihan Wang, Aohan Zeng
- **Low-Resource Inference:** Gouyang Zeng, Xu Han, Weilin Zhao, Zhiyuan Liu
- **Demo and API:** Hanyu Lai, Jifan Yu, Xiaohan Zhang, Yufei Xue, Shan Wang, Jiecai Shan, Haohan Jiang, Zhengang Guo
- **Manuscript Writing:** Xiao Liu, Yuxiao Dong, and Jie Tang wrote the main paper, and Xiao Liu, Aohan Zeng, and Zhengxiao Du wrote the Appendix.

## E.4 PROJECT MANAGEMENT

- **Student Leaders:** Aohan Zeng, Xiao Liu
- **Technical Advisors:** Yuxiao Dong, Jidong Zhai, Wenguang Chen, Zhiyuan Liu, Peng Zhang, Jie Tang
- **Project Leader:** Jie Tang

## E.5 COMPUTATION SPONSOR

- **GPU Sponsor:** Zhipu.AI

## F    A BRIEF HISTORY OF GLM-130B

The GLM-130B project[16] was conceived in Dec. 2021 in a brainstorming meeting at Tsinghua KEG. We firmly believe that it is of value to pre-train a highly accurate language model, in particular for both Chinese and English. Though GPT-3 (Brown et al., 2020) is the pioneer for this effort, it is not available to most people in the world. In addition, it supports English only. We therefore decide to initialize the project GLM-130B. Please note that the WuDao 1.75T model we built last year is a sparse model with 480 mixture-of-experts (MoE), rather than a dense one as GPT-3. Our goal then is to train a bilingual pre-trained dense model with high accuracy on downstream tasks, and to make it open to everyone in the world-anyone, anywhere can download it and use it on a single server with appropriate GPUs.

The ambitious project soon faced several important challenges:

- **Lack of computational resources**: No organization is willing to sponsor such a big project and freely make it public.
- **Lack of a robust pre-training algorithm**: Despite GPT-3's success on English corpus, it is unclear how to train a high-accurate bilingual model for both English and Chinese.
- **Lack of fast inference solutions**: Since the goal is to have the model public to everyone, we need to design fast inference solutions with low resource requirements to run the model.

For the pre-training algorithm, we finally chose GLM (Du et al., 2022) due to its high performance in practice. We eventually decided to train a GLM model of 130 billion parameters after several rounds of discussions and exploration, because such a size makes it possible to run the inference on a single A100 (40G * 8) server.

Our first attempt at training the model was in January 2022, shortly after we received a small sponsor of GPUs for test running. However, we soon realized that we had significantly underestimated the technical difficulties of pre-training a model at such a scale (>100B). It seems that pre-training a highly accurate 100B-scale model is quite different from training a 10B-scale one. Due to frequent random hardware failures, model gradients exploding, unexpected excessive memory usage in the algorithm, debug for the 3D pipeline in the new Megatron and DeepSpeed frameworks, inability to recover from optimizer states, blocked TCP responses between processes, and many many unexpected "bugs", the project was delayed for many times. The Tsinghua PACMAN team gave us a hand at this difficult time and together we successfully fixed most of the "bugs".

By March, we were still short on computational resources, but fortunately got a chance to try test runs on several other platforms, including Ascend 910, Hygon DCU, NVIDIA, and Sunway. The immediate challenge was for us to adapt our training code to these different platforms, as the underlying operators are quite different. Also, it introduced many new issues: the element-wise operators not supporting fast computation for large-dimension vectors, various issues that hindered convergence—the large gradient norms of input embeddings, native Post-LN, Pre-LN, and Sandwich-LN, dataloader state seeds, and computation precision choices in Softmax and Attention — as well as numerous mistakes we ourselves made. With tremendous help from all of our generous partners, we finally succeeded in making our pre-training algorithms runnable across all the platforms—frankly, a surprising achievement for this project. The timeline of GLM-130B in Figure 21 covers most of the issues we have encountered and addressed as of this writing.

On April 26th, we received a generous computing sponsorship from Zhipu.AI — an AI startup that aims to teach machines to think like humans. After another week of testing, we finally kicked off the training of the GLM-130B model on its 96 A100 (40G * 8) servers on May 6th. Additionally, Zhipu.AI also sent a team to help evaluate the pre-trained model and build a demonstration website.

The training period spanned two months, during which we began developing a toolkit to allow GLM-130B's inference in low-resource setting with swapping technique and quantization. Though it is already the most accessible model of its scale, together with our partner from Tsinghua NLP, we have been exploring the limit of popularized hardware platforms, which would truly make the 100B-scale model accessible to as many people as possible. To date, we managed to reach the INT4 weight quantization for GLM-130B. Importantly, the INT4 version of GLM-130B without post training

---

[16]This section is largely extracted and updated from the blog introduction of GLM-130B at http://keg.cs.tsinghua.edu.cn/glm-130b/ (Posted date: August 4, 2022).

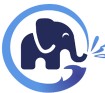 Major Issues Encountered for Training GLM-130B

**2021.12**
- The "千亿" (100B) project towards an open dense pre-trained GLM at 100B scale is conceived
- Survey pre-training strategies of existing models of similar scale, such as GPT-3, Gopher *=> Limited public info about how they were trained and issues they met*
- Search for possible GPU clusters & sponsors

**2022.1**
- Test the performance of FP16/FP32 at 100B scale on one testing cluster
- Unexpected excessive memory usage in GLM *=> Torch is better with fixed length input sequences*
- Inability to converge and try tricks from CogView and ViT *=> Use Sandwich-LN*
- Frequent random hardware failures *=> Have to run HCPG test before each run*

**2022.2**
- Very slow training speed than previously calculated *=> Optimize kernels and fuse operators  => Find the input shape is critical to kernel performance*
- Collect pre-training corpora and tokenize *=> Use icetk: the sentence piece is set to the unigram mode*
- Debug the 3D pipeline parallel in the newly-released Megatron and DeepSpeed

**2022.3**
- It can't recover perfectly from checkpoints *=> Our customized dataloader do not save its state seed properly in distributed training*
- The memory per processor is too small *=> Require too many pipeline stages => Batch size is too large (up to 12,000) => Harm the model's convergency*
- It can't launch more than 2,000 computing nodes *=> Overcome this and support 6,000-node training by tuning Linux kernel TCP parameters*
- Collect data for multi-task instruction pre-training
- Receive opportunities to test trainings on several other clusters
- Very slow training speed than expected *=> The underlying element-wise operators don't support fast computation on large-dimension vectors.*

**2022.4**
- Optimize A100 kernel's computing efficiency *=> A100 kernels prefer square-shaped inputs, and seq_len=2,048 is optimal for our hidden-state dimension (12,288)*
- Inability to converge due to large gradient norms (170+) of input embeddings *=> Try embedding norm and gradient shrink, which turn out to be almost equivalent*
- Naïve post-LN or pre-LN disconverges after several thousands of steps *=> Try Sandwich-LN with PB-Relax*
- It still disconverges after one week's trial *=> The dataloader state seeds are not unified for different pipeline stages, resulting in a mismatch of input data and labels.*
- Test two positional encodings: RoPE and Alibi  *=> Alibi can be slower as it requires element-wise manipulation on attention matrices---changing num_heads *2,048 * 2,048 scalars per layer*
- Test GeGLU and GAU  *=> GAU converges faster with relatively poor performance on fine-tuned SuperGLUE*
- Abnormal GPU memory usage of newly-added functions and classes *=> DeepSpeed hardcodes the function names for checkpoint activation*
- Decide to train GLM with 130 billion parameters *=> allow inference on a DGX-A100 40G node*

**2022.5-6**
- Implement a RoPE cuda operator in C++ *=> See unexpected precision errors and finally have it abandoned*
- Sandwich-LN still disconverges *=> 1) Reducing learning rate does not help; 2) Using Hinge cross-entropy becomes slower and harms performance; 3) Shifting to DeepNorm still disconverges*
- Use FP32 in softmax of attention *=> Success*
- Find PB-Relax unnecessary for FP32 softmax *=> It also slows down training as it needs to manipulate the whole attention score matrices*
- Experience few spikes in later training *=> 1) Reduce gradient shrink factor from 1 to 0.1: useful; 2) Reduce the learning rate: sometimes useful; 3) Jump the noisy data batches: sometimes useful*
- Find a mistake in multi-task data after training for 20,000 steps *=> Use the correct data but it does not forget*

**2022.6-7**
- Adapt the pipeline parallel checkpoints to ordinary parallel checkpoints for efficient inference on a single A100
- Work on evaluation scripts on datasets: MMLU, Big-bench, CLUE, SuperCLUE, etc.
- Implement P-Tuning and P-Tuning v2 for parameter-efficient tuning on GLM-130B for tuning on SuperGLUE
- Work with BMInf on adapting GLM-130B to perform inference on a single V100 or 3090 *=> Use pipeline-style asynchronous swapping between main memory and GPU memory*
- Try to fine-tune GLM-130B with fewer A100 nodes (i.e., 12-16 nodes) *=> Pipeline-style fails due to too many pipeline stages => Find that data parallel can not be introduced for fine-tuning => Use 32-way model parallel for fine-tuning with reasonable performance*

https://github.com/THUDM/GLM-130B

Figure 21: The timeline of major issues that training GLM-130B encountered and addressed, as of July 31st, 2022.

faces negligible performance degradation compared to its uncompressed original, while it consumes only 25% of the GPU memory required by the uncompressed version, thus supporting its effective inference on 4 × RTX 3090 Ti (24G) or 8 × RTX 2080 Ti (11G). We will attempt to further reduce the resource requirements and keep the community updated on this important working item.

# G    BROADER IMPACT

This paper introduces an open bilingual pre-trained language model with 130 billion parameters. Currently most pre-trained language models with over 100 billion parameters are privately owned by governments and large corporations (Brown et al., 2020; Thoppilan et al., 2022; Rae et al., 2021; Chowdhery et al., 2022; Wang et al., 2021). A few of them (Brown et al., 2020; Lieber et al., 2021) provide limited inference APIs with fees. In contrast, the weights and code of GLM-130B are open to anyone who is interested in LLMs. Moreover, we significantly lower the hardware requirements for inference by speed-up implementation and INT4 quantization. The paper can have a broader impact on the research community, individual developers and small companies, and society.

## G.1    IMPACT ON AI RESEARCH

Most research institutions cannot afford the substantial cost of pretraining large language models. As a result, most researchers, except employees of governments and large corporations, only have access to the limited inference APIs with fees. With the inference APIs, researchers can only analyze the outputs of models as black boxes, which limits the scope of potential work. With GLM-130B, researchers can analyze the model parameters and internal states corresponding to specific inputs, leading to in-depth studies of LLMs' theory, capacity, and flaws. Researchers can also modify the model architecture and weights, to validate the proposed algorithms to improve LLMs Zhu et al. (2020); Cao et al. (2021); Hase et al. (2021); Mitchell et al. (2022).

With INT4 quantization, GLM-130B can perform inference on popularized GPUs such as 4 × RTX 3090 or 8 × RTX 2080 Ti, which can be easily accessed from cloud service. As a result, researchers who cannot afford powerful data-center GPU servers like DGX-A100 can also utilize GLM-130B.

## G.2    IMPACT ON INDIVIDUAL DEVELOPERS AND SMALL COMPANIES

Currently, individual developers and small companies who want to integrate LLMs into their business can only choose paid inference APIs. The increased cost can hinder their attempts. Instead, GLM-130B can be deployed on popularized hardware that they own or can access via cloud service to reduce the cost. Furthermore, they can utilize distillation techniques Sanh et al. (2019); Jiao et al. (2020) to obtain smaller models that preserve comparable performance on their specific tasks. While some developers may lack the ability to complete deployment and distillation on their own, we believe with GLM-130B and more open LLMs in the future, the corresponding toolkits and service providers will become more available.

We also note that currently most applications of LLMs are based on prompt engineering, partly due to the limitation of inference APIs. In downstream scenarios such as online customer service, the companies accumulate huge amounts of human-generated data that contain domain knowledge. With the open-source weights and code, developers can finetune GLM-130B on their own data to mitigate the gap of domain knowledge.

## G.3    SOCIAL IMPACT

Large language models, together with other machine learning models in different modalities (e.g., Image (Ramesh et al., 2021; Ding et al., 2021; Saharia et al.) and Video (Hong et al., 2022)), could be used to generate synthetic text for harmful applications, such as telemarketing fraud, political propaganda, and personal harassment as is discussed in (Weidinger et al., 2021; Sheng et al., 2021; Dev et al., 2021). We do not anticipate any hazardous outputs, especially towards vulnerable and historically disadvantaged groups of people, after using the model.

While some people think that restricting access to LLMs can prevent such harmful applications, we argue that promoting LLM inclusivity can lead to better defense against potential harm caused by

LLMs. Currently, only governments and large corporations can afford the considerable costs of pre-training LLMs. There is no guarantee that organizations having the substantial financial resources to pretrain an LLM will not do harm with it. Without access to such LLMs, individuals cannot even realize the role of LLMs in harm. Conversely, releasing an open LLM can provide access and transparency to all the researchers and promote the research to reduce the potential harm of LLMs, like algorithms to identify the synthetic text Gehrmann et al. (2019) or detect fake news Li et al. (2021).

Also, it is known that LLMs can suffer from problems in fairness, bias, privacy, and truthfulness Zhang et al. (2021); Lin et al. (2022); Liang et al. (2021); Bender et al. (2021). An open LLM can reveal the model parameters and internal states corresponding to specific inputs instead of providing APIs to black-box models. In conclusion, researchers can conduct analysis of LLMs' flaws in depth and propose improved algorithms to solve the problems.

## H  ENVIRONMENTAL IMPACT

One of the major concerns about large language models is their huge energy usage and associated carbon emissions Strubell et al. (2019); Lacoste et al. (2019); Patterson et al. (2021); Bender et al. (2021). GPT-3 was estimated to use 500 tons of carbon emissions footprint (CO2eq) Patterson et al. (2021). We consumed a total of 442.4MWh of electricity over the 60-day course of training. Given the 0.5810 kg/kWh carbon efficiency of local power grid, the pre-training released 257.01 metric tons of $CO_2$. This is around half of GPT-3's carbon footprint, probably due to the efficient parallel strategies and NVIDIA's hardware improvements. The carbon emission is roughly the equivalent of the yearly emissions of 18 average Americans. However, we believe that with GLM-130B released, more carbon emissions for reproducing 100B-scale LLMs can be saved.

