# OpenReview forum: "GLM-130B: An Open Bilingual Pre-trained Model"
_ICLR.cc/2023/Conference — ICLR 2023 poster_

### Official Review · Reviewer_asnf · 2022-10-26

**Confidence:** 4
**Correctness:** 3
**Technical Novelty And Significance:** 3
**Empirical Novelty And Significance:** 4
**Recommendation:** 8

**Clarity, Quality, Novelty And Reproducibility:**

- The paper is well-written and clearly presented;
- The model is open-sourced;
- The design choice and detailed engineering efforts
- More analysis behind the surging performance will be very valuable to add, for example:
    - Any emergent abilities of the model (in terms of scale and in terms of the amount of data) as shown in [3].
    - Why the Bilingual model outperforms the Uni-lingual model by a large margin.
    - Few-shot performance of quantized GLM-130B;


**Strength And Weaknesses:**

GLM-130B: AN OPEN BILINGUAL PRE-TRAINED MODEL
*Paper Summary
This paper introduces and open-sources one of the largest 100B bilingual pre-trained language model GLM-130B. It covers a detailed pre-training process (training strategies, design choices) and the resulting model outperforms several large language models on English and Chinese zero-shot understanding benchmarks. The author also investigates the efficient inference of the GLM and show the efficacy of using Int4 quantization.

Summary Of Strengths
- The paper is well-written and clearly presented;
- It detailed the efforts made during the pre-training process for the GLM-130B, making it very valuable for practitioners in the large language model pre-training area; (though several design choices might not be feasible to ablate at that scale.).
- The resulting model was tested across benchmarks, and showed surprising robustness towards quantization, making it more accessible for academia.


Summary Of Weaknesses
- Why the bi-lingual GLM performs a lot better in LAMBADA as shown in Figure 2? Is this also true for MMLU, Pile, and NLG tasks, any intuition about this?
- Why not compare to OPT-175B models for quantization?
- Intuition why the few-shot performance of GLM-130B in Table 4. falls behind other methods? Also, will the quantized GLM-130B yield similar few-shot performance?
- Which GPT-3 175B is the GLM-130M compared to? Is it a fairer comparison to the InstructGPT variants?
- How will the author attribute the performance gain (Bi-directional modeling [1], Multi-Task Instruction Pre-Training [2], or the Bilingual corpus and model architecture)?
- It is not surprising that the Bi-directional model will outperform the Uni-directional model in understanding tasks. How are the generation tasks performance of GLM-130B compared to uni-directional counterparts?


[1] Tay, Yi, et al. "Transcending Scaling Laws with 0.1% Extra Compute." arXiv preprint arXiv:2210.11399 (2022).

[2] Chung, Hyung Won, et al. "Scaling Instruction-Finetuned Language Models." arXiv preprint arXiv:2210.11416 (2022).

**Summary Of The Paper:**

This paper introduces and open-sources one of the largest 100B bilingual pre-trained language model GLM-130B. It covers a detailed pre-training process (training strategies, design choices) and the resulting model outperforms several large language models on English and Chinese zero-shot understanding benchmarks. The author also investigates the efficient inference of the GLM and show the efficacy of using Int4 quantization.


**Summary Of The Review:**

It is not surprising that a large-scale bi-directional language model will outperform the causal counterpart on a range of zero-shot language understanding tasks. The detailed engineering efforts and design choices make this paper valuable, though my main concerns are what factors make the performant GLM-130B unclear, as well as NLG / few-shot performance.

---

> ### Author Response · Authors · 2022-11-19
> **Response to Reviewer asnf**
>
> Thank you very much for your appreciation and great suggestions for our work! Below please find the responses to the issues and questions raised.
>
> ### Reason for GLM-130B’s Outperformance & Contribution Attribution
>
> Figure 2 suggests that the outperformance is mostly from the bidirectional attention (i.e., the GLM objective). It has been proved much better than unidirectional LMs in language modeling datasets in literature [1, 2]. In addition, the Multi-task Instruction Pre-training (MIP) may contribute little improvement as well. To date, **the bilinguality is not observed to account for the outperformance.**
>
> We retrain and ablate the bidirectional attention and MIP at 10B scale GLM (~100B is just too expensive) and test them on LAMBADA. The results show that bidirectional attention contributes 73% of the performance improvement compared to unidirectional counterpart. MIP contributes the rest 27%. Please see Appendix B.9 Figure 14 (https://i.imgur.com/PRlVmtL.png) for more details.
>
> We also evaluate the bidirectional attention’s effect in Conditional NLG (Cf. Table 16) and SuperGLUE (Cf. Figure 17). These demonstrate that bidirectional attention is a crucial component for the competitiveness of GLM-130B.
>
>
> ### Unavailability of OPT-175B Checkpoint
>
> It is a pity that Meta has not officially granted us the access to OPT-175B, despite our three applications in the past half a year (haven’t received any feedback yet), as is described by the footnote on Page 7 of our original paper. The results of OPT in our paper are all taken from existing literature and materials [1, 2, 3]. That is the reason for why we cannot compare OPT-175B in the quantization experiment.
>
>
> ### Intuition on the Few-Shot Performance
>
> This is a great question. Intuitively there are some possible explanation and recent discoveries:
>
> 1. MIP training narrows the gap between zero-shot and few-shot learning. Previous literature on instruction tuning (Figure 9 and Section 4.4 in FLAN [4]) shows that, despite the additional gain from 0-shot to few-shot after instruction tuning, it can be much more insignificant than that from vanilla GPT-3. This is in line with our observation.
> 2. Bidirectional attention may need some new techniques to apply in-context learning. The existing few-shot learning for LLMs are based on in-context learning, for which some researchers theoretically argue that derives from the low intra-sample transition probability from unidirectional GPT objective [5]. Obviously, bidirectional models do not satisfy the property well, and some concurrent work also notice the problem in practice [6].
>
> Nevertheless, GLM-130B still performs well on many in-context few-shot learning missions (e.g., MMLU, Conditional NLG, and etc.). But we think there might be some more space to further boost its few-shot learning performance.
>
>
> ### The Compared GPT-3 Version
>
> We compare the original GPT-3 175B (Davinci) in our paper, and we think it is not fair to compare GLM-130B to InstructGPT. The core technique leveraged in InstructGPT is the Reinforcement Learning from Human Feedback (RLHF), in which GPT-3 175B involves two post-training stages: 1) Supervised Fine-tuning (SFT) stage and 2) RLHF stage. SFT is almost identical to FLAN [1] and T0 [2], whose inputs are from downstream NLP tasks (**90%** as is reported in their paper). In RLHF, they continually leverage a reward model trained on massive human feedback to steer GPT-3 for better human intention understanding. As GLM-130B only involves **5%** of MIP tokens in the pre-training and does not leverage any RLHF yet, it is unfair to compare its current version to InstructGPT.
>
>
>
> [1] Zhang, Susan, et al. "Opt: Open pre-trained transformer language models." arXiv preprint arXiv:2205.01068 (2022).
>
> [2] EAI-Eval BigScience Arch&Scale - Google Sheet: [https://docs.google.com/spreadsheets/d/1CI8Q9RCblLRzUOPJ6ViqBmo284-8ojluQ-CmaEuhuv0/edit#gid=1295801165](https://docs.google.com/spreadsheets/d/1CI8Q9RCblLRzUOPJ6ViqBmo284-8ojluQ-CmaEuhuv0/edit#gid=1295801165)
>
> [3] BigScience evaluation results repository in Huggingface Datasets: [https://huggingface.co/datasets/bigscience/evaluation-results/tree/main/bloom/bloomzeval/transformers/evaluation_val](https://huggingface.co/datasets/bigscience/evaluation-results/tree/main/bloom/bloomzeval/transformers/evaluation_val)
>
> [4] Wei, Jason, et al. "Finetuned Language Models are Zero-Shot Learners." International Conference on Learning Representations. 2022.
>
> [5] Xie, Sang Michael, et al. "An Explanation of In-context Learning as Implicit Bayesian Inference." International Conference on Learning Representations. 2022.
>
> [6] Bidirectional Language Models Are Also Few-shot Learners. ICLR 2023 Anonymous Submission.

---

> > ### Author Response · Authors · 2022-12-02
> > **Thanks for your great review!**
> >
> > Dear Reviewer,
> >
> > Thank you very much again for your kind and valuable reviews! We hope our previous response addresses your questions, from which this work has benefited a lot. If possible, we would be deeply grateful if you could help share suggestions to further make GLM-130B a stronger work!
> >
> > Thank you again for your encouraging comments!
> >
> > Authors of Paper3642

---

### Official Review · Reviewer_XkYq · 2022-10-27

**Confidence:** 4
**Correctness:** 3
**Technical Novelty And Significance:** 3
**Empirical Novelty And Significance:** 3
**Recommendation:** 8

**Clarity, Quality, Novelty And Reproducibility:**

Clarity:
Paper is clearly written and easy to follow.

Quality:
The work is high quality and various aspects of building the model are studied well.

Novelty:
One could argue that the paper lacks some novelty as they do not comprehensively produce new techniques that have been ablated at a smaller scale that have shown to produce improvements. But I understand that this is hard to do at such scale since they only get one shot at training the model. Smaller scale experiments with some of the new techniques could have helped here.

Reproducibility:
Overall they take good care to make the model, training log, evaluation sets are reproducible.

**Strength And Weaknesses:**

Strengths:
- The model is more reproducible than other similar sized counterparts.
- The model's performance is competitive on the benchmarks that are evaluated on and outperforms large models like GPT-3 and PaLM.
- Quantization for inference and mitigation strategies like deepnorm and embedding shrinkage for training instability are useful for the community.

Weaknesses:
-  I would have liked to see more English tasks evaluated on. The current set of results on the LaMBADA dataset a lot. I would like to see results on more GPT-3 tasks or SuperGLUE tasks for English.
- More results at different scales and projection of trends or emerging capabilities as scale increases would have been interesting. The current set of results are missing this.

**Summary Of The Paper:**

This paper introduces GLM-130B, a bilingual English and Chinese LLM. The model is trained for 400B tokens (half on English  and half on Chinese). The model follows a bidirectional attention architecture with the self-supervised blank infilling objective. They also include a small percentage of MIP tokens. Int4 quantization enables inference on a diverse set of platforms. Various model instabilities and mitigation strategies are discussed.

The model, training log and inference through free APIs are provided and an effort is made to make the entire work reproducible.

Detailed comments:
Table 4: You state that the reason for PaLM's 1-shot ability being so much better than GLM is the diverse dataset. Could you elaborate a bit more?
Section 6, subsection name "Transferring" --> consider using a better subsection name.

**Summary Of The Review:**

Overall, the paper is clearly written and details the building of the GLM-130B bilingual LLM. The model is competitive with other similar sized models and care has been taken to make the model set up reproducible. The evaluation suite for English could have been more comprehensive to substantiate the claims a bit more.  I would have also liked to see more experiments at a smaller scale and studies of emerging capabilities of the model at scale. Since this is a bilingual model, studying how much transfer or interference happens between these two languages would have been interesting.

---

> ### Author Response · Authors · 2022-11-19
> **Response to Reviewer XkYq (1/2)**
>
> Thank you very much for your appreciation and great advice for our work! We have been performing various new experiments during the past two weeks to cover more tasks and results as suggested. Here are the responses to issues raised:
>
> ### More Comprehensive Evaluation
>
> In our initial submission, we reported our evaluation of GLM-130B over **112 tasks in total**, including not only LAMBADA but also many others such as MMLU, BIG-bench-lite, Pile, CLUE & FewCLUE, and etc., covering a broad range of NLP challenges. We originally refrained from reporting many GPT-3 tasks since they have been included in our Multi-task Instruction Pre-training (MIP, 74 datasets) and we thought the report of them might cause confusion.
>
> To further evaluate GLM-130B’s quality, we additionally implemented GLM-130B for comparisons over **40+ datasets** during the rebuttal to date, including challenges on **Winograd-style tasks, CBQA, Commonsense reasoning, Chain-of-thought prompting, Conditional Text Generation, Natural Language Inference,** and **SuperGLUE**. If any other important types of tasks are missed, we will add their results in the future versions, as the goal of this work is to pre-train and evaluate an open and highly-accurate 100B-scale LLM.
>
> **Commonsense Reasoning**
>
> We implement GLM-130B and related comparison LLMs on CommonsenseQA and MC-TACO. Please see Appendix C.11 and Table 19 for details.
>
> |                      | shots    | GPT-3 (Davinci) | OPT 175B | BLOOM 176B | GLM-130B |
> | -------------------- | ---- | --------------- | -------- | ---------- | -------- |
> | Commonsense QA (Acc) | 0    | 57.2            | -        | 42.8       | 61.6     |
> | Commonsense QA (Acc) | 1    | 61.2            | -        | -          | 62.2     |
> | MC-TACO (EM)         | 0    | -               | 12.4     | 13.1       | 13.6     |
>
> **Chain-of-Thought Prompting**
>
> We implement GLM-130B on some typical reasoning datasets, including Sports Understanding, Last Letter Concatenation (LLC), Coin Flip, Coin Flip (OOD: 3), Reverse List, and Date Understanding. Due to the 2-week limit, we have not spared to test on other datasets, and we will report them in the future version. Please see Appendix C.14 and Figure 18 (https://i.imgur.com/4bZf3Sx.png) for details.
>
> **Winograd-style Tasks**
>
> We implement GLM-130B and related comparison LLMs on Winograd273 and Winogender. Since Winogrande is unfortunately included in our MIP training, we cannot evaluate it. Please see Appendix C.9 and Table 17 for details.
>
> |             | shots    | GPT-3 (Davinci) | OPT 175B | BLOOM 176B | PaLM 540B | Chinchilla | Gopher 280B | GLM-130B |
> | ----------- | ---- | --------------- | -------- | ---------- | --------- | ---------- | ----------- | -------- |
> | Winogender  | 0    | 64.2            | 54.8     | 49.1       | 75.0*     | 78.3       | 71.4        | 79.7     |
> | Winogender  | 1    | 62.6            | -        | 53.1       | 79.4      | -          | -           | 80.7     |
> | Winograd273 | 0    | 88.3            | 52.9     | 49.1       | 90.1      | -          | -           | 84.3     |
>
> *PaLM 540B did not report the exact 0-shot Winogender result, so we have to estimate a value from its plotted diagram
>
> **Closed-book Question Answering**
>
> We implement GLM-130B and related comparison LLMs on Natural Questions and StrategyQA. Since TriviaQA and WebQuestions used in GPT-3 are unfortunately included in our MIP training, we cannot evaluate them in the zero-shot manner. Please see Appendix C.10 and Table 18 for details.
>
> |                       | GPT-3(Davinci) | BLOOM 176B | PaLM 540B | Chinchilla | Gopher 280B | GLM-130B |
> | --------------------- | -------------- | ---------- | --------- | ---------- | ----------- | -------- |
> | Natural Questions(EM) | 14.6           | 13.1       | 21.2      | 16.6       | 10.1        | 11.7     |
> | StrategyQA(Acc)       | 52.3           | 49.8       | 64.0      | -          | -           | 60.6     |
>
> **SuperGLUE**
>
> Since almost all of SuperGLUE datasets have been included in the MIP training (except ReCoRD), the results of GLM-130B on SuperGLUE are not for comparison to others. We mainly compare between the performance of unidirectional and bidirectional GLM-130B on SuperGLUE as an ablation. Please see Appendix C.13 and Figure 17 for details.
>
> **Conditional Text Generation**
>
> We implement GLM-130B on 3 typical conditional text generation tasks, including WebNLG, E2E (Data-to-text) and WikiLingua (Summarization) following the practice in PaLM-540B. Please see Appendix C.8 and Table 16 (https://i.imgur.com/GVfqZqJ.png) for results. We do not include Xsum since it is used in the MIP training.

---

> > ### Author Response · Authors · 2022-11-19
> > **Response to Reviewer XkYq (2/2)**
> >
> > ### Scaling Property and Emergent Abilities in GLM-130B
> >
> > We also conducted extensive experiments to explore the scaling property and emergent abilities in GLM-130B. Following prior literature, we categorize the NLP tasks into two types based on our observations (Cf. Appendix D and Figure 19, 20 for details):
> >
> > * Log-scaling Ability Tasks (https://i.imgur.com/rPxpLoy.png): where the task performance grows logarithmically with the amount of model parameters. Typical tasks and datasets include: LAMBADA, Wikitext-103, Wikitext-2, Penn Tree Bank.
> > * Emergent Ability Tasks (https://i.imgur.com/2krQgeR.png): where the task performance only soars up when the amount of model parameters reaches a certain threshold. Typical tasks and datasets include: MMLU, hindu_knowledge, crass_ai, implicatures, understanding_fables, modified_arithmetic, implicit_relations, and gre_reading_comprehension.
> >
> > We notice that emergent abilities usually arise in long-tailed knowledge tasks and often require LLMs’ ability to memorize some rare patterns.
> >
> > ### Detailed Discussion on PaLM's Superior Performance
> > In the paper we mention the PaLM's more diverse corpus. To be specific, while the Pile dataset is around 330B tokens, PaLM's corpus is around 780B tokens as they reported, more than 2 times larger than Pile. Additionally, they also report they train the 540B PaLM for around 800B tokens, which is estimated to consume more than 16 times of computation in English than that of GLM-130B. Both of these factors could significantly improve its performance.

---

> > > ### Author Response · Authors · 2022-12-02
> > > **Thanks for your great review!**
> > >
> > > Dear Reviewer,
> > >
> > > Thank you very much again for your kind and valuable reviews! We hope our previous response addresses your questions, from which this work has benefited a lot. If possible, we would be deeply grateful if you could help share suggestions to further make GLM-130B a stronger work!
> > >
> > > Thank you again for your encouraging comments!
> > >
> > > Authors of Paper3642

---

### Official Review · Reviewer_tHby · 2022-10-30

**Confidence:** 5
**Correctness:** 3
**Technical Novelty And Significance:** 3
**Empirical Novelty And Significance:** 3
**Recommendation:** 8

**Clarity, Quality, Novelty And Reproducibility:**

GLM-130B has very clear architecture and training details, relative to some other LLM papers. While it also has relatively few truly novel architectural or training components, this is reasonable as large-scale work isn't the time to introduce those. It also has unprecedented reproducibility for an LLM training paper, due to releasing just about every important component in open source.

What appear to be the two key advances relative to other contemporary LLMs (bidirectional attention and instruction pretraining) are concurrent with other very recent work (e.g. UL2 and Flan from Google) so it could be helpful to contextualize with those.

**Strength And Weaknesses:**

Strengths:
- Very careful attention to training stability
- Comprehensive details
- Accessibility to smaller scale researchers
- Open source
- General openness even when admitting something could have been done better (e.g. "We later found the instructions to implement two-dimensional RoPE from its author’s blog https://kexue.fm/archives/8397, but our training has proceeded for weeks.")

Weaknesses:
- The work chooses optimization hparams like initial LR and LR schedule relatively unsystematically.
- They use dropout without explaining why, even though there's no repeated data in the dataset
- The precision discussion mentions avoiding bf16 because of fp32 gradient accumulation, but fp16 or even bf16 accumulation is typically fine in practice.
- There isn't really an explanation of why GLM-130 has narrower weight distribution than BLOOM (my guess is it's related to DeepNorm)?
- The paper doesn't explicitly state that the Pile test set was held out of the training set, but I'm assuming it was.
- Figure 7 leaves out PaLM 1- and 5-shot BBL, which are much better than 0-shot, and Chinchilla (which was only evaluated at 5-shot). Those are reasonable choices, but I'd prefer to err on the side of including more data points. Table 4 also leaves out PaLM and Chinchilla 5-shot.
- no explanation of how 135 TFLOPS was computed (in particular, it probably includes remat but this isn't stated). Would recommend adopting MFU/HFU from PaLM paper.

**Summary Of The Paper:**

It's always hard to summarize papers that report on such a big project, but fundamentally this is a tech report of training an English+Chinese LLM with techniques that improve on the state of the art for pretraining methodology and training stability.

GLM-130 is concurrent with OPT and BLOOM, two other GPT-3-class LLMs that were released openly this year. It has d_model=12288 (same as GPT-3, OPT, and BLOOM) and 130B parameters and was trained on 200B tokens each of English and Chinese, including English NLU tasks formatted for multitask instruction pretraining to increase zero-shot performance on other, unseen English and Chinese tasks. It used a GLM objective that mixes span-by-span span-filling and prefix-LM, enabling use of bidirectional attention at inference time to further improve zero- and few-shot performance.

It consistently outperforms GPT-3, OPT, and BLOOM on English despite using fewer parameters and less compute (also less English data than GPT-3/OPT), and it sometimes also outperforms the much larger PaLM. It also consistently outperforms ERNIE TITAN 3 on Chinese, despite fewer parameters, less compute, and less Chinese data. As it can run in both bidirectional and unidirectional mode at inference time, ablating this shows that bidirectional attention contributes greatly to its outperformance. It also exhibits less bias/toxicity than other LLMs.

In terms of optimization methodology, it uses relatively standard techniques including AdamW with LR warmup and cosine decay, and also uses early batch size warmup to a relatively large 8.65M token total batch size. For training stability, it uses post-LN with depth-based initialization ("DeepNorm"), downscaled gradient in embeddings ("Embedding Gradient Shrink"), gradient clipping, and dropout. Either the DeepNorm or something else about the model makes the weight distribution especially narrow, enabling near-lossless int4 weight quantization for small-footprint inference.

In terms of training efficiency, it achieves about 32.5% MFU at steady state [135e12/312e12 * 3/4 for remat] and 25.1% MFU end to end [(130e9*400e9*6)/(96*8*312e12*60*60*24*60)] using 8 pipeline stages with a PipeDream-Flush (aka 1F1B) schedule, 4-way tensor parallelism, and 24-way (2-way intranode and 12-way internode) data parallelism. The authors also provide an efficient FasterTransformer-based inference setup that supports consumer GPUs.

**Summary Of The Review:**

I think this really represents the gold standard in reproducibility for LLM publications, as well as the most capable open-source English or Chinese language model in existence.

---

> ### Author Response · Authors · 2022-11-19
> **Response to Reviewer tHby (1/2)**
>
> We deeply appreciate your acknowledgement of this effort and more importantly your insightful suggestions to further improve this work! Here are the responses to your questions, and it would be great if you could further share your thoughts on these issues.
>
> ### Hyper-Parameter Choices
>
> We agree that there exist systematical ways to achieve this. Though we have tried a large number of hyper-parameter combinations, including learning rate, batch size, and warmup ratio, etc., to conquer the training instability challenge, it is unfortunate that we are not allowed to conduct a more systematic ablation study on LLMs due to the high computing cost per attempt. Our final hyper-parameter choices are mainly derived from the above experiments to ensure stable training.
>
> To better illustrate this, we further append all our preliminary trial logs to the anonymous repo (https://anonymous.4open.science/r/GLM-130B/), revealing our attempts at hyper-parameter tuning and architectural selection. Hopefully these will help give more empirical insights for future LLM training.
>
> Specifically, as for the LR schedule, since it’s possible to estimate the total number of training steps given our computation budget, we choose cosine decay instead of the inverse square as we considered a well-tuned cosine decay schedule will result in better performance. Guided by the finding in Chinchilla [1], we make the cosine cycle length match the number of training steps to expect better performance.
>
> As for the dropout, considering that most existing LLMs are under-trained, we very much agree that there is no need to use dropout on a well-preprocessed dataset. We have to admit that we actually just followed the dropout settings used to train smaller models. Thanks for pointing this out!
>
>
> ### Floating-Point Format
>
> Floating point format is one of the important choices for efficient LLM training. Although BF16 has been proven to be more stable than FP16 and is widely used in LLM training [2] [3], at the time, we decided to prioritize the stability of FP16 training in order to enable GLM-130B's inference on (relatively) older GPU architectures (e.g., NVIDIA Tesla V100).
>
> For our experiments with BF16 format, since the pipeline parallelism divides the input into micro-batches, we found that the FP32 gradient accumulation is needed for smooth training due to BF16’s round-off error, which is consistent with BLOOM [2]. The ~15% memory overhead is mainly from the non-sharded gradient accumulator allocated in DeepSpeed’s BF16 optimizer, which is unacceptable as we are on tight memory on 40G A100s. This memory overhead can be reduced with using a sharded approach at the cost of training throughput, though we did not implement this considering our schedule.
>
> Fortunately, GLM-130B’s FP16 training is stabilized using DeepNorm and EGS, we also agree FP16 gradient accumulation is sufficient in this case and we indeed use it in final training.
>
> [1] Hoffmann, Jordan, et al. "Training Compute-Optimal Large Language Models." arXiv preprint arXiv:2203.15556 (2022).
>
> [2] Scao, Teven Le, et al. "BLOOM: A 176B-Parameter Open-Access Multilingual Language Model." arXiv preprint arXiv:2211.05100 (2022).
>
> [3] Chowdhery, Aakanksha, et al. "Palm: Scaling language modeling with pathways." arXiv preprint arXiv:2204.02311 (2022).

---

> > ### Author Response · Authors · 2022-11-19
> > **Response to Reviewer tHby (2/2)**
> >
> > ### Training Efficiency
> >
> > We have updated the description on training efficiency (Cf. Sec 2.3). We did use activation recomputation (aka rematerialization), resulting in the final 43.3% HFU and 32.5% MFU. It’s possible to further improve the MFU by leveraging sequence parallelism and selective activation recomputation proposed by recent work [4], and we will continue to improve training efficiency for our future training.
> >
> > ### Weight Distribution
> >
> > It’s a very interesting point, narrower weight distribution leads to less quantization error,  which enables memory-efficient inference for LLMs. To figure out whether LN or EGS contributes to the weight distribution, we further trained four GLM variants with 110M parameters, using Pre-LN, Post-LN, DeepNorm, and DeepNorm (EGS, alpha = 0.1), respectively, with other settings remaining the same as GLM-130B.
> >
> > From the figure (https://i.imgur.com/XerQfJE.png), we can see that PostLN has a slightly narrower weight distribution in general, and there is no significant difference between DeepNorm and PreLN, training with EGS almost produces the same distribution. It seems neither DeepNorm nor EGS may be the key factor, at least on the 100M-scale. We suspect that the narrower weight distribution of GLM comes from the denoising part of the objective function, as some recent work [5] points out that one of the main difficulties to quantize a generative language model is the varied distribution of weights, while there are various methods to compress BERT or its variants. Due to the time limit, we have not spared performing more experiments to examine this during this rebuttal but we plan to prioritize this in our ongoing work for a systematic analysis.
> >
> > ### Training data
> >
> > We used the official weighted train split of the Pile data set, which duplicates several datasets and thus increases the raw size 825GB to effective size 1.2T. We have updated the description of the dataset split (Cf. Sec 2.2).
> >
> > ### Big-bench-lite Diagramming
> > Thanks for your advice! Due to the 9 page limit of submission, we find it difficult to accommodate the change without massive reorganizing efforts in the current updated version. We will definitely address your concern in the future update.
> >
> >
> > [4] Korthikanti, Vijay, et al. "Reducing Activation Recomputation in Large Transformer Models." arXiv preprint arXiv:2205.05198 (2022).
> >
> > [5] Tao, Chaofan, et al. "Compression of Generative Pre-trained Language Models via Quantization." arXiv preprint arXiv:2203.10705 (2022).

---

> > > ### Author Response · Authors · 2022-12-02
> > > **Thanks for your great review!**
> > >
> > > Dear Reviewer,
> > >
> > > Thank you very much again for your kind and valuable reviews! We hope our previous response addresses your questions, from which this work has benefited a lot. If possible, we would be deeply grateful if you could help share suggestions to further make GLM-130B a stronger work!
> > >
> > > Thank you again for your encouraging comments!
> > >
> > > Authors of Paper3642

---

### Official Review · Reviewer_tevQ · 2022-11-04

**Confidence:** 4
**Correctness:** 3
**Technical Novelty And Significance:** 2
**Empirical Novelty And Significance:** 3
**Recommendation:** 8

**Clarity, Quality, Novelty And Reproducibility:**

Points of Clarification (did not impact the score):

p1 / Introduction - “However, both GPT-3 (and other 100B-scale ones)—the model itself—and how it can be trained, have been thus far not available to the public.”

It might be better that this be reframed for clarity, I think all the models describe how they are trained (i.e. methodology in their paper), two of these have weights released (OPT 175B on request and Bloom 176B on github here - https://huggingface.co/bigscience/bloom/tree/main), and atleast Bloom’s script is at https://github.com/bigscience-workshop/bigscience/tree/master/train/tr11-176B-ml#readme

p2 / Introduction — “associated with significantly less bias” any causal explanation?

p2 / Introduction — “unique property of GLM architecture” “INT4 quantization introduces very limited performance losses” : It is not clear that ability to do INT4 comes directly from GLM architecture, it could also be the other changes like training objective (MASK + gMASK) or initialization choices like DeepNorm etc or a combination of both. Fig 5 does explain why INT4 works for GLM-130B and not BLOOM, but the claim that this is due to the GLM architecture seems strong, accordingly “unique property” seems overly strong and unsubstantiated.

p2 / Introduction — “associated with significantly less bias and generation toxicity”: Would the authors make any causal explanation as to why this ought to be the case, i.e. the major contribution seems to be pre-training mechanics, I cannot see how “less bias” would follow, this might be empirically true, but lacking a causal explanation as to why this would be so (for example: data cleanup or filtering) this can only be taken as an incidental fact, rather than a recipe to have less bias.
p4 / Section 2.2 — “Self-Supervised Blank Infilling”: This section is slightly confusingly framed, as written it led me to believe that both [MASK] and [gMASK] are in the **same sequence** — it wasn’t until section 2 ending on p5 where it became clear that these are two separate tasks (due to the lengths being different in MASK and gMASK). I would advice to split the section “Self-Supervised Blank Infilling” into two, one for MASK, another for gMASK, or at least call out them separately within the Self-Supervised section.


Minor comments and typos (that did not impact the score):

 - p1 / Abstract - “disconvergence”, maybe write “divergence” since that’s the commonly used term? Or do you want to convey that disconvergence = opposite of convergence, so could be either divergence (loss steadily increasing) or NaN or just unstable loss?
 - p1 / Introduction - “capabilities suddenly arouse”, s/arouse/arise or arose.
   - Same comment as above on p9 / Sec 6 “arouse as models” → “arise as models”
   - p9 / Sec 7 Point 4 “arises” — however this point is poorly written
 - p1 / Introduction - “pioneers the studies of”, s/studies/study
 - p1 / Introduction - “we come to realize”, reframe as “we came to realize” or “we have come to realize”
 - p1 / Abstract & p2 / Introduction - “most ever affordable”, “most affordable” would suffice here.
 - p5 / Sec 3 - “We spend months” change to “We spent months”
 - p5 / Sec 3 - “in sacrifice of a significant penalty on model performance”, this is quite unclear, please reframe
 - p6 / Sec 3 - “first used in the multi-modal transformer CogView”, maybe just cite the cogview paper right there, sometimes paper title does NOT contain the name of the model, so the reference will be hard to find.
 - p8 / Sec 5.3 - “Here is out intuitive attempt” → “Here is our intuitive attempt” ; “we come up with” → “we came up with”
 - p8 / Sec 5.3 “if we ever got a chance to continu pre-training GLM-130B” - I would suggest summarizing under “future work”
 - p9 / Sec 6 - “despite many emerged 100B” cut out emerged
 - p9 / Sec 6 - “it has been concentrated on” reframe perhaps
 - p9 / Sec 7 - “architecture alternative, in addition to GPTs” →  “architecture alternative to GPTs” you are pointing out that GLM is an alternate way of self-supervision. I would rather frame it as an alternate “objective” instead.
 - p9 / Sec 7 Point 6 - “gradient to it’s 0.1” - ill framed sentence
 - p9 / Sec 7 Point 7 - “unique” is a strong word for here
 - p10 / Ethics - “And to estimate and better collaborate” no need for estimate
 - p10 / Reproducibility - PaLM citation is wrong, Paperno 2016 is mentioned instead of the PaLM citation.
 - p10 / Reproducibility - “We have paid great exertion”, better way to frame would be “We paid great effort”


**Strength And Weaknesses:**

Strengths:
 - Impressive set of techniques put together for stability and model quality (DeepNorm initialization, GLM instead of CausalLM objective, Embedding Layer Gradient Shrink)
 - Impressive performance on measured benchmarks esp compared to GPT3 & in a few cases PaLM

Weaknesses
 - Would have loved to see some ablations, especially over the first few ~100B tokens as to which technique contributes how much to performance, either within GLM-130B or implemented across models, i.e. for example with T5 style models the GLM objective would be essentially T5’s denoising task + prefixLM. Otherwise it is not clear what technique is responsible for how much of the gains.
 - More evaluations: The evaluation section is severely limited
   - It could have evaluated fine-tuning over a few tasks (Ex: PaLM has both few-shot numbers and fine-tuning numbers for a few benchmarks like SuperGlue and TyDiQA) — this is understandably partly a resource issue.
   - More extensive few-shot evaluations on say SuperGlue, Winograd style tasks, CBQA style tasks, CommonSense reasoning, Chain of Thought prompting etc — Since inference is both fast and cheap this shouldn’t be too much of an issue and would increase my confidence of this method’s applicability and more importantly in the released checkpoint.
   - Not evaluating on tasks with fixed labels (like NLI) seems like severely and unnecessarily restrictive, i.e. I don’t think Xian et al 2018’s advice applies here, one can/should do NLI tasks with encoding the label in the prompt, while not giving an example (for zero-shot) or giving a few examples of each class (in a few-shot setting) and evaluating. Since MultiTask Instruction Pre-training (MIP) is happening here, I think it is fair to only exclude tasks that are substantially similar in the actual example content, if the task type seen is similar (say classification with the same labels) this should be called out and reported separately.


**Summary Of The Paper:**

The paper trains a new 130B parameter model with the GLM architecture/objective, with contributions in automatically stabilizing spikes with existing techniques, focussing on keeping inference costs and requirements low and therefore the model accessible to a large number of people. It is an achievement to beat GPT3 numbers and in a few cases PaLM numbers.

**Summary Of The Review:**

Weak Accept owing to a severely limited evaluation section and methodology, the work itself pulls a lot of techniques together to get a stably pre-trained O(100B) model with GLM objective and stability techniques which is very valuable to get broad recognition.

---

> ### Author Response · Authors · 2022-11-19
> **Response to Reviewer tevQ (1/2)**
>
> Thank you very much for your appreciation and great suggestions for our work! During the past two weeks, we have been performing extensive experiments (within our resource constraint) including more technical ablation studies, more comprehensive few-shot evaluation (on **40+ new datasets**), and fix-label evaluation.
>
>
> ### Technique Ablations
>
> We managed to test some of the technical design choices during the submission, which are listed below for your reference:
>
> * Ablation on ordinary **PostLN and DeepNorm**: Page 3 Figure 3
> * Ablation on **Bidirectional/Unidirectional Attention** (LAMBADA): Page 3 Figure 2
> * Ablation on **Embedding Layer Gradient Shrink** (EGS): Page 5 Figure 4
> * Ablation on **Positional Encodings and FFN**: Appendix B.3 Table 8
>
> Similar to existing 100B-scale model attempts, it is unfortunate that we can't afford all possible ablations for training at this scale. But as a substitute, we try our best to conduct an ablation study at 10B-scale on the two main techniques — **the GLM objective and MIP** (Multi-task Instruction Pre-training) — that have a significant impact on the performance (Cf. Appendix B.9 Figure 14).
>
> From Figure 14 (https://i.imgur.com/PRlVmtL.png), we observe that bidirectional attention (i.e., denoising objectives and PrefixLM) in GLM makes a major contribution to the performance improvement (e.g., in LAMBADA it contributes 73% and in MMLU it contributes 90%). MIP also helps the improvements (especially on NLI-related tasks), but is not as significant as bidirectional attention in other harder challenges (e.g., MMLU), which is probably because MIP only accounts for 5% of tokens in the pre-training.
>
> In addition, we perform a series of new ablation studies on bidirectional and unidirectional attention’s effect on various tasks beyond LAMBADA, including Conditional Text Generation (Cf. Appendix C.8 and Table 16), SuperGLUE (Cf. Appendix C.13 Figure 17).
>
> ### More Comprehensive Evaluation
>
> In the original submission, we evaluated GLM-130B over **112 tasks** in total, covering a wide range of domains and application scenarios. By following your suggestion, we perform evaluations for GLM-130B and other available LLMs over **another 40+ datasets** in this rebuttal, including **Winograd-style tasks, CBQA, Commonsense reasoning, Chain-of-thought prompting, Conditional Text Generation, Emergent abilities, SuperGLUE, **and** Scaling properties.** If any other important types of tasks come out, we will continue to update new results in the future versions, with our initial goal of pre-training and evaluating an open and highly-accurate 100B-scale LLM in mind.
>
> **Commonsense Reasoning**
>
> We implement GLM-130B and related comparison LLMs on CommonsenseQA and MC-TACO. The results are reported as below (Cf. Appendix C.11 Table 19).
>
> |                      | shots    | GPT-3 (Davinci) | OPT 175B | BLOOM 176B | GLM-130B |
> | -------------------- | ---- | --------------- | -------- | ---------- | -------- |
> | Commonsense QA (Acc) | 0    | 57.2            | -        | 42.8       | 61.6     |
> | Commonsense QA (Acc) | 1    | 61.2            | -        | -          | 62.2     |
> | MC-TACO (EM)         | 0    | -               | 12.4     | 13.1       | 13.6     |
>
> **Chain-of-Thought Prompting**
>
> We implement GLM-130B on some typical reasoning datasets, including Sports Understanding, Last Letter Concatenation (LLC), Coin Flip, Coin Flip (OOD: 3), Reverse List, and Date Understanding. Due to the 2-week limit, we have not spared to test on other datasets, and we will report them in the future version. Please see Appendix C.14 and Figure 18 (https://i.imgur.com/4bZf3Sx.png) for details.
>
> **Winograd-style Tasks**
>
> We implement GLM-130B and related comparison LLMs on Winograd273 and Winogender. Since Winogrande is unfortunately included in our MIP training, we cannot evaluate it. Please see Appendix C.9 and Table 17 for details.
>
> |             | shots    | GPT-3 (Davinci) | OPT 175B | BLOOM 176B | PaLM 540B | Chinchilla | Gopher 280B | GLM-130B |
> | ----------- | ---- | --------------- | -------- | ---------- | --------- | ---------- | ----------- | -------- |
> | Winogender  | 0    | 64.2            | 54.8     | 49.1       | 75.0*     | 78.3       | 71.4        | 79.7     |
> | Winogender  | 1    | 62.6            | -        | 53.1       | 79.4      | -          | -           | 80.7     |
> | Winograd273 | 0    | 88.3            | 52.9     | 49.1       | 90.1      | -          | -           | 84.3     |
>
> *PaLM 540B did not report the exact 0-shot Winogender result, so we have to estimate a value from its plotted diagram

---

> > ### Author Response · Authors · 2022-11-19
> > **Response to Reviewer tevQ (2/2)**
> >
> > **Closed-book Question Answering**
> >
> > We implement GLM-130B and related comparison LLMs on Natural Questions and StrategyQA. Since TriviaQA and WebQuestions used in GPT-3 are unfortunately included in our MIP training, we cannot evaluate them in the zero-shot manner. Please see Appendix C.10 and Table 18 for details.
> >
> > |                       | GPT-3(Davinci) | BLOOM 176B | PaLM 540B | Chinchilla | Gopher 280B | GLM-130B |
> > | --------------------- | -------------- | ---------- | --------- | ---------- | ----------- | -------- |
> > | Natural Questions(EM) | 14.6           | 13.1       | 21.2      | 16.6       | 10.1        | 11.7     |
> > | StrategyQA(Acc)       | 52.3           | 49.8       | 64.0      | -          | -           | 60.6     |
> >
> > **SuperGLUE**
> >
> > Since almost all of SuperGLUE datasets have been included in the MIP training (except ReCoRD), the results of GLM-130B on SuperGLUE are not for comparison to others. We mainly compare between the performance of unidirectional and bidirectional GLM-130B on SuperGLUE as an ablation. Please see Appendix C.13 and Figure 17 for details.
> >
> > **Scaling Property and Emergent Abilities in GLM-130B**
> >
> > We also conducted extensive experiments to explore the scaling property and emergent abilities in GLM-130B. Following prior literature, we categorize the NLP tasks into two types based on our observations (Cf. Appendix D and Figure 19, 20 for details):
> >
> > * Log-scaling Ability Tasks (https://i.imgur.com/rPxpLoy.png): where the task performance grows logarithmically with the amount of model parameters. Typical tasks and datasets include: LAMBADA, Wikitext-103, Wikitext-2, Penn Tree Bank.
> > * Emergent Ability Tasks (https://i.imgur.com/2krQgeR.png): where the task performance only soars up when the amount of model parameters reaches a certain threshold. Typical tasks and datasets include: MMLU, hindu_knowledge, crass_ai, implicatures, understanding_fables, modified_arithmetic, implicit_relations, and gre_reading_comprehension.
> >
> > We notice that emergent abilities usually arise in long-tailed knowledge tasks and often require LLMs’ ability to memorize some rare patterns.
> >
> > **Fine-Tuning Results**
> >
> > Thanks for your understanding in advance! The computing resource needed to fine-tune such a giant LLM is really beyond our immediate capacity in the short response period. We will definitely attempt to find resources to conduct the fine-tuning experiments in the near future.
> >
> >
> > **Conditional Text Generation**
> >
> > We implement GLM-130B on 3 typical conditional text generation tasks, including WebNLG, E2E (Data-to-text) and WikiLingua (Summarization) following the practice in PaLM-540B. Please see Appendix C.8 and Table 16 (https://i.imgur.com/GVfqZqJ.png) for results. We do not include Xsum since it is used in the MIP training.
> >
> > ### Fixed-Label Task Evaluation
> >
> > Thanks a lot for the suggestion!
> >
> > We agree and realize that it is indeed better to report MIP-similar tasks’ results separately in a new subsection to avoid confusion. Frankly speaking, in such a setting GLM-130B’s zero/few-shot learning could be quite advantageous. Below, we take the Natural Language Inference as a typical example to show GLM-130B’s outperformance (Cf. Appendix C.12 for more details).
> >
> > |                                              | BLOOM 176B | OPT 175B | GLM-130B |
> > | -------------------------------------------- | ---------- | -------- | -------- |
> > | qnli(valid, median of 5 prompts)             | 50.9       | 55.4     | 86.7     |
> > | mnli(valid, median of 15 prompts)            | 35.5       | 36.0     | 85.7     |
> > | mnli_mismatched(valid, median of 15 prompts) | 35.5       | 36.0     | 84.6     |
> > | wnli(valid, median of 5 prompts)             | 57.7       | 53.5     | 67.6     |
> > | glue/cola(valid, median of 5 prompts)        | 39.0       | 44.4     | 57.6     |
> > | glue/mrpc(valid, median of 5 prompts)        | 31.6       | 44.6     | 87.3     |
> >
> > As observed, despite these datasets not being included in the MIP training, GLM-130B’s “zero-shot” performance could be much better due to the seen task type.
> >
> > ### Writing
> >
> > Thanks for your detailed corrections and suggestions! We have them fixed in the updated version.

---

> > > ### Comment · Reviewer_tevQ · 2022-11-22
> > > **Thank you**
> > >
> > > Hi All - Thank you for the responses and running many ablations and organizing the existing ablations better. I will update the score, since I'm quite satisfied with the responses.
> > >
> > > >  We will definitely attempt to find resources to conduct the fine-tuning experiments in the near future.
> > >
> > > I think this is *very* important -- many LLMs have trouble in being fine-tuned without extensive hyper-parameter search, it will be interesting to know how GLM-130B stands with this regard.
> > >
> > > A few further very minor comments:
> > >
> > > Appendix B.3 -- "RoPE (Chowdhery et al., 2022)" makes it look like Chowdhery et al invented RoPE, maybe correct the citation? If the intention was that RoPE was used in PaLM, then maybe rephrase as "RoPE as used in PaLM (Chowdhery et al., 2022)"
> > >
> > > Table 12 - "are descripted in Appendix C.2." s/descripted/described
> > >
> > > C.12 - "We include 6 widely-used NLI dataset–which are not incorporated in GLM-130B’s MIP training–as the benchamrks", replace - with commas "," and s/benchamrks/benchamarks
> > >
> > > Several places s/Chain-of-Thoughts/Chain-of-Thought i.e. no "s" in Thoughts
> > >
> > > Sec 5.3 “we come to realize”, reframe as “we came to realize” or “we have come to realize”
> > >
> > > A few typos that I previously mentioned still show up:
> > >
> > > Ex:
> > > p1 / Introduction - “capabilities suddenly arouse”, s/arouse/arise or arose.
> > >
> > > Can the authors do another pass to clean them up?

---

> > > > ### Author Response · Authors · 2022-11-25
> > > > **Thanks for your suggestions**
> > > >
> > > > Thanks a lot for your suggestions to improve this work! We appreciate your acknowledgment of the effort!!
> > > >
> > > > Yes, we very much agree that fine-tuning LLMs faces many challenges compared to smaller ones. These challenges come not only from computational efficiency but also from algorithms. Our initial attempts show that GLM-130B tends to overfit quickly within 1-2 epochs, resulting in poor val/test set performance. We will try to explore more when more resources are available. We also think that parameter-efficient (PE) fine-tuning methods may be another choice for LLMs since they require much fewer resources and are less prone to overfit. It would be interesting to compare PE methods with the full-parameter fine-tuning in LLMs. We are currently running experiments on PE methods, and we will update the results as soon as we make any progress.
> > > >
> > > > Finally, thanks again for your careful suggestions. We have them fixed and will update the paper in our next version.

---

### Author Response · Authors · 2022-11-19
**General Response**

Thank you all reviewers for the deep and constructive feedback! The reviews really help a lot to further improve this work! Briefly, the reviewers suggest having more evaluations. During the two-week rebuttal period, we have performed extensive experiments on **40+ new datasets in addition to experiments on the 112 tasks reported in the initial submission**. We also added more ablation studies (10B-scale) and fix-label evaluations as suggested. Again, thanks for the suggestions! We are very happy to have as many evaluations as possible to examine and reach the goal of this work---to pre-train an open and highly-accurate 100B-scale model. The new results are reported in the new version of the submission (+9 pages). All additional experiments, including logs of preliminary experiments, are also updated to the anonymous repository (https://anonymous.4open.science/r/GLM-130B/).

---

### Decision · Program_Chairs · 2023-01-20

**Decision:**

Accept: poster

**Justification For Why Not Higher Score:**

This paper surely deserves to be accepted, but I don't think it's very interesting for an oral since it's a technical report. I would prioritize more interesting papers for oral/spotlight.



**Justification For Why Not Lower Score:**

This paper is a good one that should be accepted.

**Metareview: Summary, Strengths And Weaknesses:**

All reviews liked the paper which includes comprehensive experiments and details. The reviewers also liked the amount of engineering effort behind this paper. Major plus for open source!

From reading the reviews and paper, I think this is a pretty neat technical report that could be accepted to ICLR.

Therefore, I recommend acceptance.

**Note From Pc:**

if the above contains the word "oral" or "spotlight" please see: "oral" presentation means -> notable-top-5% and "spotlight" means -> notable-top-25%. As stated in our emails, we are disassociating presentation type from AC recommendations